# Mechanically compliant and cost-effective 1.4Li$_2$O-0.75ZrCl$_4$-0.25AlCl$_3$ solid electrolyte for all-solid-state batteries with improved cycling stability

Lv Hu [1,2,17], Yaolong He [3,17], Dong Wang[4], Wanxia Li[5], Jingming Yao[6], Xiaolong Zhang[7], Jinfeng Zhu[8], Huaican Chen[9], Wen Yin[9], Yanru Wang[10], Kejun Yan[11], Jinzhu Wang[1], Hui Li[1], Fang Chen[1], Yating Liu[1], Junqi Lai[12], Qi Chen [12], Jie Ma [8], Shuhong Jiao [5], Guorui Wang [4], Siqi Shi [3] ✉, Liwei Chen [13,14,15], Jianyu Huang [6] & Cheng Ma [1,2,16] ✉

Although Li-ion conductivity has been the primary focus during decades of solid-electrolyte research, the mechanical compliance is equally important. For most state-of-the-art solid electrolytes, the mechanical compliance is characterized by the hardness above 1 GPa and Young's modulus above 15 GPa. Here, we report a particularly compliant solid electrolyte, 1.4Li$_2$O-0.75ZrCl$_4$-0.25AlCl$_3$, whose hardness and Young's modulus reach 0.22 and 1.41 GPa, respectively. Meanwhile, it shows an ionic conductivity of 2.55 mS cm$^{-1}$ at 25 °C and an estimated cost of \$43.70 L$^{-1}$, considerably lower than that of the Li$_2$ZrCl$_6$ solid electrolyte known for cost-effectiveness (\$140.01 L$^{-1}$). The improved mechanical compliance and fast Li-ion transport in 1.4Li$_2$O-0.75ZrCl$_4$-0.25AlCl$_3$ enable decent cell performance. With high positive electrode active material loading above 20 mg cm$^{-2}$, these two types of cells achieve areal capacities of 3.62 mAh cm$^{-2}$ (85.78% capacity retention) and 3.92 mAh cm$^{-2}$ (90.11% capacity retention), respectively, after 100 cycles under 0.1 C at 25 °C. The simultaneous achievement of highly competitive mechanical compliance, Li-ion conductivity, and cost-effectiveness in 1.4Li$_2$O-0.75ZrCl$_4$-0.25AlCl$_3$ have the potential to pave the way for the realization of commercial, practical all-solid-state Li batteries.

Efficient Li-ion transport, high mechanical compliance, and strong cost-competitiveness are three indispensable characteristics for the solid electrolyte in all-solid-state Li batteries (ASSLBs)[1–3]; while other properties like electrochemical stability and Li dendrite suppressing capability are replaceable by alternative approaches such as electrode coating[4–6] and Ag-C interlayers[7], respectively, these characteristics cannot. Among the three of them, the importance of mechanical compliance to the overall cell performance is in fact no lower than that of the much more extensively studied Li-ion conductivity, because only the highly compliant solid electrolytes can deform efficiently under pressure to maintain the intimate contact with the repeatedly expanding and contracting electrode active materials[8–10].

The mechanical compliance of solid electrolytes would become particularly essential under the cycling conditions that are closer to practical situations. Presently, to demonstrate the advantages associated with the increasingly high ionic conductivities of solid

electrolytes (many exceeding 1 mS cm$^{-1}$ at 25 °C[11–18], and some even above 10 mS cm$^{-1}$ at 25 °C[4,19–22]), the cells with typical positive electrodes (e.g., LiCoO$_2$ and LiNi$_{0.8}$Co$_{0.1}$Mn$_{0.1}$O$_2$) are often cycled at 1 C −5 C in literature[3,12,23–26], while higher rates of 20 C−50 C were also adopted occasionally[27–30]. Nevertheless, in practical situations, high-rate cycling is not the only demand. Taking electric vehicles as an example, although the charge still needs to be as fast as possible, the discharge does not. Assuming a vehicle with a 600 km range drove no faster than 120 km h$^{-1}$, the average discharge rate would not exceed 0.2 C, so most of the time the actual discharge rate should vary between 0.1 C and 0.3 C, much lower than the frequently used 1 C−5 C rates for testing[3,12,23–30]. Such low cycling rate is usually associated with larger capacity; for example, the capacity of LiNi$_{0.88}$Co$_{0.07}$Mn$_{0.05}$O$_2$ at 0.2 C has been reported to be 178.6% of that at 4 C[12]. Therefore, more Li ions would be actually inserted to or extracted from the crystal at low cycling rates, making the positive electrode active material (PEAM) particles undergo larger volume change. To sustain the intimate contact with the PEAM particles under these circumstances, the solid electrolytes have to be much more compliant. In particular, such demand would be more compelling when the areal capacity increases from the commonly adopted 1 mAh cm$^{-2}$ for laboratory testing to the level for enabling high energy densities (around or above 4 mAh cm$^{-2}$).

In order to rationally improve the mechanical compliance, the metrics for this characteristic need to be properly identified first. In the studies on mechanical properties, the Young's modulus has been frequently emphasized, but it is not the only important metric for mechanical compliance[31,32]. In fact, what Young's modulus describes is the elastic deformation, which is often limited to the initial, low-stress stage of the solid-electrolyte deformation during cycling[8,10,33]. When the stress is sufficiently large, plastic deformation would occur. The resulting plastic flow would enable much more efficient morphology change than elastic deformation, and thus usually plays a main role in sustaining the intimate solid-solid contact during cycling[8,10]. Therefore, how difficult it is to induce plastic deformation in a solid electrolyte (reflected by its hardness) is also a critical metric. In literature, the hardness and Young's modulus have been identified as the two major factors underpinning the intactness of the electrode-electrolyte contact of all-solid-state cells during cycling, and preferably both of them should be as low as possible[10]. However, the present inorganic solid electrolytes known to be highly compliant, i.e., the sulfide and halide solid electrolytes, generally show the hardnesses above 1 GPa and Young's moduli above 15 GPa[3,8,34–37]. Besides, most (if not all) of these solid electrolytes may be too costly to enable the successful commercialization of ASSLBs[38].

Here, we report a 1.4Li$_2$O·0.75ZrCl$_4$·0.25AlCl$_3$ solid electrolyte, which simultaneously achieves the hardness and Young's modulus of 0.22 and 1.41 GPa, respectively, at a decent cost. Combined with a high ionic conductivity of 2.55 mS cm$^{-1}$ at 25 °C in the meantime, this mechanical compliance allows the all-solid-state cell with LiNi$_{0.92}$Co$_{0.06}$Mn$_{0.02}$O$_2$ to deliver an areal capacity of 3.92 mAh cm$^{-2}$ and a capacity retention of 90.11% after 100 cycles at 25 °C.

## Results
### Materials design principles and Li-ion transport behaviors
Our materials design strategy for realizing high mechanical compliance is based on two principles. First of all, compared with the materials with only one type of anion, those with two types of anions have often been found to display the kind of atomic configuration that enables higher mechanical compliance[25,39,40]. In such materials, the coexisting anions frequently make the cation-anion polyhedra more distorted and disordered, so that the backbone of the atomic framework becomes more easily rotated or bended, leading to enhanced mechanical compliance[25,39,40]; this phenomenon has been observed in multiple solid electrolytes, including the O-modified Na$_3$PS$_4$[40], Li$_2$ZrCl$_6$[3,25], and LiAlCl$_4$[39]. Based on this principle, oxysulfides and

oxychlorides are both promising candidates. However, since oxysulfides are rather difficult to meet the cost-effectiveness requirement for commercialization due to the expensive raw material Li$_2$S[38], the present study will focus on oxychlorides. Secondly, among the oxychloride solid electrolytes mentioned above, the amorphous status is often associated with much higher mechanical compliance than the crystalline form[3,25,39,41,42]. Taking this phenomenon into account too, our focus is placed on amorphous oxychlorides. While this type of material is promising in achieving decent mechanical compliance, the other two "indispensable characteristics" mentioned in the Introduction, i.e., efficient Li-ion transport and strong cost-competitiveness, must also be considered. In order to realize Li-ion conductivity above 1 mS cm$^{-1}$ at 25 °C, the oxychlorides usually need to contain non-Li cations[3,16,21,24,25,42–45]. Recently, Ta$^{5+}$ was found to be a particularly effective non-Li cation for realizing fast Li-ion transport; Ta-based oxychlorides such as 1.6Li$_2$O·TaCl$_5$ and LiTaOCl$_4$ display ionic conductivities up to 12.4 mS cm$^{-1}$ at 25 °C[21,24]. However, the raw material TaCl$_5$ is known to be too costly to enable the commercialization of ASSLBs[38]. Fortunately, the ionic radius of Ta$^{5+}$ happens to lie between those of Zr$^{4+}$ and Al$^{3+}$[46], whose raw materials ZrCl$_4$ and AlCl$_3$ are both rather cost-effective[42,47]. If Zr$^{4+}$ and Al$^{3+}$ coexist in the amorphous oxychloride at a proper molar ratio, the non-Li cations may reach an effective ionic radius close to that of Ta$^{5+}$, but without inducing unacceptably high costs. Therefore, with the mechanical compliance, Li-ion conductivity, and cost-effectiveness all taken into account, the amorphous $x$Li$_2$O·(1-$y$)ZrCl$_4$·$y$AlCl$_3$ oxychlorides seem a highly promising materials system.

Based on this hypothesis, we tried to prepare a series of amorphous $x$Li$_2$O·(1-$y$)ZrCl$_4$·$y$AlCl$_3$ oxychlorides by planetary milling the stoichiometric mixtures of Li$_2$O, ZrCl$_4$, and AlCl$_3$ for 30 h (milling parameters are specified in Methods). As shown in Fig. 1a and Supplementary Fig. 1, the materials with $x$ = 1.0−1.8 and $y$ = 0.2−0.5 display high ionic conductivities that are close to or above 1 mS cm$^{-1}$ at 25 °C, and their electronic conductivities are orders of magnitude lower (Supplementary Fig. 2), confirming that these oxychlorides can serve as solid electrolytes in ASSLBs. For each series of compositions with a given $y$, the ionic conductivity peaks at $x$ = 1.4 (Fig. 1a). Among these $x$ = 1.4 compositions with different $y$ values, the most efficient Li-ion transport is realized at $y$ = 0.25 (Fig. 1b, c), i.e., the 1.4Li$_2$O·0.75ZrCl$_4$·0.25AlCl$_3$ oxychloride (abbreviated as LZACO below); it shows an ionic conductivity of 2.55 mS cm$^{-1}$ at 25 °C and an activation energy of 0.276 eV (the corresponding Arrhenius plots are displayed in Supplementary Fig. 3). Consistently with the hypothesis raised above, the Zr/Al molar ratio in this optimal composition happens to make the equivalent ionic radius of non-Li cations close to that of Ta$^{5+}$ (0.67 vs. 0.64 Å for the coordinate number 6, and this holds for other coordinate numbers too[46]), whose presence in oxychlorides is often accompanied by high Li-ion conductivities[21,24].

In addition to the ionic conductivity, the structures also meet our expectations. The X-ray diffraction (XRD) patterns suggest that all of the materials discussed above are indeed almost completely amorphized (a few representative examples shown in Fig. 1d, e). The pair distribution function (PDF) analysis based on neutron total scattering is also indicative of the same scenario. Supplementary Fig. 4 shows the PDF of LZACO obtained by neutron total scattering, as well as the simulated PDFs of LZACO with the $P\bar{3}m1$ structure and that with the $C2/m$ structure. In comparison with the two crystalline forms of LZACO, the actual LZACO displays much weaker and broader peaks in the PDF, confirming its high degree of amorphization. Beyond the qualitative evaluations above, we also conducted high-resolution transmission electron microscopy (TEM) observation to probe the volume fraction of the amorphous species in LZACO. As shown in Supplementary Fig. 5a, the volume of the crystalline phase in LZACO is so low that almost no lattice fringe is visible in the as-collected, unprocessed high-resolution TEM (HRTEM) image. Only after the fast Fourier transform

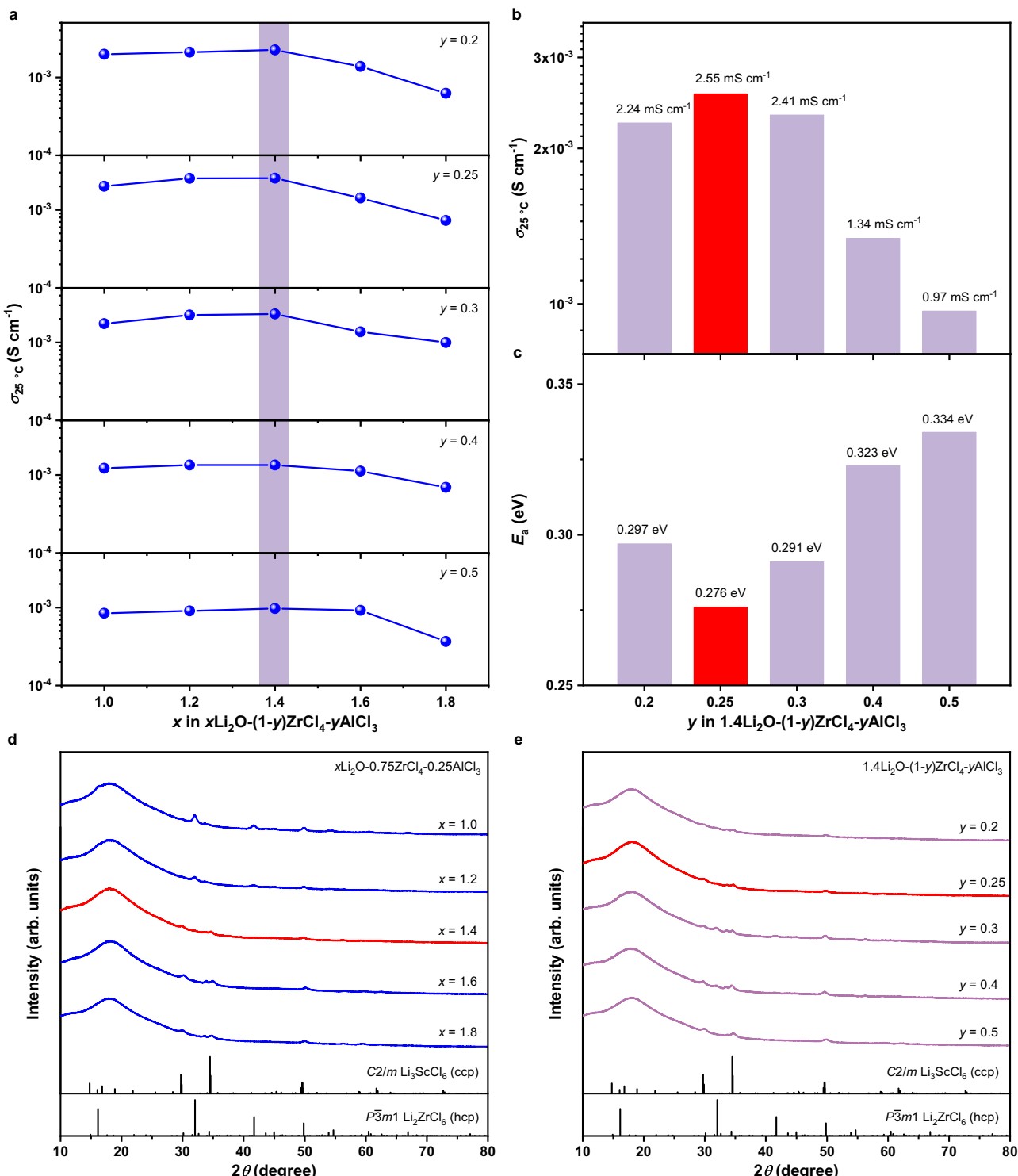

**Fig. 1 | Li-ion transport behavior and structures of $x$Li$_2$O·(1-$y$)ZrCl$_4$·$y$AlCl$_3$.** **a** Ionic conductivities of $x$Li$_2$O·(1-$y$)ZrCl$_4$·$y$AlCl$_3$ at 25 °C. **b, c** Ionic conductivities at 25 °C (**b**) and activation energies (**c**) of $x$Li$_2$O·(1-$y$)ZrCl$_4$·$y$AlCl$_3$ with $x$ = 1.4. **d** XRD patterns of $x$Li$_2$O·(1-$y$)ZrCl$_4$·$y$AlCl$_3$ with $y$ = 0.25. **e** XRD patterns of $x$Li$_2$O·(1-$y$)ZrCl$_4$·$y$AlCl$_3$ with $x$ = 1.4.

(FFT), the two weak spots indicating the existence of crystallites are disclosed (Supplementary Fig. 5b). The size and morphology of the crystalline regions can be probed by conducting inverse FFT using these two weak spots. As shown in Supplementary Fig. 5d, such processing discloses one relatively large crystalline region at the left side of the image and several scattered, small ones. Unlike the former, the latter should be the artifacts resulting from the inverse FFT processing, rather than true crystallites; when FFT was conducted locally at

different regions in Supplementary Fig. 5c, the aforementioned relatively large crystalline region is in fact the only one that can give rise to spots in the FFT pattern (Supplementary Fig. 5e), and the same happens to none of the other regions (example shown in Supplementary Fig. 5f). In this way, Supplementary Fig. 5d in fact suggests that the size of the rare crystallites in LZACO is about 15 nm. Such a crystallite size explains why the PDF of the highly amorphous LZACO material (Supplementary Fig. 4) still shows weak peaks characterizing ordered

crystalline structures up to $r = 30$ Å (3 nm). More importantly, Supplementary Fig. 5d also indicates that the crystalline phase only occupies 10 vol% in LZACO, with the remaining volume being amorphous. It must be emphasized that the compositions of such minor crystalline species are not necessarily identical with those of the amorphous matrix. Instead, it might be the Li-M-Cl (M = Zr and/or Al) precipitates, which are much more difficult to amorphize than the oxychlorides[25,42]. The precise composition determination for the minor crystalline phases here would require the elemental mapping at magnifications no lower than those in the HRTEM image presented above (Supplementary Fig. 5a), but the intense electron beam needed for such characterization would rapidly damage the electron-beam-sensitive oxychloride solid electrolytes. Consequently, the composition of the crystalline phase cannot be conclusively determined. Fortunately, their volume fraction (about 10% as mentioned above) is too low to considerably influence the overall ion transport. Although the low crystallinity of $x$Li$_2$O·(1-$y$)ZrCl$_4$·$y$AlCl$_3$ prevents the precise crystal structure determination, the XRD patterns may still disclose certain information regarding the atomic configuration. Generally speaking, the characteristic reflections of up to two types of crystal structures can be identified; both have been frequently observed in the Li$_3$MCl$_6$-type chloride solid electrolytes[48]. One of them is the structure showing the hexagonal close-packed (hcp) anion arrangement; the materials displaying this kind of structure include the Li$_2$ZrCl$_6$ with the $P\bar{3}m1$ space group[47,49] and the Li$_3$YCl$_6$ with the $Pnma$ space group[48]. The other is the structure showing the cubic close-packed (ccp) anion arrangement; the materials displaying this type of structure include the LiCl with the $Fm\bar{3}m$ space group[48] and the Li$_3$ScCl$_6$ with the $C2/m$ space group[50]. Due to the low crystallinity of LZACO, it is difficult to conclusively determine which specific structures mentioned above is displayed by each phase. Consequently, the two crystalline phases present in LZACO would be generally referred to as the hcp and ccp phases, respectively, in the discussion below. For the $x$Li$_2$O·(1-$y$)ZrCl$_4$·$y$AlCl$_3$ materials studied here, the hcp phase would transform into the ccp phase as either $x$ or $y$ increases, with the intermediate compositions displaying these two phases simultaneously (Fig. 1d, e). Similar to a recently reported high-performance solid electrolyte Li$_{1.75}$ZrCl$_{4.75}$O$_{0.5}$[3], the most Li-ion conductive LZACO material identified above is also an "intermediate composition" showing the coexisting hcp and ccp phases. The simultaneous presence of these energetically comparable states is believed to make the material more easily amorphized[3], while the resulting large contents of amorphous species have been demonstrated to be the major enabling factor for the fast Li-ion transport in multiple Zr-based oxychlorides[3,16,25,26,44,45]. Likewise, the high ionic conductivity of LZACO also originates from the amorphous species. Once the crystallinity is improved by annealing at 300 °C for 5 h, its ionic conductivity would drop by more than one order of magnitude (Supplementary Fig. 6). Although both LZACO and the previously reported Li-Zr-Cl-O oxychlorides rely on the amorphous phase to realize fast Li-ion transport, the specific atomic configurations in their amorphous phases are not the same, which leads to different ion transport efficiencies. In the Li-Zr-Cl-O oxychlorides, the averaged Zr-O and Zr-Cl bond lengths in the distorted Zr-O/Cl polyhedra are believed to be the key to the high ion transport efficiency, as they flatten the energy landscape for Li-ion migration to lower the overall migration barriers[25]. Such a disordered atomic arrangement also occurs in LZACO, but to a more pronounced degree. To characterize this behavior, we conducted X-ray absorption spectroscopy (XAS). As shown in Supplementary Fig. 7a, the Zr $K$-edge X-ray absorption near-edge structure (XANES) spectrum of LZACO does not show distinct peaks like those of ZrO$_2$ and ZrCl$_4$, but appears as a broad hump, entailing that the Zr-Cl and Zr-O bond lengths in LZACO vary in a range, instead of displaying definitive values. On this basis, more interesting behaviors were observed in the Fourier transformed extended X-ray absorption fine structure (FT-EXAFS) spectra. Unlike other Li-Zr-Cl-O

solid electrolytes[25], the Zr-O and Zr-Cl signals in LZACO are not distinguishable from each other, but nearly merge into one broad peak (Supplementary Fig. 7b). This observation suggests that the Zr-O and Zr-Cl bond lengths in LZACO distribute in a broader range than those in the Li-Zr-Cl-O solid electrolytes[25]. Besides, the FT-EXAFS spectrum of LZACO almost shows no signals above 2.5 Å, entailing the absence of ordering with coherence length exceeding such a distance; in contrast, other Li-Zr-Cl-O solid electrolytes still display minor signals in this range[25]. Consistent results were observed in the phase-uncorrected wavelet-transformed (WT) EXAFS spectrum, too. While the Zr-O and Zr-Cl bonding still give rise to two distinct signals, respectively, in the Li-Zr-Cl-O solid electrolytes[25], they merge together in LZACO (Supplementary Fig. 7c). Besides, the Zr-Zr bonding at about 3 Å in the WT-EXAFS spectra of typical Li-Zr-Cl-O solid electrolytes are absent in that of LZACO too (Supplementary Fig. 7c), confirming the absence of the ordering with such large coherence lengths. These observations consistently suggest that LZACO shows a more disordered atomic configuration than the previously reported Li-Zr-Cl-O solid electrolytes. As mentioned above, such more disordered state should supposedly favor fast Li-ion transport in Zr-based oxychlorides[25], which explains the improved ionic conductivity of LZACO with respect to those of other Li-Zr-Cl-O solid electrolytes (2.55 vs. 1–2 mS cm$^{-1}$ at 25 °C[25]).

Although the efficient ion transport of LZACO has been confirmed to rely on the amorphous phase that would inevitably crystallize at sufficiently high temperatures, it must also be emphasized that the temperatures typically needed for the fabrication of pouch-cell membranes, i.e., 80–100 °C, are still too low to cause any non-negligible conductivity degradation. To demonstrate this point, we annealed LZACO at 100 °C for 5 h, and probed the associated change of the XRD pattern and ionic conductivity. As shown in Supplementary Fig. 8a, the content of the hcp phase appears to increase slightly with respect to that of the ccp phase after annealing; as the two coexisting phases in LZACO, the hcp and ccp phases must be energetically comparable in this material, so a relatively low energy input, such as that associated with the 100 °C annealing, should be sufficient to alter the ratio between them. Regardless, both of these crystalline phases are still present after annealing, and, more importantly, they continue to occupy a negligibly small fraction in LZACO, as indicated by the diffuse Bragg reflections in the XRD pattern (Supplementary Fig. 8a). Since the majority of LZACO remains the highly Li-ion conductive amorphous species, the material still maintains a high ionic conductivity of 2.23 mS cm$^{-1}$ at 25 °C, almost identical with that before annealing (2.55 mS cm$^{-1}$ at 25 °C), as shown in Supplementary Fig. 8b. This observation confirms that the 80–100 °C temperatures needed for fabricating the pouch-cell membranes will not compromise the high Li-ion conductivity of LZACO.

## Mechanical compliance

Beyond the efficient Li-ion transport demonstrated above, the solid electrolyte must also display high mechanical compliance, to ensure the formation and maintenance of the intimate solid-solid contact during cycling. As mentioned in the Introduction, the achievement of such high compliance requires not only the Young's modulus, but also the hardness to be as low as possible. Fig. 2a shows the hardness and Young's modulus measured for LZACO. For comparison, the data of several other compliant solid electrolytes are also plotted. The Young's moduli of a few such solid electrolytes were measured too, and the results we obtained align well with those reported in literature (Supplementary Fig. 9), confirming the reliability of our mechanical characterization. As for the hardness, the values for Li$_{1.75}$ZrCl$_{4.75}$O$_{0.5}$ and Li$_2$ZrCl$_6$ are too high to be measured: when nanoindentation was conducted, their cold-pressed pellets did not deform, but collapsed locally into small pieces and loose powders like the cold-pressed pellets of the hard, brittle oxide solid electrolytes, preventing the generation of meaningful load-displacement curves. In contrast, the

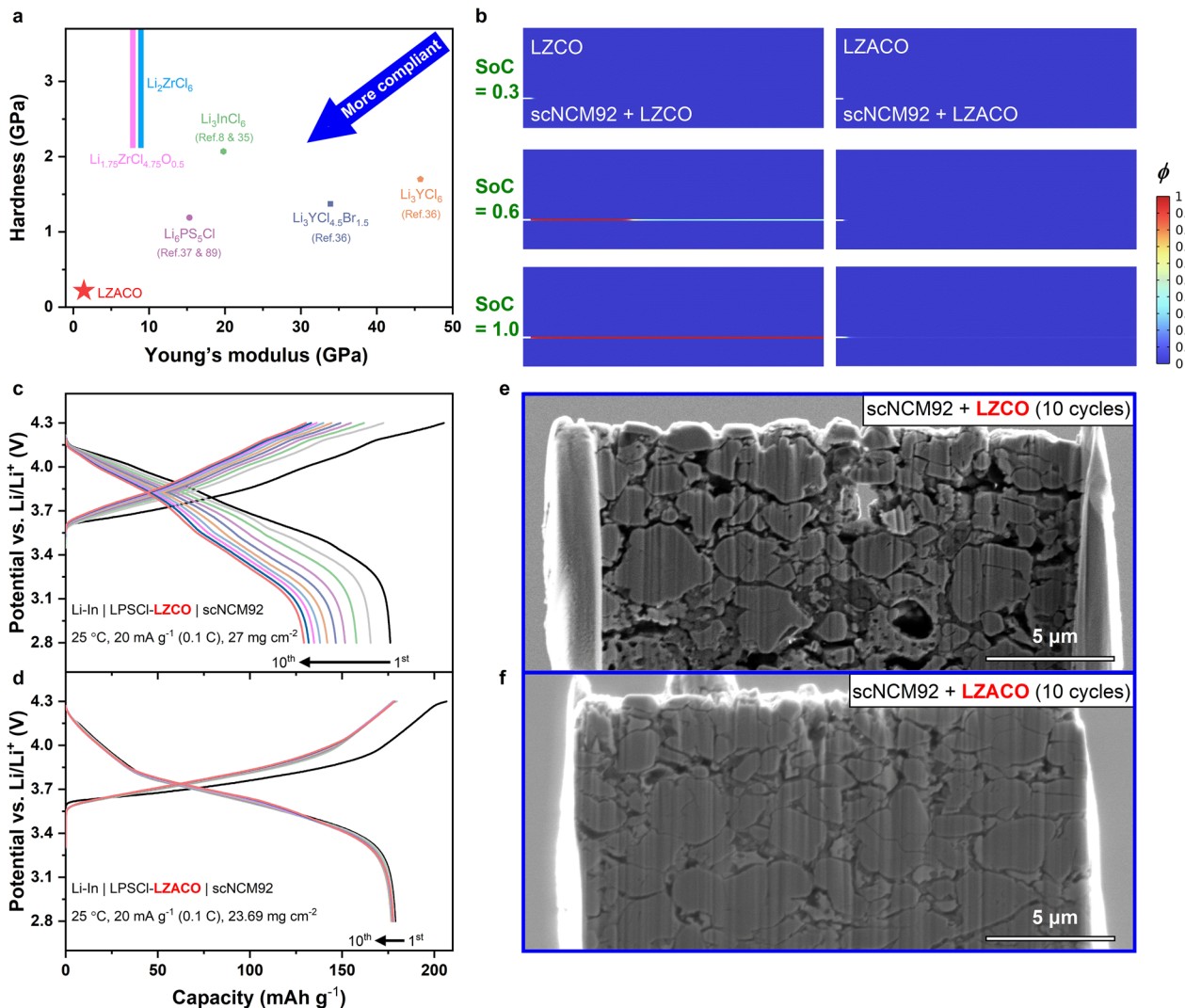

**Fig. 2 | Mechanical compliance of LZACO. a** Hardness vs. Young's modulus plot for LZACO and other solid electrolytes. The hardnesses and Young's moduli of LZACO, $Li_{1.75}ZrCl_{4.75}O_{0.5}$, and $Li_2ZrCl_6$ are obtained from our own measurement, while the rest of the data are from literature[8,35–37,89,90]. **b** Phase-field modelling of the crack development between the composite positive electrode layer and the solid-electrolyte layer when scNCM92 serves as the PEAM and either LZCO or LZACO serves as the solid electrolyte material. The pre-existing notch on the left side is an interface defect used to initiate crack development. $\phi$ is the order parameter, where $\phi = 0$ and 1 represent intactness and detachment, respectively. The SoCs of 0 and 1 correspond to the states where scNCM92 undergoes 0 and −10% strains, respectively, during delithiation. **c, d** Charge and discharge profiles of the all-solid-state Li-In | LPSCl-LZCO | scNCM92 (**c**) and Li-In | LPSCl-LZACO | scNCM92 (**d**) cells at 25 °C under the stacking pressure of 190 MPa. The potential vs. Li/Li$^+$ is calculated by adding the cell voltage and the potential of the Li-In alloy anode, i.e., 0.62 V vs. Li/Li$^+$. **e, f** SEM images of the composite positive electrode of the Li-In | LPSCl-LZCO | scNCM92 cell (**e**) and that of the Li-In | LPSCl-LZACO | scNCM92 cell (**f**) after 10 cycles under the conditions shown in (**c**) and (**d**), respectively.

hardnesses of other solid electrolytes are low enough to enable proper measurements on cold-pressed pellets. Therefore, $Li_{1.75}ZrCl_{4.75}O_{0.5}$ and $Li_2ZrCl_6$ must be no softer than the hardest one among the other solid electrolytes in Fig. 2a, i.e., the $Li_3InCl_6$ with 2.07 GPa hardness. With this taken into account, the mechanical compliance of all the aforementioned solid electrolytes can be compared comprehensively in the hardness vs. Young's modulus plot in Fig. 2a. The data suggest that the hardness and Young's modulus of LZACO are both considerably lower than those of other solid electrolytes (hardness: 0.22 vs. ≥1.19 GPa; Young's modulus: 1.41 vs. 7.92−45.75 GPa). To ensure the reliability of these mechanical characterization results, we conducted scanning electron microscopy (SEM) observation on the surface of the cold-pressed LZACO pellet fabricated under 300 MPa (the same as that used for the characterizations above in Fig. 2a). It is found that such a pellet is almost fully densified (Supplementary Fig. 10). In particular, most of the black spots that look like pores are in fact also solid substances too, as shown in the image at higher magnification

(Supplementary Fig. 10b); they could be the minor crystalline precipitates that are compositionally different from the amorphous matrix. Using the Adjust-Threshold plugin of ImageJ[51], the relative density of the pellet is measured to be 96.13%. Such an almost fully densified pellet ensures the flat, nearly pore-less surface needed for reliable nanoindentation measurements; several representative nanoindentation load-displacement curves acquired from this cold-pressed LZACO pellet are also presented in Supplementary Fig. 11. Here, it must be emphasized that, as a powdery solid electrolyte, the mechanical compliance of LZACO cannot possibly surpass the viscoelastic ones, such as the polymer solid electrolytes and the recently reported Li-Al-Cl-O solid electrolytes with unique pliability[39,41,42]. Instead, the advantage of LZACO in mechanical compliance exists only among the powdery solid electrolytes, which occupy the vast majority of the inorganic solid electrolytes studied so far and appear more compatible with the dry-film technology for pouch-cell fabrication[52,53]. Such improved mechanical compliance suggests that LZACO must be

highly competent in maintaining the intimate solid-solid contact during cycling.

This capability of sustaining the interfacial contact is substantiated both computationally and experimentally. For the computational study, we conducted phase-field modelling to compare LZACO and a compositionally similar solid electrolyte $Li_{1.75}ZrCl_{4.75}O_{0.5}$ (LZCO), based upon the hardnesses and Young's moduli displayed in Fig. 2a. Considering that the hardness of LZCO cannot be precisely determined as mentioned above, our computation adopted its lowest possible value, i.e., 2.07 GPa (as indicated in Fig. 2a), to ensure that the estimated mechanical compliance of LZCO is no poorer than reality. Even so, LZCO is still found less effective in maintaining the solid-solid contact than LZACO. As shown in Fig. 2b, when the state of charge (SoC) of the single-crystalline $LiNi_{0.92}Co_{0.06}Mn_{0.02}O_2$ (scNCM92) reaches 0.6, the composite positive electrode layer formed by 75 wt% scNCM92 and 25 wt% LZCO already partially detaches from the LZCO layer it originally contacted with (the region with the order parameter $\phi = 1$ in Fig. 2b is the detached region). At the SoC of 1.0, their detachment is complete. In contrast, the LZACO layer has not begun to detach from its composite positive electrode layer (formed by 75 wt% scNCM92 and 25 wt% LZACO) at the SoC of 1.0, thereby demonstrating an improved capability of sustaining the solid-solid contact during cycling. To verify the advantage indicated by these computational results, we compared the cycling performance of the actual all-solid-state cells and the interfacial intactness of these cells after cycling. Both experimental observations align rather well with the computational results. The all-solid-state cells assembled for this purpose use scNCM92 as the positive electrode, Li-In alloy as the negative electrode, $Li_6PS_5Cl$ (LPSCl) as the solid electrolyte contacting the negative electrode, and LZCO or LZACO as the solid electrolyte contacting the positive electrode. To make the difference caused by the mechanical compliance of solid electrolytes more easily observable, we tried to maximize the volume change of the composite positive electrode layer by employing a relatively high PEAM mass loading of around 25 mg cm$^{-2}$ and a relatively low cycling rate of 0.1 C (20 mA g$^{-1}$) for both the Li-In | LPSCl-LZCO | scNCM92 and the Li-In | LPSCl-LZACO | scNCM92 cells mentioned above. As shown in Fig. 2c, d, the cycling performances of these two cells do differ greatly from each other. In literature, LZCO has been reported to show satisfactory cycling performance at 25 °C, but only under a very high rate of 1000 mA g$^{-1}$ (leading to low discharge capacities of 70−100 mAh g$^{-1}$) and a relatively low PEAM mass loading of 5−6 mg cm$^{-2}$ [3]. In contrast, the Li-In | LPSCl-LZCO | scNCM92 cell studied here (Fig. 2c) was cycled at 25 °C under a much lower rate of 20 mA g$^{-1}$, which leads to a much higher initial discharge capacity of 176.1 mAh g$^{-1}$ and thereby a much larger volume change in the PEAM particles. Furthermore, the PEAM mass loading of this cell is more than 4 times that in the aforementioned report (27 vs. 5−6 mg cm$^{-2}$), which will also result in greater overall volume change. Under this circumstance, the LZCO with limited mechanical compliance can no longer enable decent cycling stability; the discharge capacity drops from 176.1 to 129.3 mAh g$^{-1}$ only after 10 cycles at 25 °C. In contrast, the LZACO with much lower hardness (0.22 vs. > 2.07 GPa) and much lower Young's modulus (1.41 vs. 7.92 GPa) than LZCO delivered considerably better performance; as shown in Fig. 2d, the discharge capacity of the 1st cycle of the Li-In | LPSCl-LZACO | scNCM92 cell is almost identical with that of the 10th cycle (178.9 and 177.7 mAh g$^{-1}$, respectively) at 25 °C. After such cycling, we prepared cross-sectional specimens from the composite positive electrodes using focused ion beam (FIB) and examined them using SEM. It should be noted that FIB may cut only about 10 μm deep into the surface of the pellet, which is much smaller than the thicknesses of the composite positive electrode layers with the high PEAM mass loading of around 25 mg cm$^{-2}$ here (at least tens of micrometers). Therefore, the height of the lamella prepared this way can by no means reflect the thickness of the corresponding composite positive electrode layer. Consistently

with the cycling data presented in Fig. 2c, d, the FIB-SEM observation does suggest that LZCO fail to maintain intimate contact with the PEAM particles only after 10 cycles (Fig. 2e); in contrast, LZACO delivered much better performance in this regard (Fig. 2f). Here, it needs to be emphasized that the large voids observed in Fig. 2e must mostly emerge after cycling, instead of pre-existing in as-assembled cells, because otherwise the Li-In | LPSCl-LZCO | scNCM92 cell in Fig. 2c cannot possibly deliver an initial discharge capacity similar to that of the Li-In | LPSCl-LZACO | scNCM92 cell in Fig. 2d (176.1 and 178.9 mAh g$^{-1}$, respectively). In fact, the volume of voids in the composite positive electrode of the as-assembled Li-In | LPSCl-LZCO | scNCM92 cell is indeed limited; it is comparable to the volume of voids in the Li-In | LPSCl-LZACO | scNCM92 cell after 10 cycles (Supplementary Fig. 12), where no considerable interfacial contact loss is supposed to occur, as indicated by the discharge capacity nearly identical to the initial value (177.7 and 178.9 mAh g$^{-1}$, respectively). The improved mechanical compliance of LZACO should be the key to maintaining the observed intimate interfacial contact during cycling. With the low hardness and Young's modulus shown in Fig. 2a, LZACO will deform more easily and significantly at given stresses. Consequently, even if the PEAM particles contacting LZACO undergo significant volume change repeatedly, the LZACO particles can still change their morphologies to keep the intimate solid-solid contact, thereby ensuring the high discharge capacities and decent long-term cycling stability. In practical situations, the ASSLBs tend to employ highest possible PEAM mass loading to maximize the energy density. Besides, as mentioned in the Introduction, the discharge rates would also be relatively low in practice, such as 0.1 C-0.3 C, which will make the discharge capacities with respect to the mass of the PEAM higher than those at high rates. As a result, the overall volume change of both the PEAM particles and the composite positive electrode layer would become more severe, making the highly compliant solid electrolyte like LZACO particularly important for ensuring decent cycling performance. In summary, the computational and experimental data presented in Fig. 2 above consistently suggest that the LZACO solid electrolyte reported here is not plagued by the relatively poor mechanical compliance in typical Zr-based (oxy)chloride solid electrolytes such as the $Li_2ZrCl_6$ and LZCO for comparison above. On the contrary, the combined hardness and Young's modulus of LZACO are comparable with, even surpassing, those of some widely studied sulfide and halide solid electrolytes, as shown in Fig. 2a. Therefore, although many derivatives and chemical modifications of Zr-based (oxy)chloride solid electrolytes have already been reported in literature [45,54–56], the LZACO material with the unique advantage in mechanical compliance can still play a distinctive role in ASSLBs.

## Cost-effectiveness

In many recent studies, the $50 kg$^{-1}$ solid-electrolyte cost is referred to as the threshold for the successful commercialization of ASSLBs [3,38,57], but this standard is in fact questionable. First and foremost, the solid-electrolyte cost below $50 kg$^{-1}$ cannot actually ensure that the total production cost of ASSLBs lies below those of commercial Li-ion batteries. Recently, Pasta et al. have conducted an in-depth cost study based on a comprehensive market analysis [58]. Using the actual cell parameters for the batteries in Volkswagen ID. 3, they compared the cost of the energy-dense ASSLBs and the commercial liquid-state Li-ion batteries. Such calculations suggest that the ASSLBs with the aforementioned $50 kg$^{-1}$ solid-electrolyte cost is still more expensive than the commercial Li-ion batteries ($158 kWh$^{-1}$ vs. $126 kWh$^{-1}$). In particular, even if the solid electrolyte is assumed free, the ASSLBs still cost $134 kWh$^{-1}$, exceeding the $126 kWh$^{-1}$ cost for commercial Li-ion batteries. Under these circumstances, the commercial viability of the ASSLBs depends on whether their energy density and safety advantages can justify such high costs in the market, but this cannot be predicted definitively for now. That is, the $50 kg$^{-1}$ solid-electrolyte

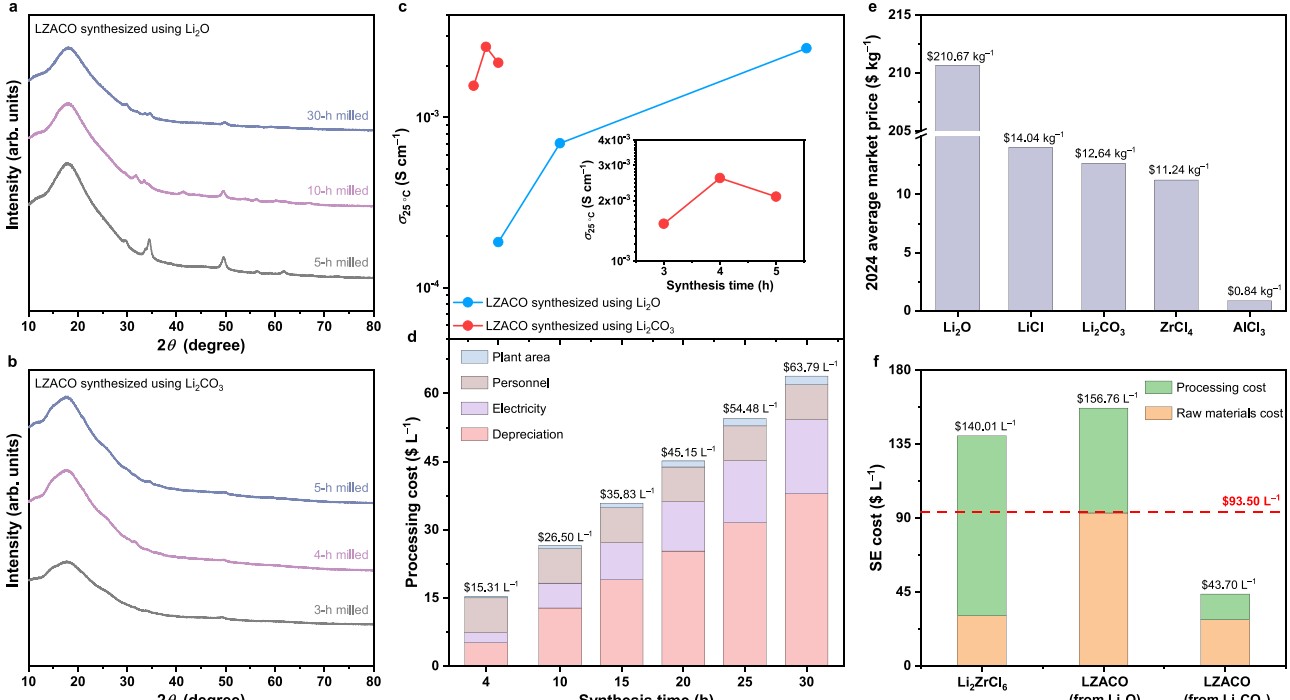

**Fig. 3 | Cost-effectiveness of LZACO. a, b** XRD patterns of the LZACO synthesized using $Li_2O$ (**a**) and that using $Li_2CO_3$ (**b**), respectively, with different planetary mill durations. **c** Ionic conductivities at 25 °C ($\sigma_{25\,°C}$) of the $Li_2O$- and $Li_2CO_3$-synthesized LZACO materials with different lengths of synthesis time. Inset: enlarged view of the variation of $\sigma_{25\,°C}$ for the $Li_2CO_3$-synthesized LZACO. **d** Processing costs of LZACO with different lengths of synthesis time (basis for estimation provided in Methods and Supplementary Tables 1–5). **e** Average market prices in 2024 for the industrial commodity chemicals involved in the cost estimation. The sources from which we obtained these prices are summarized in Supplementary Table 6. **f** Solid-electrolyte (SE) costs of $Li_2ZrCl_6$, $Li_2O$-synthesized LZACO, and $Li_2CO_3$-synthesized LZACO.

cost should by no means be regarded as a threshold guaranteeing an ASSLB cost that is lower than those of commercial Li-ion batteries; instead, it is merely a cost that may yield relatively reasonable ASSLB cost, with potential commercial viability. Additionally, compared with the cost per unit mass like the $50 kg$^{-1}$ discussed above, the cost per unit volume should be more appropriate for discussing the cost-effectiveness of solid electrolytes. In all-solid-state batteries, the optimal amount of the solid electrolyte is not actually defined by the mass, but by the volume: the composite positive electrode with the optimal tortuosity needs the solid electrolyte to occupy a certain volume fraction, while the balance between energy density and safety is realized by adjusting the thickness of the solid-electrolyte layer. Consequently, no matter what kind of solid electrolyte is employed, the optimal solid-electrolyte volume in each kWh of batteries would be the same. On this basis, as long as the cost per unit volume of the solid electrolyte is determined, the solid-electrolyte cost per kWh of batteries, i.e., the ultimate metric for the cost-effectiveness of solid electrolytes, is determined too. Therefore, compared with the widely discussed cost per unit mass of the solid electrolyte, the cost per unit volume of the solid electrolyte is more suitable for discussion. In the aforementioned study by Pasta et al., the solid electrolyte is assumed to be $Li_7P_3S_{11}$, so, using the density of 1.870 g cm$^{-3}$[59] for this material, it can be easily found that the $50 kg$^{-1}$ gravimetric solid-electrolyte cost that may enable relatively reasonable cell production cost corresponds to a volumetric solid-electrolyte cost of $93.50 L$^{-1}$.

With this standard clarified, the cost-effectiveness of LZACO may be discussed. The total cost (also referred to as "production cost" in some studies[60]) of a given solid electrolyte may be obtained by summing its raw materials cost and processing cost. For the LZACO material, both costs can be greatly decreased by replacing $Li_2O$ with $Li_2CO_3$ in the raw materials for synthesis. On the one hand, $Li_2CO_3$ is known to be much cheaper than the moisture-sensitive $Li_2O$. On the other hand, such raw materials replacement is supposed to suppress

the processing cost too, as it can effectively reduce the synthesis time: compared with the purchased $Li_2O$, the $Li_2O$ in situ formed through $Li_2CO_3$ decomposition during intense ball milling would contain a lot more active reaction centers at the surface, thereby enabling faster reaction[61]. Similar phenomena have also been observed in many other compounds[61–63]. In particular, given that LZACO is rather O-rich, with the O/Cl molar ratio (0.37) greatly exceeding those of many Li-Zr-Cl-O solid electrolytes such as $Li_{1.75}ZrCl_{4.75}O_{0.5}$ (O/Cl = 0.11)[3], $Li_{2.22}Zr_{1.11}Cl_{5.33}O_{0.67}$ (O/Cl = 0.13)[44], and $Li_3Zr_{0.75}OCl_4$ (O/Cl = 0.25)[26], the replacement of $Li_2O$ would introduce a relatively large amount of $Li_2CO_3$ in the raw materials, making the aforementioned phenomenon especially effective in reducing the synthesis time. This desirable scenario is verified experimentally. With $Li_2O$ replaced by $Li_2CO_3$ in the raw materials for synthesizing LZACO, the $Li_2CO_3$ completely decomposes after only 4-h milling, as indicated by the absence of C signals in the electron energy-loss spectroscopy (EELS) result (Supplementary Fig. 13). More importantly, after such raw material replacement, the ball milling time needed to achieve similar degree of amorphization (Fig. 3a,b) and ionic conductivity (Fig. 3c) reduces drastically from 30 h to merely 4 h; besides, the $Li_2CO_3$-synthesized LZACO is even slightly more Li-ion conductive than the $Li_2O$-synthesized one (2.59 vs. 2.55 mS cm$^{-1}$ at 25 °C), despite the much shorter synthesis time of the former. It should be noted that, among the four components of the processing cost, i.e., instrument depreciation, electricity, plant area, and personnel, most are rather sensitive to the synthesis time, as shown in Fig. 3d (detailed estimation procedure in Methods and Supplementary Tables 1–5). Specifically, the 30-h synthesis time that is rather common among oxychloride solid electrolytes[3,47] would in fact give rise to a high processing cost of $63.79 L$^{-1}$ for LZACO, even after considering its stability in the air with around 4% relative humidity (Supplementary Figs. 14–15). Nevertheless, when the synthesis time is reduced to the 4 h achieved above, the processing cost drops to only $15.31 L$^{-1}$ (Fig. 3d). In the meantime, since $Li_2CO_3$ is also much more affordable than $Li_2O$

($12.64 kg$^{-1}$ vs. $210.67 kg$^{-1}$, as shown in Fig. 3e and Supplementary Table 6), the raw materials cost also decreases significantly from $92.97 L$^{-1}$ to $28.39 L$^{-1}$. Taking both costs into account, the replacement of $Li_2O$ with $Li_2CO_3$ decreases the total production cost of LZACO from $156.76 L$^{-1}$ to $43.70 L$^{-1}$ (Fig. 3f and Supplementary Table 5), which is much lower than the aforementioned $93.50 L$^{-1}$ solid-electrolyte cost for enabling reasonable ASSLB cost. Remarkably, this $43.70 L$^{-1}$ cost for LZACO is also much lower than that of the prototypic Zr-based (oxy)chloride solid electrolyte, $Li_2ZrCl_6$; the latter has been commonly regarded as a cost-effective solid electrolyte for long due to its competitive raw materials cost ($30.71 L$^{-1}$), but, after incorporating the processing cost, its long synthesis time of 45 h[47] was found to result in a rather high processing cost of $109.30 L$^{-1}$ (Fig. 3f and Supplementary Tables 1–5), which alone greatly exceeds the $93.50 L$^{-1}$ reasonable cost for solid electrolytes (Fig. 3f). Clearly, in order to identify cost-effective, commercially viable solid electrolytes, focusing on the raw materials cost alone is not enough; developing solid electrolytes that may be rapidly synthesized to achieve decent processing cost, such as the LZACO reported here, is equally important.

Although the data above are indicative of decent cost-effectiveness for LZACO, it must also be emphasized that this material alone cannot address the cost issue for the commercialization of ASSLBs. Similar to other Zr-based oxychloride solid electrolytes[64], LZACO is not supposed to be compatible with the reductive negative electrodes such as Li metal. Therefore, it may only serve as the catholyte, which still needs to be separated from the negative electrode by the expensive solid electrolytes such as the sulfides[3]. Besides, the thin Li metal foil needed to enable high energy densities for ASSLBs is also too expensive to make the cost of ASSLBs equal to that of commercial liquid-state batteries[58]. To endow ASSLBs with commercial viability, the cost of these components must be lowered too. The identification of cost-effective catholytes such as LZACO only meets one necessary but insufficient condition for addressing the cost issue of ASSLBs.

## Electrochemical performance

While the observations above have confirmed multiple intriguing properties for LZACO, its comprehensive electrochemical performance may only be verified in all-solid-state cells. Prior to the assembly and testing of cells for this purpose, the electrochemical stability of LZACO was evaluated first. Supplementary Fig. 16 shows the linear sweep voltammetry (LSV) results of the Li | LPSCl-LZACO | LZACO + stainless steel (SS) cell. The data suggest that the reduction and oxidation of LZACO occur at 1.62 and 4.19 V vs. Li/Li$^+$, respectively. The oxidation potential of 4.19 V vs. Li/Li$^+$ is comparable with not only those of sulfide solid electrolytes (no higher than 3 V vs. Li/Li$^+$ [5,65]), but also those of typical Zr-based (oxy)chloride solid electrolytes such as $Li_2ZrCl_6$ and $Li_{1.75}ZrCl_{4.75}O_{0.5}$ (3.55 and 4.00 V vs. Li/Li$^+$, respectively[3]), so LZACO should supposedly display decent compatibility with the high-potential positive electrodes. Besides, it has been reported that the (oxy)chloride solid electrolytes can generally form stable, Li-ion conductive interphases upon oxidation, which allows for stable cycling to potentials beyond their oxidation limits[3,23,66–68]. Consequently, the measured oxidation potential of 4.19 V vs. Li/Li$^+$ entails that LZACO may direct contact high-potential positive electrodes such as $LiCoO_2$ and Ni-rich layered oxides in the all-solid-state cell, without the need of any coating layer. On the other hand, the 1.62 V vs. Li/Li$^+$ reduction potential of LZACO is lower than those of typical Zr-based (oxy)chlorides (2.16 and 1.79 V vs. Li/Li$^+$ for $Li_2ZrCl_6$ and $Li_{1.75}ZrCl_{4.75}O_{0.5}$, respectively[3]), entailing better reduction stability. Regardless, the 1.62 V vs. Li/Li$^+$ reduction potential remains much higher than those of most energy-dense negative electrodes. Furthermore, since the final reduction products of LZACO would inevitably contain electronic conductors such as Zr and Al, which cannot prevent the electrons needed for further reduction to migrate through, the reduction of this material is supposed to proceed continuously[69,70]. Such speculation is

supported by the Li | LZACO | Li and Li-In | LZACO | Li-In symmetric cells, both of which undergo a continuous voltage increase during cycling (Supplementary Fig. 17); in particular, the voltage of the Li | LZACO | Li cell reaches around 2 V after only 80 h of cycling at 0.1 mA cm$^{-2}$ (1 h per cycle). Therefore, to ensure the stable cycling of the all-solid-state cell, LZACO needs to be separated from the negative electrode by a layer of more negative-electrode-compatible solid electrolyte such as LPSCl.

Based on the electrochemical stability determined above, our study of cell performance focuses on two types of 4 V-class positive electrodes, $LiCoO_2$ (LCO) and scNCM92, with the corresponding cell configurations being Li-In | LPSCl-LZACO | LCO and Li-In | LPSCl-LZACO | scNCM92, respectively. The study begins with the ordinary PEAM mass loading of 5–6 mg cm$^{-2}$. As shown in Fig. 4a, the Li-In | LPSCl-LZACO | LCO cell with a PEAM mass loading of 5.81 mg cm$^{-2}$ delivers an initial discharge capacity of 152.6 mAh g$^{-1}$ and a Coulombic efficiency of 97.26% under 31.6 mA g$^{-1}$ (0.2 C) at 25 °C; such a discharge capacity is competitive with that of most cells with similar configuration and cycling conditions in literature (usually below 140 mAh g$^{-1}$; a few examples summarized in Supplementary Table 7)[21,23,50,71–73]. Besides, thanks to the high ionic conductivity of 2.55 mS cm$^{-1}$ for LZACO, the cell exhibits decent rate capability too; as shown in Fig. 4b, c, the average discharge capacities at 158, 316, 790, 1106, and 1580 mA g$^{-1}$ (corresponding to 1 C, 2 C, 5 C, 7 C, and 10 C, respectively) are 139.3, 132.2, 113.6, 94.3, and 69.6 mAh g$^{-1}$, respectively. Beyond $LiCoO_2$, LZACO also enables decent performance for scNCM92. With a PEAM mass loading of 5.26 mg cm$^{-2}$, the Li-In | LPSCl-LZACO | scNCM92 cell delivers an initial discharge capacity of 180.8 mAh g$^{-1}$ and a Coulombic efficiency of 90.27% under 20 mA g$^{-1}$ (0.1 C) at 25 °C (Fig. 4d), which is also competitive with that of many previously reported all-solid-state cells with the similar configuration and cycling conditions (a few examples shown in Supplementary Table 8). The average discharge capacities at 200, 400, 1000, 1400, and 2000 mA g$^{-1}$ (corresponding to 1 C, 2 C, 5 C, 7 C, and 10 C, respectively) are 156.3, 141.3, 111.3, 96.5, and 75.5 mAh g$^{-1}$, respectively (Fig. 4e, f).

More importantly, the mechanically compliant LZACO material can also enable decent long-term cycling stability at high rates. As mentioned above, oxychlorides such as LZACO usually allow for cycling beyond their oxidation potentials, because these materials can form stable and Li-ion conductive interphases upon oxidation[3,23,66–68]. Under high cycling rates, such a procedure would result in a non-negligible capacity fluctuation during the initial cycles, making it difficult to evaluate the capacity retention. Consequently, these initial cycles (displayed in Supplementary Fig. 18) are treated as formation cycles, and the cycles afterwards are presented as the long-term cycling data in Fig. 4g, h. During such tests, both cells demonstrate satisfactory performance. Under 1580 mA g$^{-1}$ (10 C) at 25 °C, the Li-In | LPSCl-LZACO | LCO cell with 5.44 mg cm$^{-2}$ PEAM mass loading shows a capacity retention of 95%, 90%, and 82.06% after 746, 1577, and 2059 cycles, respectively (Fig. 4g). Under 2000 mA g$^{-1}$ (10 C) at 25 °C, the Li-In | LPSCl-LZACO | scNCM92 cell with 5.62 mg cm$^{-2}$ PEAM mass loading shows a capacity retention of 95%, 90%, and 80% after 1018, 2776, and 4208 cycles, respectively (Fig. 4h). As shown in Supplementary Tables 9–10[3,12,16–18,21–25,43,45,49,50,57,71,74,75], among the cost-effective solid electrolytes, LZACO is the only one that can enable stable cycling of the ASSLBs at a rate as high as 10 C, and is also the one that demonstrates the largest number of cycles with the capacity retention above 80% (2059 and 4208 cycles for the Li-In | LPSCl-LZACO | LCO and Li-In | LPSCl-LZACO | scNCM92 cells, respectively). Even when compared with the state-of-the-art but unacceptably expensive solid electrolytes, LZACO still appears rather competitive (Supplementary Tables 9–10). In particular, it needs to be emphasized that the reversible specific capacities of the LZACO-based cells at 10 C is even higher than those of certain cells at 5 C or even 1 C in Supplementary Tables 9–10. Given that the ionic conductivity of 2.55 mS cm$^{-1}$ at 25 °C for LZACO is not

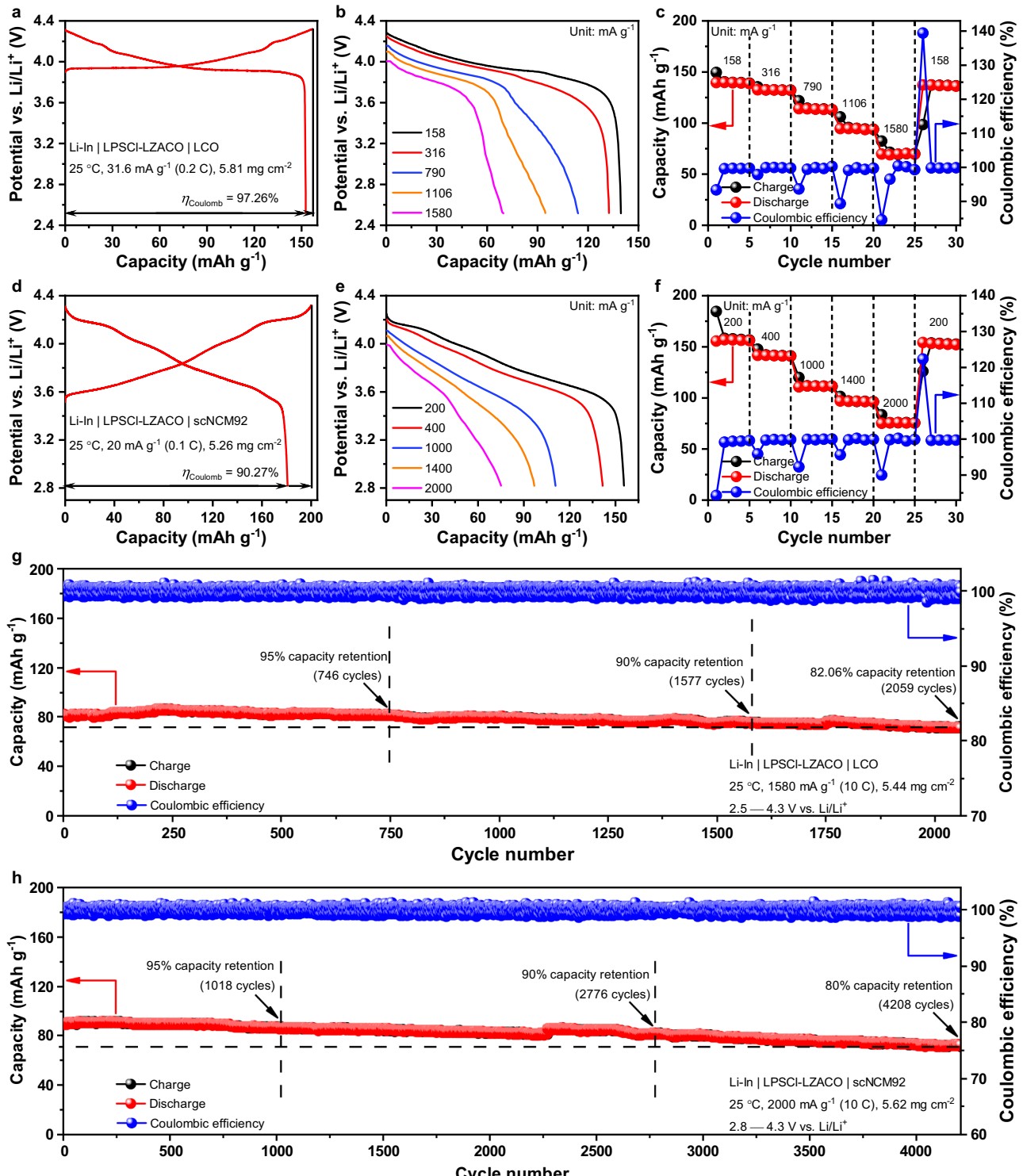

**Fig. 4 | Cycling performance of the LZACO-based ASSLBs at the mass loading of 5–6 mg cm⁻². a–c** Initial charge/discharge profiles (**a**) and the rate capability (**b** and **c**) of the Li-In | LPSCl-LZACO | LCO cell at 25 °C. $\eta_{Coulomb}$ represents the Coulombic efficiency. **d–f** Initial charge/discharge profiles (**d**) and the rate capability (**e** and **f**) of the Li-In | LPSCl-LZACO | scNCM92 cell at 25 °C. $\eta_{Coulomb}$ represents the Coulombic efficiency. **g, h** Long-term cycling performance of the Li-In | LPSCl-LZACO | LCO cell (**g**) and the Li-In | LPSCl-LZACO | scNCM92 cell (**h**) at 25 °C. The potential vs. Li/Li⁺ for each cell is calculated by adding the cell voltage and the potential of the Li-In alloy anode, i.e., 0.62 V vs. Li/Li⁺. All the cells above were cycled under the stacking pressure of 190 MPa.

significantly higher than those of other solid electrolytes in Supplementary Tables 9–10, the improved performance of the cells it forms must arise mostly from the high mechanical compliance. With this advantage, LZACO can also enable satisfactory cell performance to the positive-electrode potentials much higher than the commonly used

4.3 V vs. Li/Li⁺ for testing. As shown in Supplementary Fig. 19, when being cycled between 2.8 and 4.6 V vs. Li/Li⁺, the Li-In | LPSCl-LZACO | scNCM92 cell with 4.84 mg cm⁻² PEAM mass loading displays an initial discharge capacity of 209.58 mAh g⁻¹ and a Coulombic efficiency of 88.16% under 24 mA g⁻¹ (0.1 C) at 25 °C. After 100 cycles under

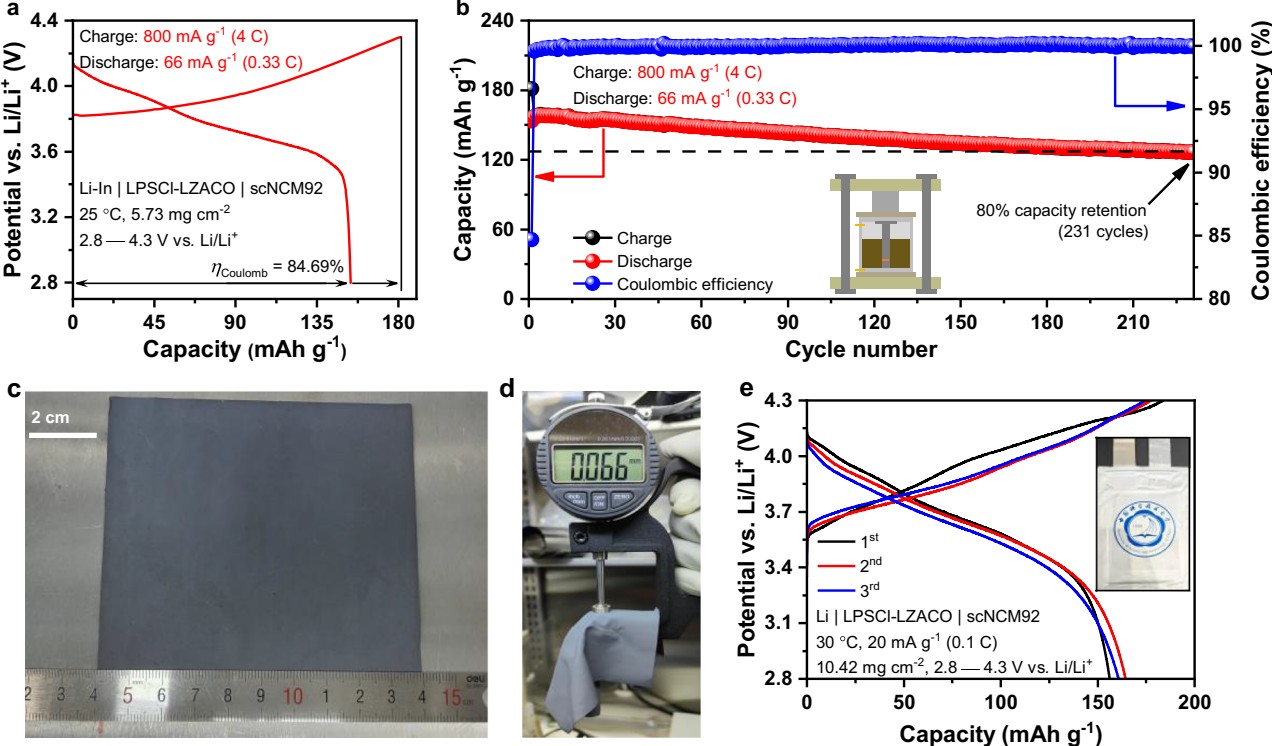

**Fig. 5 | Cycling tests under more practical conditions. a, b** Initial charge/discharge profiles (**a**) and the long-term cycling performance (**b**) of the Li-In | LPSCl-LZACO | scNCM92 cell with the specific currents of 800 mA g$^{-1}$ (4 C) for charge and 66 mA g$^{-1}$ (0.33 C) for discharge at 25 °C. The potential vs. Li/Li$^+$ is calculated by adding the cell voltage and the potential of the Li-In alloy anode, i.e., 0.62 V vs. Li/Li$^+$. As schematically illustrated in the inset of (**b**), this cell was tested in the mold.

The stacking pressure during cycling is 190 MPa. $\eta_{Coulomb}$ represents the Coulombic efficiency. **c, d** Photographs of the composite-positive-electrode sheet for the Li | LPSCl-LZACO | scNCM92 pouch cell. **e** Charge and discharge profiles of the Li | LPSCl-LZACO | scNCM92 pouch cell at 30 °C. Since the anode of this cell is Li metal, the potential vs. Li/Li$^+$ equals to the cell voltage. The stacking pressure during cycling is 10 MPa.

240 mA g$^{-1}$ (1 C) at 25 °C, it can still deliver a discharge capacity of 160.3 mAh g$^{-1}$ with the capacity retention of 89.65%.

In addition to the tests above, the performance of the LZACO-based all-solid-state cells is also evaluated under more practical cycling conditions. It is widely known that the battery-powered instruments usually desire fast charge and relatively slow discharge; for example, the present development of electric vehicles aims at the capability of 3 C–4 C charge, while the discharge rate is usually 0.1–0.3 C, as mentioned in Introduction. Nevertheless, the ASSLBs in literature are rarely cycled under such conditions, but usually operated with the charge and discharge rates being identical. To evaluate the performance of LZACO in this regard, we cycled the Li-In | LPSCl-LZACO | scNCM92 cell with the PEAM mass loading of 5.73 mg cm$^{-2}$ using the specific currents of 800 mA g$^{-1}$ (4 C) for charge and 66 mA g$^{-1}$ (0.33 C) for discharge. The initial cycle under such conditions achieves a discharge capacity of 153.6 mAh g$^{-1}$ (Fig. 5a), which is not much lower than the cell with both the charge and discharge rates set at 1 C (156.3 mAh g$^{-1}$, as shown in Fig. 4e, f). On this basis, the cell also exhibits satisfactory cycling stability; the discharge capacity and capacity retention remain above 126.8 mAh g$^{-1}$ and 80%, respectively, for 231 cycles (Fig. 5b). Beyond the mold cells studied above, we also assembled pouch cells, using the cost-effective dry-film technology. It should be noted that the mechanical compliance for the solid electrolyte has also been found crucial to the fabrication of high-quality films for pouch cells. The compliant solid electrolytes such as the sulfides and halides may easily form and maintain the intimate solid-solid contact with the assistance of the polytetrafluoroethylene (PTFE) binder; instead, if the solid electrolyte is barely compliant like the brittle oxide solid electrolytes, the films cannot enable meaningful cell performance without the addition of minor liquid electrolyte, due to the poor solid-solid

contact[76]. Thanks to the high mechanical compliance of LZACO, the preparation of films for pouch cells are rather convenient. Both the composite positive electrode films and the individual solid-electrolyte films can be easily prepared in relatively large sizes, as shown in Fig. 5c, d and Supplementary Fig. 20. Without the addition of any liquid electrolyte, the Li | LPSCl-LZACO | scNCM92 pouch cell formed by such films demonstrates a reversible capacity of about 160.8 mAh g$^{-1}$ under 20 mA g$^{-1}$ (0.1 C) at 30 °C (Fig. 5e).

The data above have demonstrated that LZACO can enable competitive cell performance at 5–10 mg cm$^{-2}$ PEAM mass loading (equivalent to about 1 mAh cm$^{-2}$ areal capacity) that is frequently used for testing in literature, but the construction of energy-dense ASSLBs requires higher PEAM loadings above 20 mg cm$^{-2}$ and larger areal capacities around 4 mAh cm$^{-2}$. Under such conditions, the volume change would be considerably larger and thus may only be accommodated by highly compliant solid electrolytes. Fortunately, LZACO is rather competent in this regard. When the PEAM mass loadings of the Li-In | LPSCl-LZACO | LCO cell reaches 28.17 mg cm$^{-2}$, an initial discharge capacity of 149.8 mAh g$^{-1}$ (equivalent to an areal discharge capacity of 4.22 mAh cm$^{-2}$) with a Coulombic efficiency of 97.35% may still be achieved under 15.8 mA g$^{-1}$ (0.1 C) at 25 °C (Fig. 6a). Likewise, the Li-In | LPSCl-LZACO | scNCM92 cell at a similar PEAM mass loading of 25.75 mg cm$^{-2}$ also delivers a competitive initial discharge capacity of 168.9 mAh g$^{-1}$ (equivalent to an areal capacity of 4.35 mAh cm$^{-2}$) with a Coulombic efficiency of 85.06% under 20 mA g$^{-1}$ (0.1 C) at 25 °C (Fig. 6b). On this basis, the cells with such high PEAM mass loadings demonstrate decent cycling stability. After 100 cycles under the conditions described above, the Li-In | LPSCl-LZACO | LCO and the Li-In | LPSCl-LZACO | scNCM92 cells deliver areal discharge capacities of 3.62 mAh cm$^{-2}$ (85.78% capacity retention) and 3.92 mAh cm$^{-2}$ (90.11% capacity

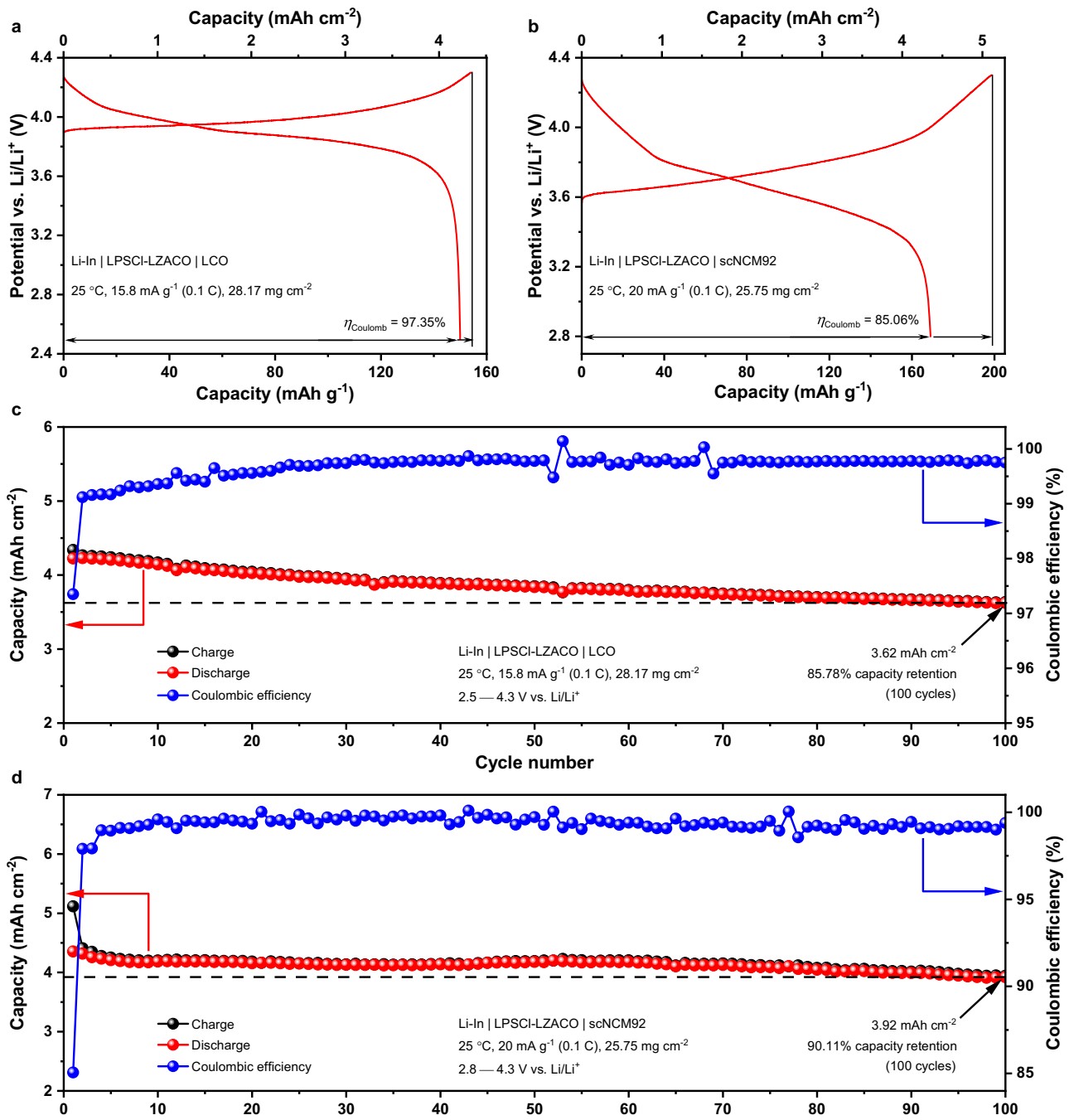

**Fig. 6 | Cycling performance of the LZACO-based ASSLBs at the mass loading above 20 mg cm⁻². a, b** Initial charge/discharge profiles of the Li-In | LPSCl-LZACO | LCO cell (**a**) and the Li-In | LPSCl-LZACO | scNCM92 cell (**b**) with the PEAM mass loading of 28.17 and 25.75 mg cm⁻², respectively, at 25 °C. The potential vs. Li/Li⁺ is calculated by adding the cell voltage and the potential of the Li-In alloy anode, i.e., 0.62 V vs. Li/Li⁺. $\eta_{Coulomb}$ represents the Coulombic efficiency. **c, d** Long-term cycling performance of the Li-In | LPSCl-LZACO | LCO cell (**c**) and the Li-In | LPSCl-LZACO | scNCM92 cell (**d**), where the PEAM mass loading and the cycling conditions are the same as those used for (**a**) and (**b**), respectively. For both cells, the stacking pressure during cycling is 190 MPa.

retention), respectively. This performance is comparable with the state-of-the-art ASSLBs showing areal discharge capacities around 4 mAh cm⁻² in literature, as shown in Supplementary Tables 11,12[12,23,28,77,78]. Nevertheless, among the solid electrolytes that can enable such cell performance, LZACO is presently the only one with decent cost-effectiveness[38] (Supplementary Tables 11,12).

While the sulfide- and halide-based all-solid-state cells in literature are usually cycled at high stacking pressures, the LZACO solid electrolyte enables decent cell performance at low stacking pressures as well. The high mechanical compliance of this material allows it to

efficiently alter the morphology to accommodate the volume change of the PEAM particles even under limited stresses. If the solid electrolytes employed are all highly compliant like LZACO, the ASSLBs do not have to operate under the impractically high stacking pressures above 100 MPa. In this way, the heavy instrument used to generate such pressures (e.g., the mold used for laboratory testing) is no longer needed, so that the ASSLB may eventually achieve decent energy densities for practical application. To demonstrate the advantage of LZACO for enabling decent low-pressure cell performance, we tried to cycle the Li₁₃Si₄ | LPSCl-LZACO | scNCM92 cell under the stacking

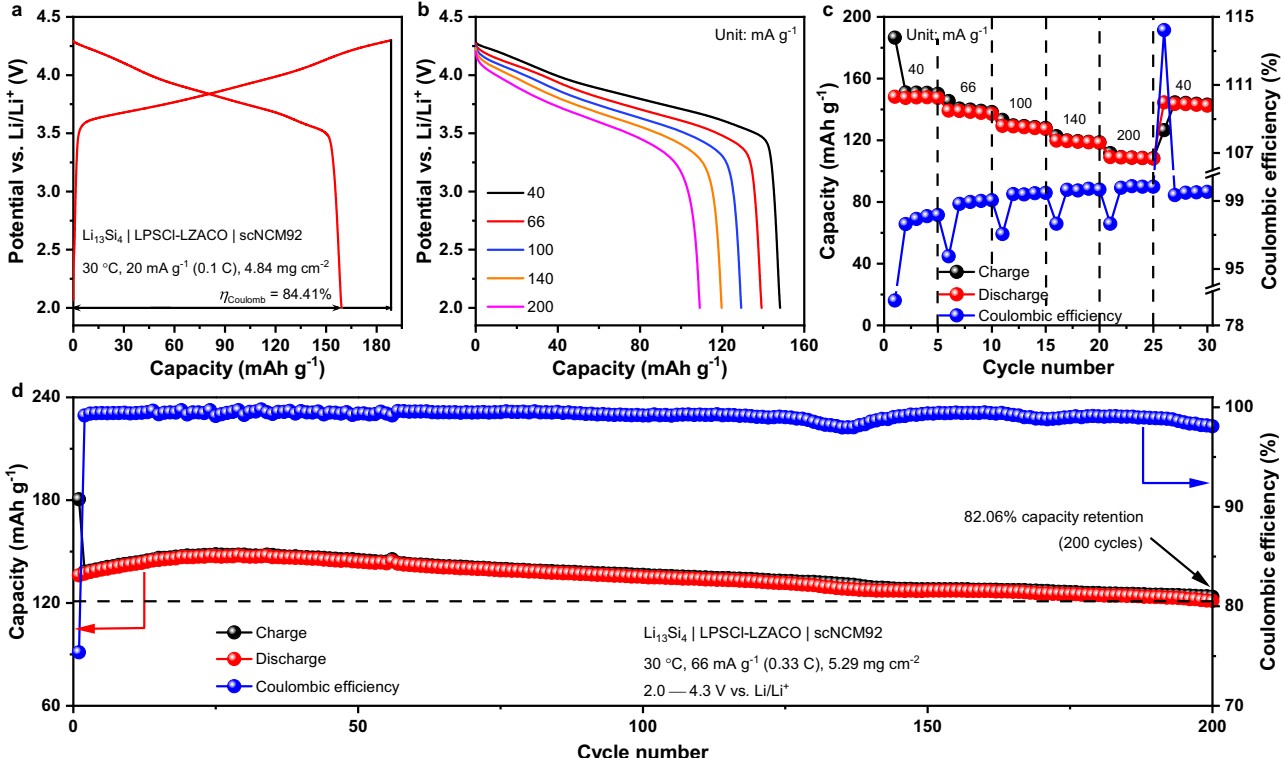

**Fig. 7 | Low-pressure (5 MPa) cycling performance of the LZACO-based ASSLBs.** **a−c** Initial charge/discharge profiles (**a**) and the rate capability (**b** and **c**) of the $Li_{13}Si_4$ | LPSCl-LZACO | scNCM92 cell at 30 °C. The potential vs. Li/Li+ is calculated by adding the cell voltage and the potential of the $Li_{13}Si_4$ anode, i.e., 0.27 V vs. Li/Li+. $\eta_{Coulomb}$ represents the Coulombic efficiency. **d** Long-term cycling performance of the $Li_{13}Si_4$ | LPSCl-LZACO | scNCM92 cell at 30 °C.

pressure of 5 MPa. As shown in Fig. 7a, such a cell delivers a discharge capacity of 159.2 mAh g⁻¹ with a Coulombic efficiency of 84.41% at 30 °C and 20 mA g⁻¹ (0.1 C) in the initial cycle. Besides, it also shows satisfactory rate performance; as shown in Fig. 7b, c, the average discharge capacities at 40, 66, 100, 140, and 200 mA g⁻¹ (corresponding to 0.2 C, 0.33 C, 0.5 C, 0.7 C, and 1 C, respectively) are 147.8, 138.3, 128.4, 119.1, and 108.7 mAh g⁻¹, respectively. When cycled between 2.0 and 4.3 V vs. Li/Li+ at 66 mA g⁻¹ and 30 °C, the $Li_{13}Si_4$ | LPSCl-LZACO | scNCM92 cell achieves a capacity retention of 82.06% after 200 cycles under 5 MPa (Fig. 7d). For comparison, we also tried to replace the LZACO in the cells above with $Li_2ZrCl_6$ (LZC) or LZCO (both are much less compliant than LZACO, as shown in Fig. 2a), and then conducted the cycling tests under the same conditions, especially the low stacking pressure of 5 MPa. The variation tendency of the cell performance was found consistent with that of the mechanical compliance disclosed in Fig. 2a. As shown in Supplementary Fig. 21, the least compliant LZC solid electrolyte results in the poorest cycling performance; the capacity retention after 100 cycles is only 51.09%. In comparison, the relatively more compliant LZCO enables better cycling stability, with the capacity retention being 69.21% after 100 cycles. As for LZACO, its mechanical compliance greatly exceeds those of both LZC and LZCO (Fig. 2a), so this material enables the most stable cycling performance; after 100 cycles, the capacity retention is 91.52%. In addition to the two typical Zr-based (oxy)chloride solid electrolytes for comparison above, the mechanically compliant LZACO can also enable improved cell performance than many other widely studied solid electrolytes. As shown in Supplementary Table 13, the cycling performance achieved by the LZACO-based cells at 5 MPa is comparable with that of certain cells at higher pressures such as 10 MPa[11,35,79]. These observations demonstrate the advantage of the mechanically compliant LZACO in enabling decent cell performance at low stacking pressures, which is a prerequisite for constructing practical ASSLBs. Furthermore, it should

also be noted that the low-pressure cell performance achieved here still has very large room for improvement, because, due to the poor reduction stability of LZACO (reduction potential 1.62 V vs. Li/Li+, as shown in Supplementary Fig. 16), the anolyte of the cell above is still the commonly used LPSCl, which is much less compliant than LZACO (as shown in Fig. 2a) and thus unlikely to perform well at low stacking pressures. If an negative-electrode-compatible solid electrolyte that is as compliant as LZACO can be developed in the future, the low-pressure cell performance similar to those at high stacking pressures should be achievable. To this end, the interphase resulting from the reduction of the solid electrolyte must be electron-blocking, so that the electrons needed for further reduction cannot migrate through to reach the unreacted solid electrolyte[80]. Considering that the $Zr^{4+}$ and $Al^{3+}$ in LZACO would inevitably be reduced into electronic conductors such as Zr and Al by Li metal[81,82], the presence of such metal cations should preferably be eliminated in the material. Instead, the solid electrolyte needs to be populated by non-metal cations such as $P^{5+}$ and $B^{3+}$, which will only be reduced into electron-blocking species[80]. If desirable mechanical compliance and ionic conductivity can be realized in such materials through rational materials design too, the cost-effective ASSLBs that can operate at sufficiently low stacking pressures for practical application may be expected.

In addition to the decent cell performance, the data above also demonstrate one crucial point: the importance of the mechanical compliance for solid electrolytes. In the discussion above, the all-solid-state cells formed by LZACO are compared with those based on other solid electrolytes when different PEAMs, PEAM mass loadings, cycling conditions, etc., are employed (Supplementary Tables 7−13). Among all the solid electrolytes involved in such comparison, LZACO is in fact not the one with the highest ionic conductivity. Regardless, the cell constructed by LZACO can still display improved performance with respect to those based on other more Li-ion conductive solid

electrolytes. The improved mechanical compliance of LZACO (0.22 GPa hardness and 1.41 GPa Young's modulus, as shown in Fig. 2a) plays a pivotal role here. In fact, when the ionic conductivity of the solid electrolyte exceeds 1 mS cm$^{-1}$, improving its mechanical compliance should be more effective in optimizing the cell performance than further increasing the ionic conductivity, as the bottleneck of the overall ion transport has shifted from the interior of the solid electrolyte to the interfacial contact between different cell components. Under such circumstances, if the cell employs highly compliant solid electrolytes such as LZACO, it can more easily achieve and maintain the intimate solid-solid contact, thereby effectively improving the overall ion transport efficiency. From this perspective, the report of the mechanically compliant LZACO has pointed out an effective entry point for further optimizing the comprehensive performance of the solid electrolytes that are already highly Li-ion conductive.

## Discussion

In summary, by designing and synthesizing the solid electrolyte LZACO, we achieve a mechanical compliance improving those of most previously reported inorganic solid electrolytes (hardness: 0.22 vs. > 1 GPa; Young's modulus: 1.41 vs. > 15 GPa), as well as a high Li-ion conductivity (2.55 mS cm$^{-1}$ at 25 °C) and a strong cost-competitiveness (estimated to be \$43.70 L$^{-1}$, below the \$93.50 L$^{-1}$ solid-electrolyte cost for enabling relatively reasonable cell production cost[38]). The combination of such high mechanical compliance and Li-ion conductivity enables cell performance comparable to those of the cells formed by state-of-the-art but unacceptably expensive solid electrolytes. With the ordinary PEAM mass loading of 5–6 mg cm$^{-2}$, the LZACO-based cells with LCO and scNCM92 deliver initial discharge capacities of 152.6 and 180.8 mAh g$^{-1}$ under 0.2 C and 0.1 C, respectively, at 25 °C; under 10 C at 25 °C, these two cells show capacity retentions of 82.06% after 2059 cycles and 80% after 4208 cycles, respectively. When being cycled under conditions closer to practical situations, such as 4 C charge and 0.33 C discharge at 25 °C, the scNCM92 cell exhibits an initial discharge capacity of 153.6 mAh g$^{-1}$, and sustains the capacity retention above 80% for 231 cycles. More importantly, decent performance was also achieved with the PEAM mass loading above 20 mg cm$^{-2}$. After 100 cycles under 0.1 C at 25 °C, the cell with the LCO mass loading of 28.17 mg cm$^{-2}$ demonstrates an areal discharge capacity of 3.62 mAh cm$^{-2}$ with 85.78% capacity retention, while the cell with the scNCM92 mass loading of 25.75 mg cm$^{-2}$ demonstrates an areal discharge capacity of 3.92 mAh cm$^{-2}$ with 90.11% capacity retention. Considering that the LZACO material enabling such cell performance also shows a reasonable cost as mentioned above[38], it could potentially become a suitable solid electrolyte for the commercial, practical ASSLBs in the future.

## Methods

### Materials synthesis

The $x$Li$_2$O·(1-$y$)ZrCl$_4$-$y$AlCl$_3$ solid electrolytes were synthesized mechanochemically by milling the stoichiometric amount of Li$_2$O (Aladdin, 99.99%) or Li$_2$CO$_3$ (Sigma-Aldrich, 99.997%), ZrCl$_4$ (Acros Organics BVBA, 98%), and AlCl$_3$ (Alfa Aesar, 99%) with zirconia balls (5 mm diameter) at a ball-to-powder mass ratio of 20:1 (the mass of the balls is 40 g in each pot) in 80 ml zirconia pots using a planetary mill (FRITSCH, Pulverisette 7 premium line). The milling was conducted at 500 rpm, for the different durations specified in the main text. The Li$_{1.75}$ZrCl$_{4.75}$O$_{0.5}$ solid electrolyte for comparison was synthesized using the same method and instrument from LiCl (Alfa Aesar, 99.9%), Li$_2$O (Aladdin, 99.99%), and ZrCl$_4$ (Acros Organics BVBA, 98%), with the ball-to-powder mass ratio, mass of the balls in each pot, milling speed, and milling time being 15:1, 30 g, 600 rpm, and 40 h, respectively. For all the mechanochemical syntheses, the milling was stopped for 6 min after every 6 min of milling. During the entire procedure, the materials were handled in an Ar-filled glovebox with H$_2$O and O$_2$ contents both

below 0.01 ppm, while the planetary mill was also conducted with the pots sealed to prevent air exposure. Unless otherwise specified, the Li source for all the $x$Li$_2$O·(1-$y$)ZrCl$_4$-$y$AlCl$_3$ solid electrolytes in the presented study is Li$_2$O. For the $x$Li$_2$O·(1-$y$)ZrCl$_4$-$y$AlCl$_3$ solid electrolytes synthesized using Li$_2$CO$_3$ as the Li source, the absence of C in the product is verified by electron energy-loss spectroscopy (EELS) in scanning transmission electron microscopy (STEM) from at least 30 different regions in the sample. Although each individual EELS measurement only collects information from a rather local region of nanometer size, in such a region EELS could detect the minor amount of C species that is invisible in other macroscopic characterization techniques such as X-ray photoelectron spectroscopy (XPS). Since such highly sensitive characterization did not detect the existence of any C in at least 30 different regions in the sample as mentioned above, it should be safe to conclude that the amount of C is negligibly small.

### Structure and morphology characterizations

X-ray diffraction (XRD) experiments were conducted on a Rigaku Ultima IV powder X-ray diffractometer operating at 40 kV and 40 mA with a scanning step of 0.02° and scanning speed of 2° min$^{-1}$, using Cu Kα1 radiation ($\lambda = 1.5406$ Å). During the measurement, Kapton films were used to prevent the sample from being exposed to air. The neutron total scattering experiment for the determination of the pair distribution function (PDF) was conducted on the Multi-Physics Instrument (MPI) at the China Spallation Neutron Source (CSNS)[83]. For each measurement, approximately 1.5 grams of the solid-electrolyte powder was sealed in a cylindrical vanadium nickel alloy container under a helium gas atmosphere. The acquired total neutron scattering datasets of PDFs with a maximum momentum transfer $Q_{max} = 31.5$ Å$^{-1}$ were obtained through the Mantid software[84] and Fourier transformation. The simulation of the PDFs was conducted using the PDFgui software[85]. The scanning electron microscopy (SEM) observation was conducted using a ZEISS GeminiSEM 500 scanning electron microscope operated at 3 kV. The density of the solid electrolyte was calculated from the density of its cold-pressed pellet and the relative density of this pellet; the latter was determined by averaging the relative densities of at least 30 regions of the cold-pressed pellet (the relative density of each region is determined from the SEM image using the Adjust-Threshold plugin of ImageJ[51]). The focused-ion beam (FIB) sectioning of the composite positive electrodes of the cycled all-solid-state cells were performed at 30 kV and 15 nA using a ThermoFisher Helios G4 Plasma-FIB instrument. For both the SEM and FIB experiments, the samples were transferred by air-tight transfer holders to avoid exposure to air. The high-resolution transmission electron microscopy (HRTEM) observation was conducted on powder samples using a JEOL JEM-F200 microscope operated at 200 kV. To suppress the electron-beam irradiation damage, the observation was conducted using a cryogenic TEM holder (which will also prevent air exposure during specimen transfer) at −170 °C. The electron energy-loss spectroscopy (EELS) measurement was conducted on powder samples in a Thermo Fisher Scientific Titan ETEM G2 TEM operated at 300 kV using a cryogenic TEM holder, which ensures both the absence of air exposure during specimen transfer and an observation temperature of −170 °C (for the suppression of electron-beam irradiation damage). The EELS data were collected in the scanning transmission electron microscopy mode using a Gatan Image Filter Quantum-965 spectrometer, where a 5 mm aperture and an energy dispersion of 0.5 eV per channel were adopted.

### Mechanical characterizations

The mechanical characterizations in the present study were all conducted on the pellets formed by cold pressing the solid-electrolyte powder at 300 MPa. The hardness of the solid electrolyte was measured by conducting nanoindentation within SEM using the Hysitron PI 88 in situ nanoindenter with a Berkovich tip under a high load regime.

The samples were transferred into the SEM by an air-tight transfer holder to avoid air exposure. The hardness was calculated from the unloading curve using the method developed by Oliver and Pharr[86,87]. The Young's modulus of the solid electrolyte was measured by atomic force microscopy (AFM) in peak force quantitative nanomechanical mapping mode and analyzed by Derjaguin-Muller-Toporov model. The AFM tip (RTESPA-300) is made of Sb-doped Si; it shows the rectangular geometry, a radius of 8 nm, and a spring constant of 40 N m$^{-1}$. The AFM instrument was placed in an Ar-filled glovebox with $H_2O$ and $O_2$ contents both below 0.01 ppm.

## Phase-field modelling

The simulation of the interfacial cracking between the solid-electrolyte layer and the composite positive electrode layer is based upon the model described schematically in Supplementary Fig. 22; the PEAM in such simulation is the single crystalline $LiNi_{0.92}Co_{0.06}Mn_{0.02}O_2$ (scNCM92), and the solid-electrolyte material is either $Li_{1.75}ZrCl_{4.75}O_{0.5}$ (LZCO) or $1.4Li_2O \cdot 0.75ZrCl_4 \cdot 0.25AlCl_3$ (LZACO). Considering that the property of the interface differs from that of the bulk, an interface layer was introduced between the solid-electrolyte layer and the composite positive electrode layer, and a pre-existing notch was added to initiate the crack development. In the phase-field simulation, the cracking at the interface was analyzed by coupling an order parameter $\phi$, representing damage which in a continuous manner separates detachment ($\phi = 1$) from intact ($\phi = 0$), with the boundary value problem of continuum mechanics. Since the current rate is only 0.1 C in the experiment, a quasi-static degradation model based on a previous study[88] and controlled by the shrinkage deformation of scNCM92 was used for the analysis. The simulation was conducted in COMSOL, based upon the material parameters indicated in Supplementary Table 14.

## Cost analysis

The production cost of the solid electrolyte is estimated by summing its raw materials cost and processing cost. The raw materials cost per unit volume of the solid electrolyte is calculated by multiplying the cost per unit mass of the solid electrolyte and the density of the solid electrolyte. While the method of density measurement has already been described in the "Structure and morphology characterizations" section above, the cost per unit mass of the solid electrolyte was calculated straightforwardly from the average market prices of the corresponding raw materials as the industrial commodity chemicals in the year 2024. These prices and the information sources are listed in Supplementary Table 6. To estimate the processing cost, we assume the solid-electrolyte production to take place in a walk-in enclosure with $3 \times 20$ m$^2$ area and 2.5 m height (Vigor Gas Purification Technologies Co., Ltd.), within which the relative humidity is maintained at 4% and 28 industrial planetary mills (XQM-12T, Changsha Tianchuang Powder Technology Co., Ltd.) are used to fabricate the solid electrolyte. These industrial planetary mills can produce 28 kg of solid electrolyte per batch. Therefore, dividing the processing cost per batch of production by 28 kg and then multiplying the result by the density of the solid electrolyte would yield the processing cost per unit volume of the solid electrolyte. The processing cost per batch of production that is needed for this calculation consists of four components: instrument depreciation cost, electricity cost, personnel cost, and plant area cost. The depreciation cost of each instrument was calculated via dividing its price by the depreciation period (acquired from the manufacturer) and multiplying the result by the working time of this instrument during each batch of production. The electricity cost of each instrument was calculated from the average industrial electricity price for China in September 2025, the rated power of this instrument, and the time it operates at the rated power. For the planetary mill, since it needs to be operated intermittently (such as 5-min milling followed by 5-min break for cooling), the working time for depreciation cost

estimation is considered twice the effective milling time for electricity cost estimation. The personnel cost was calculated from the time needed for operating each instrument and the average salaries of battery industry in China reported in January, 2025[58]. The plant area cost was estimated from the time needed for each batch of production, the area of the aforementioned walk-in enclosure, and the industrial facility rent in China, which was obtained by averaging the rents of 30 industrial parks located in different cities in September 2025.

## Humidity tolerance test

The humidity tolerance test was conducted by placing 0.3 g loose powder of the solid electrolyte either within an environmental chamber (Shanghai Duohe Instrument, DHTSLH-27-O-AR-SD) or a dry room filled with dry air of 4% relative humidity at 25 °C for 12 h. Afterwards, the XRD and EIS tests were conducted on the exposed powder and the 300 MPa-cold-pressed pellet of the exposed powder, respectively, at 25 °C. The data were then compared with those of the non-exposed solid electrolyte.

## Conductivity measurements

The ionic and electronic conductivities were both measured using pellets prepared by cold pressing the solid-electrolyte powder at 300 MPa for 1 min, and this pressure was maintained during the measurement as well. The ionic conductivity was measured by electrochemical impedance spectroscopy (EIS) with two stainless steel rods as the blocking electrodes, using a Bio-Logic MTZ-35 impedance analyzer with an amplitude of 10 mV and frequency range of 35 MHz to 1 Hz. The applied signal is potentiostatic. Each Nyquist plot collected in this condition consists of 35 − 45 data points. No quasi-stationary potential or current was applied prior to measurement. The electronic conductivity was determined by the direct-current (DC) polarization measurements using CH Instruments CHI630E electrochemical analyzer with the applied voltage of 1 V.

## Linear sweep voltammetry (LSV) measurements

To assemble the cell for LSV measurements, 60 mg of the $1.4Li_2O \cdot 0.75ZrCl_4 \cdot 0.25AlCl_3$ powder was first cold-pressed at 150 MPa for 1 min in the mold with 10 mm diameter, and then 60 mg of $Li_6PS_5Cl$ powder (Shenzhen Kejing Star Technology) was pressed on one side of the $1.4Li_2O \cdot 0.75ZrCl_4 \cdot 0.25AlCl_3$ layer under 150 MPa for 1 min; the final thicknesses of these two solid-electrolyte layers are around 280 and 375 μm, respectively. Afterwards, 20 mg of the mixture of $1.4Li_2O \cdot 0.75ZrCl_4 \cdot 0.25AlCl_3$ and 304 L stainless-steel (SS) powders with the weight ratio of 1:3 was uniformly sprinkled on the other side of the $1.4Li_2O \cdot 0.75ZrCl_4 \cdot 0.25AlCl_3$ layer and pressed at 300 MPa for 3 min. Finally, a piece of fresh Li metal foil (100 μm thick, Tianjin Energy Lithium Co., LTD) was attached to the $Li_6PS_5Cl$ layer. The LSV measurements were conducted using a CH Instruments CHI630E electrochemical analyzer with a scan rate of 0.1 mV s$^{-1}$ under an external pressure of 10 MPa at 25 °C.

## Assembly and tests of all-solid-state cells

The positive electrode active material (PEAM) powders, i.e., single crystalline $LiNi_{0.92}Co_{0.06}Mn_{0.02}O_2$ (scNCM92; Minmetals New Energy Materials Co., LTD) and $LiCoO_2$ (LCO; Guangdong Canrd New Energy Technology Co., LTD.), were dried at 100 °C in a drying oven for 10 h before use. The composite positive electrodes were prepared by firstly mixing the PEAM and the oxychloride solid-electrolyte powders at a weight ratio of 75:25 in an agate mortar for 30 min, and then in a vortex mixer (Haimen Kylin-Bell Lab Instruments, QL-866) for 10 min at 1200 rpm. The preparation of the Li-In alloy negative electrode will be described below, along with the cell assembly procedure. The composite negative electrode based on $Li_{13}Si_4$ was prepared by mixing the $Li_{13}Si_4$ (Onstar New Carbon Materials Changzhou Co., LTD.) and the

$Li_6PS_5Cl$ (Shenzhen Kejing Star Technology Company) powders at a weight ratio of 60:40 in an agate mortar for 20 min. The assembly of the mold cells began with pressing 25 mg of the oxychloride solid-electrolyte powder at 150 MPa for 1 min in a mold with 10 mm diameter (final thickness around 115 μm). Subsequently, the composite positive electrode powder was uniformly spread onto one side of the cold-pressed oxychloride solid-electrolyte layer and pressed under 300 MPa for 3 min. To avoid reaction between the oxychloride solid electrolyte and the negative electrode, 35 mg of $Li_6PS_5Cl$ powder was pressed on the other side of the oxychloride solid-electrolyte layer at 150 MPa for 1 min (final thickness around 220 μm). For the cells with the Li-In alloy negative electrode, a piece of indium foil (100 μm thick, Alfa Aesar) was then placed on top of the $Li_6PS_5Cl$ layer, followed by the attachment of a piece of fresh Li metal foil (50 μm thick, Tianjin Energy Lithium Co., LTD). For the cells with the $Li_{13}Si_4$ negative electrode, 25 mg of the aforementioned composite negative electrode, i.e., the one prepared by mixing $Li_{13}Si_4$ and $Li_6PS_5Cl$ powders, was pressed at the surface of the $Li_6PS_5Cl$ layer under 100 MPa for 1 min. The all-solid-state cell formed this way was sandwiched between two stainless-steel rods as current collectors. Finally, the stacking pressure (specified in the figure captions for each cell discussed) was applied to the cell and maintained during the cycling tests. The assembly of the all-solid-state pouch cells began with preparing the composite-positive-electrode and solid-electrolyte sheets by the dry-film processing method. Specifically, 0.5 wt% polytetrafluoroethylene (PTFE; number-average molecular weight 7700 kg mol$^{-1}$, Guangdong Canrd New Energy Technology Co., LTD.) was mixed with the aforementioned composite-positive-electrode or solid-electrolyte powders to form doughs, and then the doughs were calendared to the desired thickness (~70 and ~80 um for the composite-positive-electrode and solid-electrolyte sheets, respectively). Afterwards, the composite-positive-electrode sheet, $1.4Li_2O \cdot 0.75ZrCl_4 \cdot 0.25AlCl_3$ sheet, $Li_6PS_5Cl$ sheet, and Li metal foil (50 μm thick, Tianjin Energy Lithium Co. LTD) were stacked and packed into a laminate bag. After applying a vacuum to the laminate bag, it was sealed and pressed at 10 MPa (this pressure was maintained during cycling tests too). All the cell fabrication processes were carried out in an Ar-filled glove box with $H_2O$ and $O_2$ contents both below 0.01 ppm. The cycling tests were conducted using the Neware CT-3000N and LAND CT2001A battery testing systems. The temperature during the cycling tests was maintained using an incubator (Tianjin Hongnuo Instrument, SPX-250B, temperature accuracy ± 1 °C). Prior to the tests, all the cells were placed in the incubator for 5 h to reach thermal equilibrium at the target temperatures. To ensure reproducibility, at least ten cells were tested for each electrochemical experiment, and the one with the performance closest to the average among these cells is selected for display in the figures.

## Data availability

The source data generated in this study are provided in the Source Data file. Source data are provided with this paper.

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

## Acknowledgements

C.M. acknowledges the financial support from the Strategic Priority Research Program of the Chinese Academy of Sciences (XDB1040200, XDB0450201), the National Natural Science Foundation of China (22479138), the USTC Research Funds of the Double First-Class Initiative (YD2060002033), the Fundamental Research Funds for the Central Universities (WK2060000060), and the National Synchrotron Radiation Laboratory (KY2060000199). S.S. acknowledges the financial support from the National Natural Science Foundation of China (92472207). J.Z. and J.M. acknowledge the financial support from the Guangdong Provincial Key Laboratory of Extreme Conditions (2023B1212010002). J.Y. and J.H. acknowledge the financial support from the National Natural Science Foundation of China (U20A20336 and 21935009) and the Hebei Natural Science Foundation (B2024203054). This work was partially carried out at the Instruments Center for Physical Science, University of Science and Technology of China.

## Author contributions

C.M. planned and supervised the research. L.H. performed the materials synthesis, conductivity measurements, structural characterization, morphology characterization, humidity stability tests, and electrochemical tests. Y.H. and S.S. conducted the phase-field modelling. D.W. and G.W. conducted the nanoindentation tests. W.L., S.J., J.L., Q.C., and L.C. conducted the AFM tests. J.Y. and J.H. conducted the EELS measurement. X.Z. conducted the FIB-SEM observation. J.Z., H.C., W.Y., and J.M. conducted the neutron total scattering measurements and determined the pair distribution functions. Y.W. performed the cryo-TEM observation. K.Y. conducted the XAS measurements and assisted in the related data analysis. J.W., H.L., F.C., and Y.L. assisted in the electrochemical tests. L.H. drafted the manuscript. C.M. revised the manuscript.

## Competing interests

The authors declare no competing interests.

## Additional information

[1]Hefei National Research Center for Physical Sciences at the Microscale, University of Science and Technology of China, Hefei, Anhui, China. [2]Gu-ning Aevum New Energy Technology Co., Ltd, Hefei, Anhui, China. [3]State Key Laboratory of Materials for Advanced Nuclear Energy, Shanghai University, Shanghai, China. [4]CAS Key Laboratory of Mechanical Behavior and Design of Materials, University of Science and Technology of China, Hefei, Anhui, China. [5]Key Laboratory of Precision and Intelligent Chemistry, University of Science and Technology of China, Hefei, Anhui, China. [6]Clean Nano Energy Center, State Key Laboratory of Metastable Materials Science and Technology, Yanshan University, Qinhuangdao, Hebei, China. [7]ZepTools Technology Co., Ltd, Tongling, Anhui, China. [8]Key Laboratory of Artificial Structures and Quantum Control, Shanghai Jiao Tong University, Shanghai, China. [9]China Spallation Neutron Source, Institute of High Energy Physics, Chinese Academy of Sciences, Beijing, China. [10]Instruments Center for Physical Science, University of Science and Technology of China, Hefei, Anhui, China. [11]Shenzhen Key Laboratory of Advanced Energy Storage, Southern University of Science and Technology, Shenzhen, Guangdong, China. [12]i-Lab, Suzhou Institute of Nano-Tech and Nano-Bionics, Chinese Academy of Sciences, Suzhou, Jiangsu, China. [13]Shanghai Electrochemical Energy Device Research Center (SEED), Shanghai Jiao Tong University, Shanghai, China. [14]Frontiers Science Center for Transformative Molecules, Shanghai Jiao Tong University, Shanghai, China. [15]Future Battery Research Center, Global Institute of Future Technology, Shanghai Jiao Tong University, Shanghai, China. [16]National Synchrotron Radiation Laboratory, Hefei, Anhui, China. [17]These authors contributed equally: Lv Hu, Yaolong He. ✉e-mail: sqshi@shu.edu.cn; mach16@ustc.edu.cn

