## [Transparent Peer Review file · Nature Communications]

Mechanically compliant and cost-effective $1.4\text{Li}_2\text{O}-0.75\text{ZrCl}_4-0.25\text{AlCl}_3$ solid electrolyte for all-solid-state batteries with improved cycling stability

Corresponding Author: Professor Cheng Ma

Version 0:

Reviewer comments:

Reviewer #1

(Remarks to the Author)

This manuscript addresses the importance of plastic deformation in solid electrolytes for all-solid-state batteries. It reports the discovery of a Li-Al-Zr-Cl-O composition that exhibits not only high ionic conductivity with relatively low raw material costs but also an exceptionally low hardness of 0.22 GPa. This hardness is significantly lower than that of other solid electrolytes previously considered compliant materials. While individual properties of the Li-Al-Cl-O and Li-Zr-Cl-O material classes have been reported previously and are not entirely novel, this manuscript provides helpful clarification by demonstrating that these desirable characteristics can be simultaneously achieved by combining these two material classes. Although this finding is valuable, the materials characterization presented in the manuscript is relatively weak, despite the work focusing on the identification of a specific composition within an already known material system. In this regard, I believe the manuscript has the potential to be published in Nature Communications after addressing the following major concerns.

Major Points for Revision

(1) Clarification on Mixed-Anion Materials and Mechanical Properties (Page 6):

The authors state that "materials with two types of anions are usually much more compliant than those with only one." This statement is misleading, as "softness" is not a general characteristic of mixed-anion materials. Mechanical properties are primarily determined by bonding nature within specific crystal structures or local atomic arrangements, not simply by the presence of multiple anions. The authors should clarify the specific mechanisms or structural features by which anion mixing leads to enhanced compliance in this system.

(2) Misleading Analysis of Amorphous and Crystalline Phases:

i) XRD Interpretation (Figure 1d, 1e, and Supplementary Figures):

The small peaks observed in the XRD patterns are attributed to either P3m1 or C2/m space groups. The manuscript presents this as though portions of LZACO crystallize in these specific phases. However, these peaks correspond to characteristic reflections of fcc or hcp anion packing (e.g., LiCl typically exhibits fcc anion packing). The C2/m and P3m1 structures in this material class are part of these broader anion packing frameworks. It is therefore an overstatement to attribute these peaks definitively to C2/m or P3m1 space groups without further evidence.

ii) Precipitation vs. Allotropes in XRD Observations:

The XRD peaks observed are more likely due to the precipitation of Li-(Al, Zr)-Cl phases rather than crystalline forms of the oxychloride. This suggests segregation of stable phases from a metastable amorphous matrix. However, the authors suggest that the oxychloride phase itself consists of crystalline and amorphous regions, which is not convincingly supported by the data. If the authors contend that these crystalline phases are allotropes of the oxychloride, additional supporting evidence should be provided.

iii) PDF Analysis (Supplementary Figure 4):

The pair distribution function (PDF) analysis shows distinct peaks extending to long radial distances, which is characteristic of crystalline rather than amorphous materials. While radial distribution function (RDF) peaks can sometimes result from artifacts in the Fourier transform process, the current interpretation of the PDF data is misleading. The authors should revise

their interpretation or reanalyze the data to avoid misinforming readers.

iv) TEM/FFT Analysis (Supplementary Figure 5):

The interpretation of the TEM and FFT patterns in Supplementary Figure 5 appears inconsistent. If the two spots observed in the FFT pattern (Figure 5b) represent a crystalline feature, they suggest a single crystal orientation. However, the regions claimed to be crystalline in the TEM image in Figure 5d are scattered, which should result in a ring-like FFT pattern, not discrete spots. This inconsistency needs to be addressed, as it undermines the interpretation of the sample's microstructure.

(3) Nanoindentation Data Interpretation (Figure 2a):

Figure 2a presents the Young's modulus and hardness of various materials measured by nanoindentation on cold-pressed pellets using a Berkovich tip. While nanoindentation is a well-established method, accurate measurements require samples with surfaces flat relative to the tip radius and near-full densification ($\approx 100\%$ density). Cold-pressed pellets often do not meet these criteria. Therefore, the manuscript should include surface micrographs and representative load-displacement (or stress-strain) curves for each measurement, rather than reporting only final values. This additional data would substantiate the reliability of the mechanical measurements.

(4) Comparison with Li-Al-Cl-O Reference Data:

Previous studies have already demonstrated that Li-Al-Cl-O materials are exceptionally compliant compared to other chloride-based solid electrolytes. Thus, it is important for the authors to contextualize the mechanical properties of LZACO by comparing them directly with both Li-Zr-Cl-O and Li-Al-Cl-O. While data for Li-Zr-Cl-O is included in Figure 2a, comparable data for Li-Al-Cl-O—arguably a more relevant benchmark—is missing. The authors should provide nanoindentation results for Li-Al-Cl-O in Figure 2a for a more comprehensive comparison.

(5) Battery Performance and SEM Analysis (Figures 2c and 2e):

The cycling performance of the battery incorporating LZCO (Figure 2c) appears significantly poorer than comparable data for chloride or oxychloride electrolytes in the literature. This could mislead readers regarding the material's true potential. Moreover, the SEM image in Figure 2e reveals an excessive volume of voids. If these voids result from the volume change of cathode materials, then the entire cathode layer thickness should expand proportionally, which is not evident. It is more likely that these voids were present prior to battery assembly, suggesting poor electrode fabrication. This fabrication issue could account for the poor electrochemical performance shown in Figure 2c. The authors should clarify this point and, if applicable, improve their cell assembly to better represent the intrinsic properties of the electrolyte.

Summary

The manuscript presents an important advancement in the design of compliant solid electrolytes for all-solid-state batteries. However, substantial revisions are required to strengthen the materials characterization, clarify data interpretation, and ensure the reliability of the reported findings. Addressing the points above will significantly enhance the manuscript's clarity, rigor, and impact, making it suitable for publication in Nature Communications.

Reviewer #2

(Remarks to the Author)

I believe this manuscript will make a significant contribution to the solid state battery field, presenting a low cost solid state electrolyte for dry and wet process of fabrication cells with high nickel content. The authors' approach integrates this research from lab scale to large scale application – pouch cell. To enhance clarity and support the authors' claims, the following points should be considered:

1. Please consider about the title associated with the work, because the work is related to the low cost of materials and solid state battery, examples below. I believe that you have opinions and understanding which title is better for your work.

- Cost-Effective and Mechanically Compliant LZACO: Advancing All-Solid-State Li Batteries,
- Enhancing All-Solid-State Li Battery Performance with LZACO: A New Standard in Electrolytes,
- LZACO Solid Electrolyte: Achieving High Performance and Commercial Viability in Li Batteries.

2. How does the mechanical compliance of LZACO compare to other solid electrolytes in practical applications? What specific tests or methods were used to measure the hardness and Young's modulus of LZACO?

3. The cost-effectiveness of LZACO with a raw materials cost of \$10.61/kg. How was this cost calculated, and what factors contribute to its competitiveness? Are there any potential fluctuations in raw material prices that could affect the cost-effectiveness of LZACO?

4. What is the environmental impact of producing and using LZACO, considering both the raw materials and the manufacturing process?

5. What future research directions or improvements are suggested by the authors to further enhance the performance and applicability of LZACO? Are there any potential barriers to further development and commercialization that need to be addressed?

6. Phase-field modeling to compare LZACO and LZCO. How do the computational results align with the experimental findings, and what insights do they provide about the mechanical compliance of LZACO? What specific experimental setups

and conditions were used to validate the computational results, and how do these setups ensure the reliability of the findings?

7. The cycling performance of cells using LZACO and LZCO. How does the mechanical compliance of LZACO contribute to its superior cycling performance compared to LZCO? What are the implications of the observed differences in cycling performance for the practical application of LZACO in all-solid-state batteries?

8. The impact of volume change in the composite cathode layer on the cycling stability. How does the mechanical compliance of LZACO help in accommodating these volume changes, and what advantages does this offer for long-term stability? How do the experimental conditions (e.g., CAM mass loading, cycling rate) affect the observed performance, and what considerations should be taken into account for practical applications?

9. Li₂O can be replaced by the cheaper Li₂CO₃ in the synthesis of LZACO. How does this substitution impact the overall cost and quality of the solid electrolyte? What methods were used to confirm the complete decomposition of Li₂CO₃ during the synthesis process, and how reliable are these methods?

10. the structural and ionic conductivity similarities between LZACO synthesized using Li₂O and Li₂CO₃. How do the XRD and EELS data support these findings, and what implications do they have for the practical application of LZACO? How does the ionic conductivity of LZACO synthesized using Li₂CO₃ compare to other solid electrolytes, and what factors contribute to its high performance?

11. the potential competitiveness of LZACO in terms of production cost due to its stability in humid air. How does this stability compare to other solid electrolytes, and what advantages does it offer for large-scale production? What specific measures or conditions are required to maintain the stability of LZACO during handling and storage, and how do these measures impact the overall production cost?

12. The reduction and oxidation potentials of LZACO. How do these potentials compare to other solid electrolytes, and what implications do they have for the stability and performance of LZACO in all-solid-state cells?

13. The performance of cells with LiCoO₂ and scNCM92 cathodes. How do the initial discharge capacities and Coulombic efficiencies of these cells compare to those reported in the literature for similar configurations?

14. LZACO is the only solid electrolyte with sufficiently low cost for commercialization that can enable stable cycling at a rate as high as 10 C. How was this cost-effectiveness determined, and what factors contribute to its competitiveness compared to other solid electrolytes?

15. The performance of LZACO-based pouch cells. How do the mechanical properties of LZACO contribute to the fabrication and performance of these pouch cells?

16. The performance of LZACO-based cells at low stacking pressures. How does the mechanical compliance of LZACO enable decent performance under these conditions, and what advantages does this offer for practical applications? What potential improvements could be made to further enhance the low-pressure cell performance, and how might these improvements impact the overall usability of LZACO?

Reviewer #3

(Remarks to the Author)

This paper reports Li₂O-ZrCl₄-AlCl₃ solid electrolytes exhibiting ionic conductivity of ~2.2 mS/cm and its cost-effective synthesis using Li₂CO₃ as a precursor. ASSBs using these solid electrolytes as the catholyte are demonstrated. The finding is technically important, however, the novelty of this work is not enough to satisfy the high standard of Nat. Commun. Rather this work is worth to be reported in other more specialized journals.

As for novelty, many oxy-halide SEs exhibiting mS/cm-class ionic conductivity have been reported (e.g. LiNbOCl₄, LiAlCl_{2.5}O_{0.75}, many Li-Zr-Cl-O compounds, xLi₂O-MCl_y (M = Ta, Hf)) while many of those previous reports already highlighted the role of amorphous matrix for achieving high ionic conductivity. Compare to the previous papers on oxy-halide SEs, this work does not provide new scientific insights in terms of material characterization, structure-property relation, or performances.

The change of XRD patterns of LZACO after annealing should be discussed in detail. After 100 °C annealing, it seems to transform to the trigonal structure than it decomposed with the appearance of LiCl?

It is recommended to examine air- and dry-room stability of LZACO to estimate its availability in a practical cell manufacturing process.

Reviewer #4

(Remarks to the Author)

This study emphasizes the critical role of mechanical compliance in solid electrolytes for all-solid-state batteries, alongside ionic conductivity and cost. In this regard, the authors introduce xLi₂O-(1-y)ZrCl₄-yAlCl₃ (LZACO, x = 1.0 ~ 1.8, y = 0.2 ~ 0.5) as a novel oxychloride electrolyte with remarkably low hardness (0.22 GPa) and Young's modulus (1.41 GPa), targeting

commercialization viability. LZACO is synthesized via mechanochemical milling of Li_2CO_3 , ZrCl_4 , and AlCl_3 , achieving an ionic conductivity of 2.55 mS cm^{-1} at 25°C through amorphous-phase dominance. The authors estimated that the cost of raw material is using a log-log scaling model, with Li_2CO_3 replacing costly Li_2O . Given the result from cost modeling, the authors suggest the low cost of raw material of $\$10.61 \text{ kg}^{-1}$ for LZACO by substituting Zr^{4+} and Al^{3+} for Ta^{5+} from $1.6\text{Li}_2\text{O}\cdot\text{TaCl}_5$ and replacing Li_2O to Li_2CO_3 for synthesis. The synthesized LZACO exhibits superior deformability compared to sulfide/halide electrolytes and the phase-field modeling further supports the importance of low hardness and Young's modulus. The authors demonstrated the effect of improved mechanical compliance of LZACO by showing long term cycling of all-solid-state batteries using LiCoO_2 and NCM92 cathodes achieving $3.62\text{--}3.92 \text{ mAh cm}^{-2}$ areal capacity (85–90% retention after 100 cycles) at 25°C . While the authors demonstrate the excellent electrochemical properties of LZACO in all solid state batteries (ASSBs), I believe this paper ultimately falls short of the requirements for publication in nature communications for the reasons articulated below.

1. Page 6, line 2: The authors propose a chemical price threshold of $\$50 \text{ kg}^{-1}$ for the commercialization of ASSBs. However, this paper provides no supporting evidence for this value, nor do the references cited offer any justification. To substantiate this claim, the authors should present a comprehensive market analysis, including data from at least the past two years, as well as a universal framework for estimating the chemical cost associated with ASSB commercialization.

2. Supplementary table 1 and 2: The authors present laboratory-scale chemical prices for solid electrolyte fabrication in Supplementary Tables 1 and 2. However, these prices are derived solely from two chemical suppliers, Alfa Aesar and Aladdin, and are limited to data from January 2025. Relying on arbitrarily selected suppliers over such a narrow timeframe undermines the credibility of the economic and scientific analysis. A broader dataset encompassing multiple suppliers and a longer time period is necessary to provide a robust evaluation.

3. In addition to the chemical cost of solid electrolytes, the overall price of ASSBs is influenced by various hidden manufacturing expenses, such as synthesis processing fees, chemical maintenance costs, and others. Moreover, material costs can fluctuate significantly during mass production for commercialization. Importantly, LZACO requires co-utilization with sulfide-based solid electrolytes or Li-alloy anodes due to its incompatibility with Li metal anodes, which reduces the energy density of ASSBs and increases manufacturing costs. Consequently, laboratory-scale chemical pricing alone is insufficient to convincingly demonstrate LZACO's advantages for ASSB commercialization.

4. Page 9, line 3-16: The authors attribute LZACO's high ionic conductivity to material amorphization, referencing studies on Zr-based halide solid electrolytes. While "intermediate compositions" have been reported in prior research in references, the correlation between these compositions, amorphous phases in synthesized materials, and their impact on ionic conductivity remains unclear. As ball milling-induced amorphization introduces atomic site disorder within materials, this phenomenon should be thoroughly discussed in relation to its influence on LZACO's ion conductivity.

5. The synthesis of amorphized Li_2ZrCl_6 solid electrolytes via anion substitution (e.g., Cl^- replaced by other anions) has already been extensively explored in previous studies (e.g., J. Cheng et al., *Journal of Energy Storage*, 2024, 89, 111700, D. Xu et al., *J. Mater. Chem. A*, 2024, 12, 27694, F. Hussain et al., *npj Comput Mater*, 2024, 10, 148, and etc.). Similarly, substituting Zr with Ta has been reported (e.g., X. Liu et al., *Energy Storage Materials*, 2024, 72, 103737 and etc.). Furthermore, the economic evaluation of oxyhalide solid electrolytes was addressed in earlier work (L. Hu, *Nat Commun.*, 2023, 14, 3807). While this paper provides additional insights into LZACO's mechanical properties, its lack of novelty regarding the material itself diminishes its overall contribution to the field.

6. The authors highlight the low Young's modulus and hardness of LZACO as critical factors for stable contact between LZACO and cathode active materials during ASSB cycling. However, their experiments employed an impractically high stack pressure of 190 MPa with low cathode active material loading of $\sim 5.5 \text{ mg cm}^{-2}$, which fails to substantiate the mechanical compliance benefits of LZACO under realistic conditions. Furthermore, Figures 6 and 7 show that LZACO cells with high cathode loading of $> 25 \text{ mg cm}^{-2}$ cycled at low stack pressure of 5 MPa do not outperform previously reported oxyhalide solid electrolytes in ASSBs in terms of capacity retention. A systematic comparison of stack pressure effects on ASSB cycling using LZACO versus other solid electrolytes such as LZCO or LZO would provide more compelling evidence for the importance of mechanical compliance.

Version 1:

Reviewer comments:

Reviewer #1

(Remarks to the Author)

All my concerns are addressed properly. I now consider that this manuscript is suited to be published.

Reviewer #2

(Remarks to the Author)

Dear Authors,

Thank you for your comprehensive and thoughtful responses to the reviewer comments. I have carefully reviewed your revisions and replies, and I am satisfied that all of my concerns have been adequately addressed.

The manuscript has been significantly improved in both clarity and content, and I have no further major suggestions. I would like to raise one final, brief point: if feasible, please consider including recent references regarding developments in low-pressure and current collectors, lithium metal integration, and dry/wet chemistry processes involving the LPSC electrolyte.

Overall, I believe the revised version meets the standards for publication, and I support its acceptance.

Sincerely, Reviewer

Reviewer #3

(Remarks to the Author)

The reviewer considers highlighting mechanical properties of LZACO does not confer further novelty for the publication since the incorporation of Al, Cl, and O is quite expectable considering the reported viscoelasticity of Li-Al-Cl-O materials (Nat. Energy, 8, 1221 (2023)).

Furthermore, the measured Young's modulus value of LZACO should be explained carefully. As the authors stated "the mechanical compliance of LZACO cannot possibly surpass the viscoelastic ones, such as the polymer solid electrolytes....", generally, Young's modulus of solid materials will be inorganic > polymeric materials. The measured Young's modulus of LZACO is 1.41, which is even lower than many polymers such as PETs, nylons, and etc. Is there any possibility that the measurement method underestimate Young's modulus? In this context, it is recommended to measure some representative materials including sulfides, halides, and polymers with the same method. If 1.4Li₂O-0.75ZrCl₄-0.25AlCl₃ (LZACO) shows the lower Young's modulus than polymers, the appropriate chemical explanation should be discussed in detail.

Reviewer #4

(Remarks to the Author)

The authors have made clear efforts to address the concerns raised in the first review round through additional experiments and analyses. These revisions have strengthened the manuscript; however, the work may not yet fully meet the exceptionally high bar of novelty generally expected for publication in Nature Communications.

The Li-Zr-Cl-O family of solid electrolytes investigated in this study has already been extensively reported in the literature, and Al-doping strategies have likewise been previously explored (e.g., Energy Storage Materials, Vol. 70, June 2024, 103444; Tan et al., Nano Research 17, 8826-8833, 2024, among others). In addition, the synthesis route employed in the present work—ball milling—while effective for laboratory-scale preparation, is associated with relatively high production costs, hindering scalability and commercial viability. Consequently, it is difficult to regard the proposed material as a cost-competitive candidate for large-scale application in solid-state batteries.

That said, this work offers several notable strengths. The comprehensive crystallinity analysis of the highly amorphous electrolyte, the careful evaluation of its feasibility as a solid electrolyte, and the demonstration of its electrochemical performance in pouch cells using a dry-film electrode configuration represent meaningful contributions that could serve as useful benchmarks for future studies in the field of solid-state batteries.

Therefore, I would recommend submission to a more specialized journal focused on electrochemical energy storage or secondary battery technology, rather than Nature Communications.

Version 2:

Reviewer comments:

Reviewer #3

(Remarks to the Author)

The revised manuscript does not provide new content to justify the claimed novelty, which is in line with the previous Nat. Energy paper.

Reviewer #4

(Remarks to the Author)

All my concerns are addressed properly. I now consider that this manuscript is suited to be published.

Responses to Reviewers' Comments

Unless otherwise specified, all page numbers, figure numbers, and reference numbers mentioned below refer to the numbering sequence in the revised manuscript.

The contents copied from the revised manuscript are highlighted in yellow.

Reviewers' comments:

Reviewer #1 (Remarks to the Author):

This manuscript addresses the importance of plastic deformation in solid electrolytes for all-solid-state batteries. It reports the discovery of a Li-Al-Zr-Cl-O composition that exhibits not only high ionic conductivity with relatively low raw material costs but also an exceptionally low hardness of 0.22 GPa. This hardness is significantly lower than that of other solid electrolytes previously considered compliant materials. While individual properties of the Li-Al-Cl-O and Li-Zr-Cl-O material classes have been reported previously and are not entirely novel, this manuscript provides helpful clarification by demonstrating that these desirable characteristics can be simultaneously achieved by combining these two material classes.

Although this finding is valuable, the materials characterization presented in the manuscript is relatively weak, despite the work focusing on the identification of a specific composition within an already known material system. In this regard, I believe the manuscript has the potential to be published in Nature Communications after addressing the following major concerns.

(1) Clarification on Mixed-Anion Materials and Mechanical Properties (Page 6):

The authors state that "materials with two types of anions are usually much more compliant than those with only one." This statement is misleading, as "softness" is not a general characteristic of mixed-anion materials. Mechanical properties are primarily determined by bonding nature within specific crystal structures or local atomic arrangements, not simply by the presence of multiple anions. The authors should clarify the specific mechanisms or structural features by which anion mixing leads to enhanced compliance in this system.

(Response) We sincerely thank the reviewer for his/her careful assessment of our manuscript and the insightful, constructive suggestions. The direct cause of the mechanical compliance is indeed the atomic configuration and the associated bonding nature, instead of the coexistence of multiple anions itself. The mechanism that the materials with mixed anions are frequently more compliant is now indicated on page 6: "First of all, compared with the materials with only one type of anion, those with two types of anions have often been found to display the kind of atomic configuration that enables higher mechanical compliance^{25,39,40}. In such materials, the coexisting anions frequently make the cation-anion polyhedra more distorted and disordered, so that the backbone of the atomic framework becomes more easily rotated or bended, leading to enhanced mechanical compliance^{25,39,40}; this phenomenon has been observed in multiple solid electrolytes, including the O-modified Na₃PS₄⁴⁰,

Li_2ZrCl_6 ^{3,25}, and LiAlCl_4 ³⁹”.

(2) Misleading Analysis of Amorphous and Crystalline Phases:

i) XRD Interpretation (Figure 1d, 1e, and Supplementary Figures):

The small peaks observed in the XRD patterns are attributed to either $P3m1$ or $C2/m$ space groups. The manuscript presents this as though portions of LZACO crystallize in these specific phases. However, these peaks correspond to characteristic reflections of fcc or hcp anion packing (e.g., LiCl typically exhibits fcc anion packing). The $C2/m$ and $P3m1$ structures in this material class are part of these broader anion packing frameworks. It is therefore an overstatement to attribute these peaks definitively to $C2/m$ or $P3m1$ space groups without further evidence.

(Response) We fully agree with the reviewer that these peaks should not be attributed definitively to the $C2/m$ or the $P\bar{3}m1$ structures, and we thank the reviewer for helping us clarify this important point. In the revised manuscript, the structures originally assigned as the $C2/m$ and the $P\bar{3}m1$ space groups are only described as the structures showing the cubic close-packed (ccp) and the hexagonal close-packed (hcp) anion configurations, respectively. The revision is made on page 10: “Although the low crystallinity of $x\text{Li}_2\text{O}-(1-y)\text{ZrCl}_4-y\text{AlCl}_3$ prevents the precise crystal structure determination, the XRD patterns may still disclose certain information regarding the atomic configuration. Generally speaking, the characteristic reflections of up to two types of crystal structures can be identified; both had been frequently observed in the Li_3MCl_6 -type chloride solid electrolytes⁴⁸. One of them is the structure showing the hexagonal close-packed (hcp) anion arrangement; the materials displaying this kind of structure include the Li_2ZrCl_6 with the $P\bar{3}m1$ space group^{47,49} and the Li_3YCl_6 with the $Pnma$ space group⁴⁸. The other is the structure showing the cubic close-packed (ccp) anion arrangement; the materials displaying this type of structure include the LiCl with the $Fm\bar{3}m$ space group⁴⁸ and the Li_3ScCl_6 with the $C2/m$ space group⁵⁰. Due to the low crystallinity of LZACO, it is difficult to conclusively determine which specific structures mentioned above is displayed by each phase. Consequently, the two crystalline phases present in LZACO would be generally referred to as the hcp and ccp phases, respectively, in the discussion below.”.

ii) Precipitation vs. Allotropes in XRD Observations:

The XRD peaks observed are more likely due to the precipitation of Li-(Al, Zr)-Cl phases rather than crystalline forms of the oxychloride. This suggests segregation of stable phases from a metastable amorphous matrix. However, the authors suggest that the oxychloride phase itself consists of crystalline and amorphous regions, which is not convincingly supported by the data. If the authors contend that these crystalline phases are allotropes of the oxychloride, additional supporting evidence should be provided.

(Response) We thank the reviewer for pointing out this important issue. It is indeed possible that the crystalline phases are precipitates showing different compositions from the amorphous matrix. Precisely determining the composition of the extremely small crystallites in LZACO (about 15 nm) requires elemental mapping at very high magnifications using transmission electron microscopy, but the intense electron beam needed for such characterization would damage the electron-beam sensitive oxychloride solid electrolytes, preventing the conclusive composition determination. In light of this fact, we cease to definitively describe the crystalline phases as the allotropes of LZACO, but emphasize

the possibility that the crystalline and amorphous phases in this material might be compositionally different. The revision is made on page 9: “It must be emphasized that the compositions of such minor crystalline species are not necessarily identical with that of the amorphous matrix. Instead, it might be the Li-M-Cl (M = Zr and/or Al) precipitates, which are much more difficult to amorphize than the oxychlorides^{25,42}. The precise composition determination for the minor crystalline phases here would require the elemental mapping at magnifications no lower than those in the HRTEM image presented above (Supplementary Fig. 5a), but the intense electron beam needed for such characterization would rapidly damage the electron-beam sensitive oxychloride solid electrolytes. Consequently, the composition of the crystalline phase cannot be conclusively determined. Fortunately, their volume fraction (about 10% as mentioned above) is too low to considerably influence the overall ion transport.”.

iii) PDF Analysis (Supplementary Figure 4):

The pair distribution function(PDF) analysis shows distinct peaks extending to long radial distances, which is characteristic of crystalline rather than amorphous materials. While radial distribution function(RDF) peaks can sometimes result from artifacts in the Fourier transform process, the current interpretation of the PDF data is misleading. The authors should revise their interpretation or reanalyze the data to avoid misinforming readers.

(Response) We thank the reviewer for prompting us to consider this important issue. It should be noted that the PDF data presented in Supplementary Fig. 4 only describes the atomic configuration up to r of 30 Å, i.e., 3 nm, while the crystallites scattered in the amorphous matrix could in fact be as large as around 15 nm, according to the HRTEM image in Supplementary Fig. 5d. Consequently, at the $r \leq 3$ nm range for the PDF, the weak crystalline characteristics are supposed to be present, and such a feature does not conflict with the low crystallinity indicated by HRTEM. This point is emphasized on page 9: “Such a crystallite size (around 15 nm) explains why the PDF of the highly amorphous LZACO material (Supplementary Fig. 4) still shows weak peaks characterizing ordered crystalline structures up to $r = 30$ Å (3 nm).”.

iv) TEM/FFT Analysis (Supplementary Figure 5):

The interpretation of the TEM and FFT patterns in Supplementary Figure 5 appears inconsistent. If the two spots observed in the FFT pattern (Figure 5b) represent a crystalline feature, they suggest a single crystal orientation. However, the regions claimed to be crystalline in the TEM image in Figure 5d are scattered, which should result in a ring-like FFT pattern, not discrete spots. This inconsistency needs to be addressed, as it undermines the interpretation of the sample's microstructure.

(Response) We thank the reviewer for raising this important point. Supplementary Fig. 5d in the original submission is the inverse FFT pattern generated from the two spots of the FFT pattern in Supplementary Fig. 5b, so what can possibly appear in Supplementary Fig. 5d are only the crystallites giving rise to these two particular spots in the FFT pattern. However, among the crystalline regions disclosed this way, only the relatively large region at the left side of Supplementary Fig. 5d is truly crystalline, while the other extremely small “crystalline” regions are the artifacts associated with the inverse FFT processing, rather than real crystallites. This can be proved by conducting FFT analysis on different regions of the original HRTEM image: except the aforementioned region at the left side

of the image, none of the other regions can give rise to visible spots in the FFT pattern (examples are presented as Supplementary Fig. 5e and f, which are also pasted below). That is, the only true crystalline region in Supplementary Fig. 5d is the relatively large one at the left side; there is indeed only one crystalline region here, rather than multiple scattered, extremely small ones, as pointed out by the reviewer. This is clarified on page 9: “The size and morphology of the crystalline regions can be probed by conducting inverse FFT using these two weak spots. As shown in Supplementary Fig. 5d, such processing discloses one relatively large crystalline region at the left side of the image and several scattered, extremely small ones. Unlike the former, the latter should be the artifacts resulting from the inverse FFT processing, rather than true crystallites; when FFT was conducted locally at different regions in Supplementary Fig. 5c, the aforementioned relatively large crystalline region is in fact the only one that can give rise to spots in the FFT pattern (Supplementary Fig. 5e), and the same happens to none of the other regions (example shown in Supplementary Fig. 5f).”.

Supplementary Fig. 5. **a** HRTEM image of LZACO. **b** FFT pattern of the area delineated by the red dashed rectangle in (a). The spots arising from the crystalline phase are highlighted by the red circles. **c** HRTEM image of the area delineated by the red dashed rectangle in (a). **d** Inverse FFT pattern using the spots circled in red in

(b). Region I in (c) are the same area as Region I in (d), while the same applies to Region II as well. e, f FFT patterns of Region I (e) and Region II (f) in (c), respectively, with the spots arising from the crystalline phase highlighted by the red circles.

(3) Nanoindentation Data Interpretation (Figure 2a):

Figure 2a presents the Young's modulus and hardness of various materials measured by nanoindentation on cold-pressed pellets using a Berkovich tip. While nanoindentation is a well-established method, accurate measurements require samples with surfaces flat relative to the tip radius and near-full densification ($\approx 100\%$ density). Cold-pressed pellets often do not meet these criteria. Therefore, the manuscript should include surface micrographs and representative load-displacement (or stress-strain) curves for each measurement, rather than reporting only final values. This additional data would substantiate the reliability of the mechanical measurements.

(Response) We sincerely thank the reviewer for this valuable suggestion. As mentioned in the main text, among the solid electrolytes with definitive values for both the hardness and Young's modulus in Fig. 2a, only LZACO was characterized through our own nanoindentation measurement; most data in this figure are in fact the reported values in literature. Thanks to the excellent mechanical compliance of LZACO, its cold-pressed pellet may indeed reach a relative density of 96.13%, rather close to 100%. Therefore, the data should be reliable. In the revised manuscript, the SEM image of the cold-pressed LZACO pellet and the representative load-displacement curves are both presented. These contents are presented on page 15: "To ensure the reliability of these mechanical characterization results, we conducted scanning electron microscopy (SEM) observation on the surface of the cold-pressed LZACO pellet fabricated under 300 MPa (the same as that used for the characterizations above in Fig. 2a). It is found that such a pellet is almost fully densified (Supplementary Fig. 9). In particular, most of the black spots that look like pores are in fact also solid substances too, as shown in the image at higher magnification (Supplementary Fig. 9b); they could be the minor crystalline precipitates that are compositionally different from the amorphous matrix. Using the Adjust-Threshold plugin of ImageJ⁵¹, the relative density of the pellet is measured to be 96.13%. Such an almost fully densified pellet ensures the flat, nearly pore-less surface needed for reliable nanoindentation measurements; several representative nanoindentation load-displacement curves acquired from this cold-pressed LZACO pellet are also presented in Supplementary Fig. 10."

Supplementary Fig. 9. SEM images of the cold-pressed LZACO pellet fabricated under 300 MPa (the same as that for the cold-pressed pellet used in the nanoindentation experiments).

Supplementary Fig. 10. Representative nanoindentation load-displacement curves for the cold-pressed LZACO pellet fabricated under 300 MPa.

(4) Comparison with Li-Al-Cl-O Reference Data:

Previous studies have already demonstrated that Li-Al-Cl-O materials are exceptionally compliant compared to other chloride-based solid electrolytes. Thus, it is important for the authors to contextualize the mechanical properties of LZACO by comparing them directly with both Li-Zr-Cl-O and Li-Al-Cl-O. While data for Li-Zr-Cl-O is included in Figure 2a, comparable data for Li-Al-Cl-O—arguably a more relevant benchmark—is missing. The authors should provide nanoindentation results for Li-Al-Cl-O in Figure 2a for a more comprehensive comparison.

(Response) We thank the reviewer for raising this important point. The Li-Al-Cl-O solid electrolytes can indeed achieve exceptional mechanical compliance; when displaying the optimal mechanical properties, this type of materials even ceases to take the powdery form, but become a viscoelastic substance resembling polymer, whose mechanical compliance will surpass the powdery solid electrolytes like LZACO for sure. However, such viscoelastic solid electrolytes cannot be easily fabricated into pouch-cell films using the cost-effective dry-film technology, as it would easily overflow under the pressure during rolling and could also stick to the rollers. In this regard, the LZACO material that remains powdery but displays a relatively high mechanical compliance appears more desirable.

Although the viscoelastic Li-Al-Cl-O is certainly more compliant than the powdery LZACO as mentioned above, the quantitative comparison requested by the reviewer is challenging: the extremely low hardness of these materials makes it difficult to conduct the nanoindentation measurements. Regardless, we still qualitatively emphasize this fact on page 16: “Here, it must be emphasized that, as a powdery solid electrolyte, the mechanical compliance of LZACO cannot possibly surpass the viscoelastic ones, such as the polymer solid electrolytes and the recently reported Li-Al-Cl-O solid electrolytes with unique pliability^{39,41,42}. Instead, the advantage of LZACO in mechanical compliance exists only among the powdery solid electrolytes, which occupy the vast majority of the inorganic solid electrolytes studied so far and appear more compatible with the dry-film technology for pouch-cell fabrication^{52,53}”.

(5) Battery Performance and SEM Analysis (Figures 2c and 2e):

The cycling performance of the battery incorporating LZCO (Figure 2c) appears significantly poorer than comparable data for chloride or oxychloride electrolytes in the literature. This could mislead

readers regarding the material's true potential.

(Response) We thank the reviewer for these valuable and insightful questions. Here we would like to emphasize that the cell configuration and cycling conditions for the LZCO-based cell in Fig. 2c are much harsher than those usually applied in literature: the cathode active material mass loading reaches 27 mg cm^{-2} , instead of the $5\text{--}6 \text{ mg cm}^{-2}$ typically employed in previous studies, and the cell is also cycled at a relatively low rate of 0.1 C , so that the volume change of both the cathode active material particles and the entire composite cathode layer would be much larger than that in the typical LZCO-based cells in literature. Under this circumstance, only the highly compliant solid electrolyte can enable decent cycling performance. With the limited compliance observed for LZCO (as shown in Fig. 2a), we indeed found this material not capable of enabling satisfactory cycling stability at such conditions.

Moreover, the SEM image in Figure 2e reveals an excessive volume of voids. If these voids result from the volume change of cathode materials, then the entire cathode layer thickness should expand proportionally, which is not evident.

(Response) Regarding the thickness of the entire composite cathode layer, we would like to clarify that the heights of the lamellas in Fig. 2e and f are in fact not the thicknesses of the composite cathode layers. Since the cathode active material mass loading of the composite cathodes studied here is around 25 mg cm^{-2} , the thickness of the composite cathode layer should at least be tens of micrometers; in contrast, the “lift-out” method for focused-ion beam (FIB) sample preparation may only cut a much smaller depth (about $10 \text{ }\mu\text{m}$) into the surface of the pellet. Therefore, no matter how thick the composite cathode layer is, the lamella samples prepared by FIB would always be about $10 \text{ }\mu\text{m}$ high, as shown Fig. 2e and f, and such heights by no means reflect the thicknesses of the composite cathode layers. We apologize for not making this point clear enough in the original submission. It is now emphasized on page 18: “It should be noted that FIB may cut only about $10 \text{ }\mu\text{m}$ deep into the surface of the pellet, which is much smaller than the thicknesses of the composite cathode layers with the high CAM mass loading of around 25 mg cm^{-2} here (at least tens of micrometers). Therefore, the height of the lamella prepared this way can by no means reflect the thickness of the corresponding composite cathode layer.”.

It is more likely that these voids were present prior to battery assembly, suggesting poor electrode fabrication. This fabrication issue could account for the poor electrochemical performance shown in Figure 2c. The authors should clarify this point and, if applicable, improve their cell assembly to better represent the intrinsic properties of the electrolyte.

(Response) We thank the reviewer for pointing out this important issue. However, we believe the voids should still form after cycling, because otherwise the LZCO-based cell in Fig. 2c cannot possibly reach an initial discharge capacity almost identical with that of the LZACO-based cell in Fig. 2d (176.1 and 178.9 mAh g^{-1} , respectively). To demonstrate this point, we conducted another FIB-SEM observation to the composite cathode of the LZCO-based cell before cycling. It was found that the volume of voids is no larger than that of the composite cathode of the LZACO-based cell after 10 cycles, which shows a discharge capacity nearly the same as the initial value (177.7 vs. 178.9 mAh g^{-1}) and thus is not supposed to undergo any non-negligible interfacial contact loss yet. These contents are presented on page 18: “Here, it needs to be emphasized that the large voids observed in Fig. 2e must mostly emerge

after cycling, instead of pre-existing in as-assembled cells, because otherwise the Li-In | LPSCI-LZCO | scNCM92 cell in Fig. 2c cannot possibly deliver an initial discharge capacity similar to that of the Li-In | LPSCI-LZACO | scNCM92 cell in Fig. 2d (176.1 and 178.9 mAh g⁻¹, respectively). In fact, the volume of voids in the composite cathode of the as-assembled Li-In | LPSCI-LZCO | scNCM92 cell is indeed limited; it is comparable to the volume of voids in the Li-In | LPSCI-LZACO | scNCM92 cell after 10 cycles (Supplementary Fig. 11), where no considerable interfacial contact loss is supposed to occur, as indicated by the discharge capacity nearly identical to the initial value (177.7 and 178.9 mAh g⁻¹, respectively).”.

Supplementary Fig. 11. a SEM image of the composite cathode of the Li-In | LPSCI-LZCO | scNCM92 cell prior to cycling. **b** SEM image of the composite cathode of the Li-In | LPSCI-LZACO | scNCM92 cell after 10 cycles under the conditions shown in Fig. 2d.

Summary

The manuscript presents an important advancement in the design of compliant solid electrolytes for all-solid-state batteries. However, substantial revisions are required to strengthen the materials characterization, clarify data interpretation, and ensure the reliability of the reported findings. Addressing the points above will significantly enhance the manuscript's clarity, rigor, and impact, making it suitable for publication in Nature Communications.

(Response) Once again, we sincerely thank the reviewer for his/her positive comments on our work and the insightful, valuable suggestions that greatly helped in strengthening the present study. Hopefully the revision presented here can properly address the reviewer's concerns.

Reviewer #2 (Remarks to the Author):

I believe this manuscript will make a significant contribution to the solid state battery field, presenting a low cost solid state electrolyte for dry and wet process of fabrication cells with high nickel content. The authors' approach integrates this research from lab scale to large scale application – pouch cell. To enhance clarity and support the authors' claims, the following points should be considered:

(1). Please consider about the title associated with the work, because the work is related to the low cost of materials and solid state battery, examples below. I believe that you have opinions and understanding which title is better for your work.

- Cost-Effective and Mechanically Compliant LZACO: Advancing All-Solid-State Li Batteries,*
- Enhancing All-Solid-State Li Battery Performance with LZACO: A New Standard in Electrolytes,*
- LZACO Solid Electrolyte: Achieving High Performance and Commercial Viability in Li Batteries.*

(Response) We deeply appreciate the reviewer's positive feedback and insightful suggestions. To further emphasize the advantage of the material reported here, we changed the title to “**Mechanically compliant and cost-effective $1.4\text{Li}_2\text{O}-0.75\text{ZrCl}_4-0.25\text{AlCl}_3$ solid electrolyte for all-solid-state batteries with superior cycling stability**”.

(2) How does the mechanical compliance of LZACO compare to other solid electrolytes in practical applications?

(Response) We thank the reviewer for raising this important point. The mechanical compliance of LZACO, as reflected by its hardness and Young's modulus, greatly exceeds those of other widely studied solid electrolytes, which has been illustrated in Fig. 2a. In practical applications, this advantage is reflected in the cell performance at low stacking pressures; since the high stacking pressures above 100 MPa for laboratory testing are nearly impossible to realize in practical situations, the all-solid-state cell at the practically viable but much lower stacking pressures would have to rely on the high mechanical compliance of the solid electrolyte to maintain the intimate solid-solid contact and thereby the decent cycling stability. To probe this characteristic for LZACO, we tried to compare the performance of the all-solid-state cells formed by LZACO and those formed by two other prototypic Zr-based (oxy)chloride solid electrolytes, Li_2ZrCl_6 (LZC) and $\text{Li}_{1.75}\text{ZrCl}_{4.75}\text{O}_{0.5}$ (LZCO), under a low stacking pressure of 5 MPa. Additionally, the performance of such LZACO-based cells was also compared with other cells under low stacking pressures in literature. These comparisons all suggest that the mechanically compliant LZACO is advantageous in enabling low-pressure cell performance, so it should favor practical applications. The detailed comparison and discussion are presented on page 36: “For comparison, we also tried to replace the LZACO in the cells above with Li_2ZrCl_6 (LZC) or LZCO (both are much less compliant than LZACO, as shown in Fig. 2a), and then conducted the cycling tests under the same conditions, especially the low stacking pressure of 5 MPa. The variation tendency of the cell performance was found consistent with that of the mechanical compliance disclosed in Fig. 2a. As shown in Supplementary Fig. 19, the least compliant LZC solid electrolyte results in the poorest cycling performance; the capacity retention after 100 cycles is only 51.09%. In comparison, the relatively more compliant LZCO enables better cycling stability, with the capacity retention being 69.21% after 100 cycles. As for LZACO, its mechanical compliance greatly exceeds

those of both LZC and LZCO (Fig. 2a), so this material enables the most stable cycling performance; after 100 cycles, the capacity retention is 91.52%. In addition to the two typical Zr-based (oxy)chloride solid electrolytes for comparison above, the mechanically compliant LZACO can also enable better cell performance than many other widely studied solid electrolytes. As shown in Supplementary Table 8, the cycling performance achieved by the LZACO-based cells at 5 MPa even surpasses that of certain cells at higher pressures such as 10 MPa^{11,35,78}. These observations demonstrate the advantage of the mechanically compliant LZACO in enabling decent cell performance at low stacking pressures, which is a prerequisite for constructing practical ASSLBs⁷⁹.

Supplementary Fig. 19. a–c Long-term cycling performance of the $\text{Li}_{13}\text{Si}_4$ | LPSCI-LZC | scNCM92 cell (a), $\text{Li}_{13}\text{Si}_4$ | LPSCI-LZCO | scNCM92 cell (b), and $\text{Li}_{13}\text{Si}_4$ | LPSCI-LZACO | scNCM92 cell (c) under 5 MPa at 30 °C.

Supplementary Table 8. Low-pressure cycling performance of the LZACO-based ASSLBs and those based on other solid electrolytes.

CAM	Catholyte	Specific current (mA g ⁻¹)	Temperature (°C)	Stacking pressure (MPa)	Capacity retention	Ref.
LiNi _{0.92} Co _{0.06} Mn _{0.02} O ₂	LZACO	66 (0.33 C)	30	5	95% (69 cycles)	This work
					90% (119 cycles)	
					80% (215 cycles)	
LiCoO ₂	Li _{1.625} Al _{0.375} Zr _{0.625} Cl _{5.25}	~70 (0.5 C)	30	9.62	~58.71% (282 cycles)	(78)
LiCoO ₂	Li ₂ ZrCl ₆	~70 (0.5 C)	30	9.62	~48.74% (109 cycles)	(78)
LiNi _{0.83} Co _{0.11} Mn _{0.06} O ₂	Li ₃ InCl ₆	n/a	80	2	65% (50 cycles)	(35)
				10	93% (50 cycles)	(35)
LiNi _{0.5} Co _{0.2} Mn _{0.3} O ₂	Li _{0.388} Ta _{0.238} La _{0.475} Cl ₃	~66 (0.44 C)	30	~2	81.6% (100 cycles)	(11)

What specific tests or methods were used to measure the hardness and Young's modulus of LZACO?

(Response) The hardness and Young's modulus were measured by SEM-based nanoindentation and atomic-force spectroscopy, respectively. The details of measurements are described on page 43: "The hardness of the solid electrolyte was measured by conducting nanoindentation within SEM using the Hysitron PI 88 in situ nanoindenter with a Berkovich tip under a high load regime. The samples were transferred into the SEM by an air-tight transfer holder to avoid air exposure. The hardness was calculated from the unloading curve using the method developed by Oliver and Pharr^{85,86}. The Young's modulus of the solid electrolyte was measured by atomic force microscopy (AFM) in peak force quantitative nanomechanical mapping mode and analyzed by Derjaguin-Muller-Toporov model. The AFM tip (RTESPA-300) is made of Sb-doped Si; it shows the rectangular geometry, a radius of 8nm, and a spring constant of 40 N m⁻¹. The AFM instrument was placed in an Ar-filled glovebox with H₂O and O₂ contents both below 0.01 ppm."

(3). The cost-effectiveness of LZACO with a raw materials cost of \$10.61/kg. How was this cost calculated, and what factors contribute to its competitiveness? Are there any potential fluctuations in raw material prices that could affect the cost-effectiveness of LZACO?

(Response) We thank the reviewer for these thoughtful inquiries. To begin with, we would like to first clarify that our discussion in the revised manuscript is no longer based on the cost per unit mass of the solid electrolyte, but on the cost per unit volume of the solid electrolyte; since the optimal volume of

the solid electrolyte in all-solid-state cells (consisting of the solid-electrolyte volume to realize optimal tortuosity in the composite cathode and that to realize the desirable thickness for the individual solid-electrolyte layer) does not vary with the type of the solid electrolyte, the cost per unit volume of the solid electrolyte can more straightforwardly reflect the contribution of the solid-electrolyte cost to the overall cell production cost. The method for calculating this cost is described on page 44: “The cost per unit volume of the solid electrolyte for discussion here is calculated by multiplying the cost per unit mass of the solid electrolyte and the density of the solid electrolyte. While the method of density measurement has already been described in the “Structure and morphology characterizations” section above, the cost per unit mass of the solid electrolyte was calculated straightforwardly from the average market prices of the corresponding raw materials as the industrial commodity chemicals in the year of 2024. These prices and the information sources are listed in Supplementary Table 1.”

Supplementary Table 1. Average market prices of the industrial commodity chemicals in 2024 and the sources from which these prices are acquired.

Commodity chemical	Average market price in 2024 (\$ kg ⁻¹)	Sources
Li ₂ O	210.67	LB Group Co., Ltd.
LiCl	14.04	Shandong Hongyang Chemical Co., Ltd.
Li ₂ CO ₃	12.64	Chengxin Lithium Group Co., Ltd.
ZrCl ₄	11.24	CBC Metal (https://www.cbcie.com/)
AlCl ₃	0.84	Asian Metal (https://www.asianmetal.cn/) Shanghai Metals Market (https://www.smm.cn/)

As for the factors contributing to the cost-competitiveness of LZACO and the influence of potential price fluctuations, these two issues may be discussed together. The major reason that LZACO shows desirable cost-effectiveness lies in the fact that it can be synthesized by using highly affordable chemicals only, i.e., Li₂CO₃, ZrCl₄, and AlCl₃. If the prices of these chemicals fluctuate significantly, the cost of the solid electrolyte will inevitably be influenced, as noted by the reviewer. Fortunately, these chemical prices are relatively stable in the past two years. Regardless, we still tried to avoid the potential distraction from chemical price fluctuation by utilizing the average market price in 2024 for estimation in the revised manuscript. The results of such cost analysis are presented on page 23: “These data suggest that high-quality LZACO can be synthesized from the cost-effective Li₂CO₃, ZrCl₄, and AlCl₃, whose average market prices in 2024 as the industrial commodity chemicals are \$12.64 kg⁻¹, \$11.24 kg⁻¹, and \$0.84 kg⁻¹, respectively (Fig. 3d and Supplementary Table 1). Based on these chemical prices and the density of LZACO (measured to be 2.15 g cm⁻³; detailed procedure described in Methods), the raw materials cost of this solid electrolyte can be estimated as \$22.77 L⁻¹ (Fig. 3e).

In comparison, if the much more expensive Li_2O ($\$210.67 \text{ kg}^{-1}$), instead of the Li_2CO_3 with $\$12.64 \text{ kg}^{-1}$ cost, is used to synthesize LZACO, the raw materials cost would be $\$92.94 \text{ L}^{-1}$, which is over four times that of the Li_2CO_3 -synthesized LZACO ($\$22.77 \text{ L}^{-1}$, as mentioned above) and also greatly exceeds the $\$46.75 \text{ L}^{-1}$ solid-electrolyte raw materials cost for enabling reasonable ASSLB cost. Therefore, the fact that LZACO may be synthesized directly from the affordable Li_2CO_3 is crucial to the cost-effectiveness of this solid electrolyte. In addition, since the synthesis of LZACO involves the rather cost-effective AlCl_3 ($\$0.84 \text{ kg}^{-1}$) but does not rely on the relatively expensive LiCl ($\$14.04 \text{ kg}^{-1}$), its raw materials cost is also lower than those of other previously reported Zr-based (oxy)chloride solid electrolytes (Fig. 3e).”.

Fig. 3 | Cost-effectiveness of LZACO. **a** Electron energy-loss spectra of the LZACO synthesized using Li_2CO_3 , ZrCl_4 , and AlCl_3 as the raw materials. For comparison, the C-K spectrum of CaCO_3 is also displayed. **b** XRD pattern of the LZACO synthesized using Li_2CO_3 , ZrCl_4 , and AlCl_3 as the raw materials. **c** Nyquist plot of the LZACO synthesized using Li_2CO_3 , ZrCl_4 , and AlCl_3 as the raw materials. The measurement was conducted at 25°C . **d** Average market prices in 2024 for the industrial commodity chemicals involved in the cost estimation. **e** Raw materials costs of LZACO and two other representative Zr-based (oxy)chloride solid electrolytes, Li_2ZrCl_6 and $\text{Li}_{1.75}\text{ZrCl}_{4.75}\text{O}_{0.5}$.

(4). What is the environmental impact of producing and using LZACO, considering both the raw materials and the manufacturing process?

(Response) We thank the reviewer for raising this important point. The environmental impact is discussed on page 25: “For a 100-m^3 dry room, the power needed to maintain the 1% relative humidity is about 1.5 times that for the 3.85% relative humidity, and the difference will become larger for larger dry rooms. Consequently, the superior humidity tolerance of LZACO (stable up to around 4% relative

humidity at 25 °C as mentioned above) can greatly save the power needed for handling and storage, which makes its large-scale production cost-effective and also environmentally friendly. Besides, even if LZACO or any of its raw materials is accidentally exposed to the atmosphere with overly high humidity, it will not release the toxic and flammable H₂S gas like the sulfide solid electrolytes⁶¹, but is only supposed to generate the much less hazardous HCl like other chloride solid electrolytes^{61,62}. The cost-effectiveness and eco-friendliness of LZACO make it suitable for large-scale commercial utilization.”.

(5) What future research directions or improvements are suggested by the authors to further enhance the performance and applicability of LZACO? Are there any potential barriers to further development and commercialization that need to be addressed?

(Response) We thank the reviewer for these thoughtful and insightful inquiries. The future directions for improvement and the potential barriers are discussed on page 37: “Furthermore, it should also be noted that the low-pressure cell performance achieved here still has very large room for improvement, because, due to the poor reduction stability of LZACO (reduction potential 1.62 V vs. Li/Li⁺, as shown in Supplementary Fig. 14), the anolyte of the cell above is still the commonly used LPSCI, which is much less compliant than LZACO (as shown in Fig. 2a) and thus unlikely to perform well at low stacking pressures. If an anode-compatible solid electrolyte that is as compliant as LZACO can be developed in the future, the low-pressure cell performance similar to those at high stacking pressures should be achievable. To this end, the interphase resulting from the reduction of the solid electrolyte must be electron-blocking, so that the electrons needed for further reduction cannot migrate through to reach the unreacted solid electrolyte⁷⁹. Considering that the Zr⁴⁺ and Al³⁺ in LZACO would inevitably be reduced into electronic conductors such as Zr and Al by Li metal^{80,81}, the presence of such metal cations should preferably be eliminated in the material. Instead, the solid electrolyte needs to be populated by non-metal cations such as P⁵⁺ and B³⁺, which will only be reduced into electron-blocking species⁷⁹. If desirable mechanical compliance and ionic conductivity can be realized in such materials through rational materials design too, the cost-effective ASSLBs that can operate at sufficiently low stacking pressures for practical application may be expected.”

(6) Phase-field modeling to compare LZACO and LZCO. How do the computational results align with the experimental findings, and what insights do they provide about the mechanical compliance of LZACO?

(Response) We appreciate the reviewer's attention to these important issues. As mentioned on pages 17–19, the computational results align well with the experimental findings. While the phase-field modeling suggests that LZCO would be more easily detached from the cathode than LZACO, this characteristic is indeed verified by the direct SEM observation of the composite cathode of the cycled cells (Fig. 2e and f); besides, the LZCO-based cell also shows much poorer cycling stability than the LZACO-based one with the same cell configuration (Fig. 2c and d), consistent with the aforementioned SEM observation. These corroborating computational and experimental results suggest that LZACO displays much better mechanical compliance than LZCO.

What specific experimental setups and conditions were used to validate the computational results, and

how do these setups ensure the reliability of the findings?

(Response) The experimental verification of the computational results was conducted by comparing the LiIn | LPSCI-LZCO | scNCM92 and LiIn | LPSCI-LZACO | scNCM92 cells. Both cells employ relatively low cycling rate of 0.1 C and relatively high cathode active material mass loading of around 25 mg cm⁻² to maximize the volume change of the cathode active material particles and the composite cathode layer, respectively, so that only the highly compliant solid electrolytes may accommodate such large volume change during cycling. As mentioned on pages 17–19, the LZCO-based cell under such conditions does display much poorer cycling stability than the cell based on the more compliant LZACO, consistent with the computational prediction that LZCO is less effective in sustaining the solid-solid contact than LZACO. Besides, the direct SEM observation of the composite cathodes in these two cells after cycling also indicates that the LiIn | LPSCI-LZCO | scNCM92 cell undergoes much more severe electrode-electrolyte detachment than the LiIn | LPSCI-LZACO | scNCM92 cell. In a word, the computational results are supported both indirectly (by the cycling stabilities of the actual cells, as shown in Fig. 2c and d) and directly (by the SEM observation of the cycled composite cathodes, as shown in Fig. 2e and f) through experiments. Therefore, the results should be reliable.

(7) The cycling performance of cells using LZACO and LZCO. How does the mechanical compliance of LZACO contribute to its superior cycling performance compared to LZCO? What are the implications of the observed differences in cycling performance for the practical application of LZACO in all-solid-state batteries?

(Response) We are grateful to the reviewer for underscoring the importance of these issues. They are now discussed on page 19: “The superior mechanical compliance of LZACO should be the key to maintaining the observed intimate interfacial contact during cycling. With the extremely low hardness and Young’s modulus shown in Fig. 2a, LZACO will deform more easily and significantly at given stresses. Consequently, even if the CAM particles contacting LZACO undergo significant volume change repeatedly, the LZACO particles can still change their morphologies to keep the intimate solid-solid contact, thereby ensuring the high discharge capacities and decent long-term cycling stability. In practical situations, the ASSLBs tend to employ highest possible CAM mass loading to maximize the energy density. Besides, as mentioned in Introduction, the discharge rates would also be relatively low in practice, such as 0.1–0.3 C, which will make the discharge capacities with respect to the mass of the CAM higher than those at high rates. As a result, the overall volume change of both the CAM particles and the composite cathode layer would become more severe, making the highly compliant solid electrolyte like LZACO particularly important for ensuring the decent cycling performance.”.

(8) The impact of volume change in the composite cathode layer on the cycling stability. How does the mechanical compliance of LZACO help in accommodating these volume changes, and what advantages does this offer for long-term stability? How do the experimental conditions (e.g., CAM mass loading, cycling rate) affect the observed performance, and what considerations should be taken into account for practical applications?

(Response) We thank the reviewer for asking these insightful questions. However, since the answers to them happen to be covered by the response to the last question too, we choose not to repeat it here

again to save the reviewer's time.

(9) Li₂O can be replaced by the cheaper Li₂CO₃ in the synthesis of LZACO. How does this substitution impact the overall cost and quality of the solid electrolyte? What methods were used to confirm the complete decomposition of Li₂CO₃ during the synthesis process, and how reliable are these methods?

(Response) We thank the reviewer for these thoughtful inquiries. The overall cost of the Li₂O-synthesized LZACO and that of the Li₂CO₃-synthesized LZACO are now compared on page 24: “In comparison, if the much more expensive Li₂O (\$210.67 kg⁻¹), instead of the Li₂CO₃ with \$12.64 kg⁻¹ cost, is used to synthesize LZACO, the raw materials cost would be \$92.94 L⁻¹, which is over four times that of the Li₂CO₃-synthesized LZACO (\$22.77 L⁻¹, as mentioned above) and also greatly exceeds the \$46.75 L⁻¹ solid-electrolyte raw materials cost for enabling reasonable ASSLB cost. Therefore, the fact that LZACO may be synthesized directly from the affordable Li₂CO₃ is crucial to the cost-effectiveness of this solid electrolyte.”.

As for the quality of the LZACO, it is not affected by replacing Li₂O with Li₂CO₃ in the raw materials. In fact, according to the EIS data in Fig. 3c, the ionic conductivity of the Li₂CO₃-synthesized LZACO is even slightly higher than that of the Li₂O-synthesized one (2.59 vs. 2.55 mS cm⁻¹ at 25 °C).

The method used to confirm the complete decomposition of Li₂CO₃ is electron energy-loss spectroscopy (EELS) in the scanning transmission electron microscopy (STEM) mode. As a microscopy-based spectroscopy, it can detect the composition in rather local areas of the material with ultrahigh spatial resolution. Consequently, even the impurities whose concentration is too low to be detected by macroscopic techniques such as X-ray diffraction can be identified by this STEM-EELS approach. Therefore, the result should be very reliable.

(10) The structural and ionic conductivity similarities between LZACO synthesized using Li₂O and Li₂CO₃. How do the XRD and EELS data support these findings, and what implications do they have for the practical application of LZACO? How does the ionic conductivity of LZACO synthesized using Li₂CO₃ compare to other solid electrolytes, and what factors contribute to its high performance?

(Response) We thank the reviewer for raising these important points. As mentioned in the main text, the EELS data confirm that the Li₂CO₃-synthesized LZACO does not contain any C. Therefore, the Li₂CO₃ used to replace Li₂O in the raw materials should completely decompose into Li₂O and CO₂ during mechanochemical synthesis, resulting in the same composition as that of the Li₂O-synthesized LZACO. The XRD and EIS tests further confirm that both the structure and the ionic conductivity of the LZACO synthesized through these two different approaches are the same. As for the implication to the practical application, it has in fact been mentioned in the response to the last question too; with the expensive Li₂O replaced by Li₂CO₃ during synthesis, the raw materials cost of LZACO decreases significantly from \$92.64 L⁻¹ to \$22.77 L⁻¹, which is much lower than the \$46.75 L⁻¹ solid-electrolyte raw materials cost for enabling reasonable total cell production cost.

The ionic conductivity of LZACO reaches 2.55 mS cm⁻¹ at 25 °C, which greatly exceeds those of the typical Zr-based oxychloride solid electrolytes (1–2 mS cm⁻¹ at 25 °C) as mentioned in the main text.

As for the factors contributing to this high performance, we believe that the more disordered atomic configuration of LZACO should be the major cause. This is discussed in-depth on page 11: “Although both LZACO and the previously reported Li-Zr-Cl-O oxychlorides rely on the amorphous phase to realize fast Li-ion transport, the specific atomic configurations in their amorphous phases are not the same, which leads to different ion transport efficiencies. In the Li-Zr-Cl-O oxychlorides, the averaged Zr-O and Zr-Cl bond lengths in the distorted Zr-O/Cl polyhedra are believed to be the key to the superior ion transport efficiency, as they flatten the energy landscape for Li-ion migration to lower the overall migration barriers²⁵. Such a disordered atomic arrangement also occurs in LZACO, but to a more pronounced degree. To characterize this behavior, we conducted the X-ray absorption spectroscopy (XAS). As shown in Supplementary Fig. 7a, the Zr *K*-edge X-ray absorption near edge structure (XANES) spectrum of LZACO does not show distinct peaks like those of ZrO₂ and ZrCl₄, but appears as a broad hump, entailing that the Zr-Cl and Zr-O bond lengths in LZACO vary in a range, instead of displaying definitive values. On this basis, more interesting behaviors were observed in the Fourier transformed extended X-ray absorption fine structure (FT-EXAFS) spectra. Unlike other Li-Zr-Cl-O solid electrolytes²⁵, the Zr-O and Zr-Cl signals in LZACO are not distinguishable from each other, but nearly merge into one broad peak (Supplementary Fig. 7b). This observation suggests that the Zr-O and Zr-Cl bond lengths in LZACO distribute in a broader range than those in the Li-Zr-Cl-O solid electrolytes²⁵. Besides, the FT-EXAFS spectrum of LZACO almost shows no signals above 2.5 Å, entailing the absence of ordering with coherence length exceeding such a distance; in contrast, other Li-Zr-Cl-O solid electrolytes still display minor signals in this range²⁵. Consistent results were observed in the phase-uncorrected wavelet transformed (WT) EXAFS spectrum too. While the Zr-O and Zr-Cl bonding still give rise to two distinct signals, respectively, in the Li-Zr-Cl-O solid electrolytes²⁵, they merge together in LZACO (Supplementary Fig. 7c). Besides, the Zr-Zr bonding at about 3 Å in the WT-EXAFS spectra of typical Li-Zr-Cl-O solid electrolytes are absent in that of LZACO too (Supplementary Fig. 7c), confirming the absence of the ordering with such large coherence lengths. These observations consistently suggest that LZACO shows a more disordered atomic configuration than the previously reported Li-Zr-Cl-O solid electrolytes. As mentioned above, such more disordered state should supposedly favor fast Li-ion transport in Zr-based oxychlorides²⁵, which explains the superior ionic conductivity of LZACO with respect to those of other Li-Zr-Cl-O solid electrolytes (2.55 vs. 1–2 mS cm⁻¹ at 25 °C²⁵).”.

Supplementary Fig. 7. a,b XANES (a) and FT-EXAFS (b) spectra of LZACO at the Zr *K*-edge. For comparison, the spectra of ZrCl₄ and ZrO₂ are also presented. **c** WT-EXAFS contour plot of LZACO at the Zr *K*-edge.

(11) The potential competitiveness of LZACO in terms of production cost due to its stability in humid air. How does this stability compare to other solid electrolytes, and what advantages does it offer for large-scale production? What specific measures or conditions are required to maintain the stability of LZACO during handling and storage, and how do these measures impact the overall production cost?

(Response) We appreciate the reviewer's insightful questions. These issues are discussed in detail on page 24: "When the LZACO powder was exposed to a dry room with the relative humidity of 3.85% (dew point set as $-20\text{ }^{\circ}\text{C}$) at $25\text{ }^{\circ}\text{C}$ for 12 h, the XRD pattern and ionic conductivity do not undergo considerable change either (Supplementary Fig. 13). In contrast, most sulfide and chloride solid electrolytes cannot survive the atmosphere with relative humidity above 1%^{2,48}; after 12-h exposure to such atmosphere, the ionic conductivities of these materials will generally decrease by at least one order of magnitude. For a 100-m^3 dry room, the power needed to maintain the 1% relative humidity is about 1.5 times that for the 3.85% relative humidity, and the difference will become larger for larger dry rooms. Consequently, the superior humidity tolerance of LZACO (stable up to around 4% relative humidity at $25\text{ }^{\circ}\text{C}$ as mentioned above) can greatly save the power needed for handling and storage, which makes its large-scale production cost-effective and also environmentally friendly. Besides, even if LZACO or any of its raw materials is accidentally exposed to the atmosphere with overly high humidity, it will not release the toxic and flammable H_2S gas like the sulfide solid electrolytes⁶¹, but is only supposed to generate the much less hazardous HCl like other chloride solid electrolytes^{61,62}. The cost-effectiveness and eco-friendliness of LZACO make it suitable for large-scale commercial utilization."

(12) The reduction and oxidation potentials of LZACO. How do these potentials compare to other solid electrolytes, and what implications do they have for the stability and performance of LZACO in all-solid-state cells?

(Response) We thank the reviewer for raising these important points. The redox potentials of LZACO and their implications are now discussed in more detail on page 27: "The oxidation potential of 4.19 V vs. Li/Li^+ greatly surpasses not only those of sulfide solid electrolytes (no higher than 3 V vs. Li/Li^+ ^{5,64}), but also those of typical Zr-based (oxy)chloride solid electrolytes such as Li_2ZrCl_6 and $\text{Li}_{1.75}\text{ZrCl}_{4.75}\text{O}_{0.5}$ (3.55 and 4.00 V vs. Li/Li^+ , respectively³), so LZACO should supposedly display superior compatibility with the high-voltage cathodes. Besides, it has been reported that the (oxy)chloride solid electrolytes can generally form stable, Li-ion conductive interphases upon oxidation, which allows for stable cycling to voltages beyond their oxidation potentials^{3,23,65-67}. Consequently, the measured oxidation potential of 4.19 V vs. Li/Li^+ entails that LZACO may direct contact high-potential cathodes such as LiCoO_2 and Ni-rich layered oxides in the all-solid-state cell, without the need of any coating layer. On the other hand, the 1.62 V vs. Li/Li^+ reduction potential of LZACO is lower than those of typical Zr-based (oxy)chlorides (2.16 and 1.79 V vs. Li/Li^+ for Li_2ZrCl_6 and $\text{Li}_{1.75}\text{ZrCl}_{4.75}\text{O}_{0.5}$, respectively³), entailing better reduction stability. Regardless, the 1.62 V vs. Li/Li^+ reduction potential remains much higher than those of most energy-dense anodes. Furthermore, since the final reduction products of LZACO would inevitably contain electronic conductors such as Zr and Al, which cannot prevent the electrons needed for further reduction to migrate through, the reduction of this material is supposed to proceed continuously^{68,69}. Such speculation is supported by the $\text{Li} | \text{LZACO} | \text{Li}$ and $\text{Li-In} | \text{LZACO} | \text{Li-In}$ symmetric cells, both of which undergo a continuous

voltage increase during cycling (Supplementary Fig. 15); in particular, the voltage of the Li | LZACO | Li cell reaches around 2 V after only 80 h of cycling at 0.1 mA cm⁻² (1 h per cycle). Therefore, to ensure the stable cycling of the all-solid-state cell, LZACO needs to be separated from the anode by a layer of more anode-compatible solid electrolyte such as LPSCl.”.

(13) *The performance of cells with LiCoO₂ and scNCM92 cathodes. How do the initial discharge capacities and Coulombic efficiencies of these cells compare to those reported in the literature for similar configurations?*

(Response) We are grateful to the reviewer’s valuable queries. The initial discharge capacities and Coulombic efficiencies of other reported cells with similar configurations are now compared with those of the LZACO-based cells in Supplementary Tables 2–3. It was found that LZACO can indeed enable superior cell performance.

Supplementary Table 2. Comparison of the initial discharge capacity and Coulombic efficiency between the LZACO-based cell in Fig. 4a and those with the similar cell configuration.

CAM	Catholyte	Specific current (mA g ⁻¹)	Initial discharge capacity (mAh g ⁻¹)	Coulombic efficiency	Ref.
LCO	LZACO	31.6 (0.2 C)	152.6	97.26%	This work
LCO	Li _{1.75} ZrCl _{4.75} O _{0.5}	79 (0.5 C)	145.6	97.73%	(3)
LCO	Li _{3.4} ZrCl _{4.6} O _{1.4}	14 (0.1 C)	160.9	97.3%	(45)
LCO	Li _{5/3} Cr _{1/3} Zr _{1/3} Cl ₄	~31.6 (0.2 C)	108.2	96.09%	(70)
LCO	Li ₂ ZrCl ₆	14 (0.1 C)	137	97.9%	(47)
LCO	Li ₃ ZrCl ₄ O _{1.5}	70 (0.5 C)	~113.5	94.7%	(25)
LCO	LiNbOCl ₄	~14 (0.1 C)	~125	95%	(21)
LCO	Li ₃ SeCl ₆	~14.6 (0.1 C)	126.2	90.3%	(50)

Supplementary Table 3. Comparison of the initial discharge capacity and Coulombic efficiency between the LZACO-based cell in Fig. 4d and those with the similar cell configuration.

CAM	Catholyte	Specific current (mA g ⁻¹)	Initial discharge capacity (mAh g ⁻¹)	Coulombic efficiency	Ref.
LiNi _{0.92} Co _{0.06} Mn _{0.02} O ₂	LZACO	20 (0.1 C)	180.8	90.27%	This work

$\text{LiNi}_{0.8}\text{Co}_{0.1}\text{Mn}_{0.1}\text{O}_2$	$\text{Li}_{1.75}\text{ZrCl}_{4.75}\text{O}_{0.5}$	20 (0.1 C)	173.96	87.31%	(3)
$\text{LiNi}_{0.88}\text{Co}_{0.11}\text{Al}_{0.01}\text{O}_2$	Li_2ZrCl_6	~24 (0.1 C)	206	85.8%	(49)
$\text{LiNi}_{0.5}\text{Co}_{0.2}\text{Mn}_{0.3}\text{O}_2$	$\text{Li}_3\text{Zr}_2\text{Si}_2\text{PO}_{12}$	15 (0.1 C)	162	85.8%	(74)
$\text{LiNi}_{0.5}\text{Co}_{0.2}\text{Mn}_{0.3}\text{O}_2$	$\text{Li}_3\text{PW}_{12}\text{O}_{40}$	~15 (0.1 C)	165.6	98.1%	(57)

(14) LZACO is the only solid electrolyte with sufficiently low cost for commercialization that can enable stable cycling at a rate as high as 10 C. How was this cost-effectiveness determined, and what factors contribute to its competitiveness compared to other solid electrolytes?

(Response) We thank the reviewer for raising this important point. The method for determining the cost is described on page 44: “The cost per unit volume of the solid electrolyte for discussion here is calculated by multiplying the cost per unit mass of the solid electrolyte and the density of the solid electrolyte. While the method of density measurement has already been described in the “Structure and morphology characterizations” section above, the cost per unit mass of the solid electrolyte was calculated straightforwardly from the average market prices of the corresponding raw materials as the industrial commodity chemicals in the year of 2024. These prices and the information sources are listed in Supplementary Table 1.”.

Supplementary Table 1. Average market prices of the industrial commodity chemicals in 2024 and the sources from which these prices are acquired.

Commodity chemical	Average market price in 2024 (\$ kg ⁻¹)	Sources
Li_2O	210.67	LB Group Co., Ltd.
LiCl	14.04	Shandong Hongyang Chemical Co., Ltd.
Li_2CO_3	12.64	Chengxin Lithium Group Co., Ltd.
ZrCl_4	11.24	CBC Metal (https://www.cbcc.com/)
AlCl_3	0.84	Asian Metal (https://www.asianmetal.cn/) Shanghai Metals Market (https://www.smm.cn/)

The factors contributing to the cost-effectiveness are discussed on page 23: “These data suggest that high-quality LZACO can be synthesized from the cost-effective Li_2CO_3 , ZrCl_4 , and AlCl_3 , whose

average market prices in 2024 as the industrial commodity chemicals are \$12.64 kg⁻¹, \$11.24 kg⁻¹, and \$0.84 kg⁻¹, respectively (Fig. 3d and Supplementary Table 1). Based on these chemical prices and the density of LZACO (measured to be 2.15 g cm⁻³; detailed procedure described in Methods), the raw materials cost of this solid electrolyte can be estimated as \$22.77 L⁻¹ (Fig. 3e). In comparison, if the much more expensive Li₂O (\$210.67 kg⁻¹), instead of the Li₂CO₃ with \$12.64 kg⁻¹ cost, is used to synthesize LZACO, the raw materials cost would be \$92.94 L⁻¹, which is over four times that of the Li₂CO₃-synthesized LZACO (\$22.77 L⁻¹, as mentioned above) and also greatly exceeds the \$46.75 L⁻¹ solid-electrolyte raw materials cost for enabling reasonable ASSLB cost. Therefore, the fact that LZACO may be synthesized directly from the affordable Li₂CO₃ is crucial to the cost-effectiveness of this solid electrolyte. In addition, since the synthesis of LZACO involves the rather cost-effective AlCl₃ (\$0.84 kg⁻¹) but does not rely on the relatively expensive LiCl (\$14.04 kg⁻¹), its raw materials cost is also lower than those of other previously reported Zr-based (oxy)chloride solid electrolytes (Fig. 3e).”.

Fig. 3 | Cost-effectiveness of LZACO. **a** Electron energy-loss spectra of the LZACO synthesized using Li₂CO₃, ZrCl₄, and AlCl₃ as the raw materials. For comparison, the C-K spectrum of CaCO₃ is also displayed. **b** XRD pattern of the LZACO synthesized using Li₂CO₃, ZrCl₄, and AlCl₃ as the raw materials. **c** Nyquist plot of the LZACO synthesized using Li₂CO₃, ZrCl₄, and AlCl₃ as the raw materials. The measurement was conducted at 25 °C. **d** Average market prices in 2024 for the industrial commodity chemicals involved in the cost estimation. **e** Raw materials costs of LZACO and two other representative Zr-based (oxy)chloride solid electrolytes, Li₂ZrCl₆ and Li_{1.75}ZrCl_{4.75}O_{0.5}.

(15) The performance of LZACO-based pouch cells. How do the mechanical properties of LZACO contribute to the fabrication and performance of these pouch cells?

(Response) We thank the reviewer for this thoughtful inquiry. The influence of the mechanical properties of LZACO on the fabrication and performance of the pouch cells is now discussed on page

32: “It should be noted that the superior mechanical compliance for the solid electrolyte has also been found crucial to the fabrication of high-quality films for pouch cells. The compliant solid electrolytes such as the sulfides and halides may easily form and maintain the intimate solid-solid contact with the assistance of the polytetrafluoroethylene (PTFE) binder; instead, if the solid electrolyte is barely compliant like the brittle oxide solid electrolytes, the films cannot enable meaningful cell performance without the addition of minor liquid electrolyte, due to the poor solid-solid contact⁷⁵. Thanks to the high mechanical compliance of LZACO, the preparation of films for pouch cells are rather convenient. Both the composite cathode films and the individual solid-electrolyte films can be easily prepared in relatively large sizes, as shown in Fig. 5c–d and Supplementary Fig. 18. Without the addition of any liquid electrolyte, the Li | LPSCI-LZACO | scNCM92 pouch cell formed by such films demonstrates a reversible capacity of about 160.8 mAh g⁻¹ under 20 mA g⁻¹ (0.1 C) at 30 °C (Fig. 5e).”.

Supplementary Fig. 18. **a** LZACO film prepared by the dry-film technology. **b–e** Composite cathode film (**b**), LZACO film (**c**), LPSCI film (**d**), and Li metal anode (**e**) for all-solid-state pouch cells. The composite cathode film, LZACO film, and LPSCI film are all prepared by the dry-film technology.

(16) The performance of LZACO-based cells at low stacking pressures. How does the mechanical compliance of LZACO enable decent performance under these conditions, and what advantages does this offer for practical applications? What potential improvements could be made to further enhance the low-pressure cell performance, and how might these improvements impact the overall usability of LZACO?

(Response) We sincerely appreciate the reviewer’s perceptive questions. The reason why the mechanical compliance of LZACO may enable decent low-pressure cell performance, along with the advantage of this characteristic for practical applications, is discussed on page 36: “The high mechanical compliance of this material allows it to efficiently alter the morphology to accommodate

the volume change of the CAM particles even under limited stresses. If the solid electrolytes employed are all highly compliant like LZACO, the ASSLBs do not have to operate under the impractically high stacking pressures above 100 MPa. In this way, the heavy instrument used to generate such pressures (e.g., the mold used for laboratory testing) is no longer needed, so that the ASSLB may eventually achieve decent energy densities for practical application.”.

The potential improvements that could be made to improve the low-pressure cell performance and the associated impact on the overall usability of LZACO are discussed on page 37: “Furthermore, it should also be noted that the low-pressure cell performance achieved here still has very large room for improvement, because, due to the poor reduction stability of LZACO (reduction potential 1.62 V vs. Li/Li⁺, as shown in Supplementary Fig. 14), the anolyte of the cell above is still the commonly used LPSCI, which is much less compliant than LZACO (as shown in Fig. 2a) and thus unlikely to perform well at low stacking pressures. If an anode-compatible solid electrolyte that is as compliant as LZACO can be developed in the future, the low-pressure cell performance similar to those at high stacking pressures should be achievable. To this end, the interphase resulting from the reduction of the solid electrolyte must be electron-blocking, so that the electrons needed for further reduction cannot migrate through to reach the unreacted solid electrolyte⁷⁹. Considering that the Zr⁴⁺ and Al³⁺ in LZACO would inevitably be reduced into electronic conductors such as Zr and Al by Li metal^{80,81}, the presence of such metal cations should preferably be eliminated in the material. Instead, the solid electrolyte needs to be populated by non-metal cations such as P⁵⁺ and B³⁺, which will only be reduced into electron-blocking species⁷⁹. If desirable mechanical compliance and ionic conductivity can be realized in such materials through rational materials design too, the cost-effective ASSLBs that can operate at sufficiently low stacking pressures for practical application may be expected.”.

Reviewer #3 (Remarks to the Author):

This paper reports Li₂O-ZrCl₄-AlCl₃ solid electrolytes exhibiting ionic conductivity of ~2.2 mS/cm and its cost-effective synthesis using Li₂CO₃ as a precursor. ASSBs using these solid electrolytes as the catholyte are demonstrated. The finding is technically important, however, the novelty of this work is not enough to satisfy the high standard of Nat. Commun. Rather this work is worth to be reported in other more specialized journals.

As for novelty, many oxy-halide SEs exhibiting mS/cm-class ionic conductivity have been reported (e.g. LiNbOCl₄, LiAlCl_{2.5}O_{0.75}, many Li-Zr-Cl-O compounds, xLi₂O-MCl_y (M = Ta, Hf)) while many of those previous reports already highlighted the role of amorphous matrix for achieving high ionic conductivity. Compare to the previous papers on oxy-halide SEs, this work does not provide new scientific insights in terms of material characterization, structure-property relation, or performances.

(Response) We sincerely thank the reviewers for the careful assessment of our manuscript and the valuable critiques, which offer us essential guidance for improvements. Regarding the reviewer's concern on novelty, we would like to emphasize that ionic conductivity is not the only metric for the advantage of solid electrolytes; as mentioned in Introduction, the mechanical compliance is equally important. In this aspect, the LZACO material reported here achieves the hardness and Young's modulus considerably lower than those of other widely studied solid electrolytes (hardness: 0.22 vs. \geq 1.19 GPa; Young's modulus: 1.41 vs. \geq 7.92 GPa), demonstrating a superior mechanical compliance (Fig. 2a). With such a desirable characteristic, the LZACO may enable the construction of all-solid-state cells that even outperform those based on more Li-ion conductive solid electrolytes, as compared comprehensively in Supplementary Tables 2–8. We believe this exceptional mechanical compliance and the superior cell performance it can lead to endow the LZACO material with significant novelty.

We apologize for not presenting this point clearly enough in the original submission. In the revised manuscript, we added one paragraph to emphasize it on page 39: “In addition to the decent cell performance, the data above also demonstrate one crucial point: the importance of the mechanical compliance for solid electrolytes. In the discussion above, the all-solid-state cells formed by LZACO are compared with those based on other solid electrolytes when different CAMs, CAM mass loadings, cycling conditions, etc., are employed (Supplementary Tables 2–8). Among all the solid electrolytes involved in such comparison, LZACO is in fact not the one with the highest ionic conductivity. Regardless, the cell constructed by LZACO can still outperform those based on other more Li-ion conductive solid electrolytes. The superior mechanical compliance of LZACO (0.22 GPa hardness and 1.41 GPa Young's modulus, as shown in Fig. 2a) plays a pivotal role here. In fact, when the ionic conductivity of the solid electrolyte exceeds 1 mS cm⁻¹, improving its mechanical compliance should be more effective in optimizing the cell performance than further increasing the ionic conductivity, as the bottleneck of the overall ion transport has shifted from the interior of the solid electrolyte to the interfacial contact between different cell components. Under such circumstances, if the cell employs highly compliant solid electrolytes such as LZACO, it can more easily achieve and maintain the intimate solid-solid contact, thereby effectively improving the overall ion transport efficiency. From this perspective, the discovery of the mechanically compliant LZACO has pointed out an effective entry point for further optimizing the comprehensive performance of the solid electrolytes that are

already highly Li-ion conductive.”.

The change of XRD patterns of LZACO after annealing should be discussed in detail. After 100 °C annealing, it seems to transform to the trigonal structure than it decomposed with the appearance of LiCl?

(Response) We thank the reviewer for raising this important point. When the crystallinity is as low as the LZACO materials studied here, the XRD cannot actually distinguish LiCl from the $C2/m$ structure that had been observed in many (oxy)chloride solid electrolytes. Therefore, although the Bragg reflections resembling those of LiCl are present as noted by the reviewer, it does not necessarily indicate the existence of LiCl. Considering that the low crystallinity prevents the conclusive crystal structure determination and that both the LiCl and the $C2/m$ structure mentioned above show cubic close-packed (ccp) anion configuration, we follow the suggestion from the other reviewer (Reviewer #1, Question 2) to refer to this phase more generally as the “ccp phase”, instead of designating a definitive crystal structure. Similarly, the specific structure of the “trigonal” phase noted by the reviewer cannot be conclusively determined either, and the only thing for sure is its hexagonal close-packed (hcp) anion configuration. Therefore, this phase is referred to as the “hcp phase” in the revised manuscript. The structural evolution noted by the reviewer is in fact the change of the molar ratio between the ccp and hcp phases after annealing. Since these two phases originally coexist in the as-synthesized LZACO, they must be energetically comparable in this material. Consequently, minor energy disruption such as that induced by the 100 °C annealing is supposed to slightly vary their molar ratio. However, such annealing does not considerably improve the crystallinity, so the material is still dominated by the highly Li-ion conductive amorphous species. This point is clarified on page 12: “To demonstrate this point, we annealed LZACO at 100 °C for 5 h, and probed the associated change of the XRD pattern and ionic conductivity. As shown in Supplementary Fig. 8a, the content of the hcp phase appears to increase slightly with respect to that of the ccp phase after annealing; as the two coexisting phases in LZACO, the hcp and ccp phases must be energetically comparable in this material, so a relatively low energy input, such as that associated with the 100 °C annealing, should be sufficient to alter the ratio between them. Regardless, both of these crystalline phases are still present after annealing, and, more importantly, they continue to occupy a negligibly small fraction in LZACO, as indicated by the diffuse Bragg reflections in the XRD pattern (Supplementary Fig. 8a). Since the majority of LZACO remains the highly Li-ion conductive amorphous species, the material still maintains a high ionic conductivity of 2.23 mS cm⁻¹ at 25 °C, almost identical with that before annealing (2.55 mS cm⁻¹ at 25 °C), as shown in Supplementary Fig. 8b.”

It is recommended to examine air- and dry-room stability of LZACO to estimate its availability in a practical cell manufacturing process.

(Response) We sincerely thank the reviewer for this constructive suggestion. While the humidity tolerance test in our original submission was conducted in air using an environmental chamber, in the revised manuscript we also supplemented the testing results in a dry room. The data and discussion are presented on page 24: “Besides, decent humidity tolerance was observed in the dry room too. When the LZACO powder was exposed to a dry room with the relative humidity of 3.85% (dew point set as -20 °C) at 25 °C for 12 h, the XRD pattern and ionic conductivity do not undergo considerable change

either (Supplementary Fig. 13):”.

Supplementary Fig. 13. Humidity tolerance test conducted in a dry room. **a** XRD patterns of the LZACO powders before and after exposure to the air with 3.85% relative humidity at 25 °C. **b** Nyquist plots of LZACO before and after exposure to the air with 3.85% relative humidity at 25 °C. The exposure experiment was conducted using the powder samples, while the EIS measurements were conducted on the pellets prepared by cold pressing the exposed or non-exposed powders under 300 MPa, at 25 °C.

Reviewer #4 (Remarks to the Author):

This study emphasizes the critical role of mechanical compliance in solid electrolytes for all-solid-state batteries, alongside ionic conductivity and cost. In this regard, the authors introduces $x\text{Li}_2\text{O}-(1-y)\text{ZrCl}_4-y\text{AlCl}_3$ (LZACO, $x = 1.0 \sim 1.8$, $y = 0.2 \sim 0.5$) as a novel oxychloride electrolyte with remarkably low hardness (0.22 GPa) and Young's modulus (1.41 GPa), targeting commercialization viability. LZACO is synthesized via mechanochemical milling of Li_2CO_3 , ZrCl_4 , and AlCl_3 , achieving an ionic conductivity of 2.55 mS cm^{-1} at 25°C through amorphous-phase dominance. The authors estimated that the cost of raw material is using a log-log scaling model, with Li_2CO_3 replacing costly Li_2O . Given the result from cost modeling, the authors suggest the low cost of raw material of $\$10.61 \text{ kg}^{-1}$ for LZACO by substituting Zr^{4+} and Al^{3+} for Ta^{5+} from $1.6\text{Li}_2\text{O}-\text{TaCl}_5$ and replacing Li_2O to Li_2CO_3 for synthesis. The synthesized LZACO exhibits superior deformability compared to sulfide/halide electrolytes and the phase-field modeling further supports the important of low hardness and Young's modulus. The authors demonstrated the effect of improved mechanical compliance of LZACO by showing long term cycling of all-solid-state batteries using LiCoO_2 and NCM92 cathodes achieving $3.62\text{--}3.92 \text{ mAh cm}^{-2}$ areal capacity (85–90% retention after 100 cycles) at 25°C . While the authors demonstrate the excellent electrochemical properties of LZACO in all solid state batteries (ASSBs), I believe this paper ultimately falls short of the requirements for publication in nature communications for the reasons articulated below.

1. Page 6, line 2: The authors propose a chemical price threshold of $\$50 \text{ kg}^{-1}$ for the commercialization of ASSBs. However, this paper provides no supporting evidence for this value, nor do the references cited offer any justification. To substantiate this claim, the authors should present a comprehensive market analysis, including data from at least the past two years, as well as a universal framework for estimating the chemical cost associated with ASSB commercialization.

(Response) We are deeply grateful to the reviewer for his/her thorough review and insightful feedbacks; the constructive suggestions he/she raised are highly valuable for improving our work. Although the $\$50 \text{ kg}^{-1}$ solid-electrolyte cost has been referred to as the threshold for commercializing all-solid-state batteries in the studies from different research groups recently, there is indeed no direct evidence supporting this point in literature, as noted by the reviewer. Therefore, the meaning of this benchmark needs to be carefully discussed. Fortunately, the kind of comprehensive, in-depth market analysis needed for such a purpose has been published only a few months ago by Pasta et al. (Nature Energy 10, 135–147, 2025), and they also assumed the solid-electrolyte cost to be the aforementioned $\$50 \text{ kg}^{-1}$ for estimation. Interestingly, under this assumption, the all-solid-state batteries with the energy-dense Li metal anode still cost more than that of the commercial liquid-state Li-ion batteries ($\$158 \text{ kWh}^{-1}$ vs. $\$126 \text{ kWh}^{-1}$); in particular, even if the solid electrolyte is assumed free, the cost of all-solid-state batteries in such comprehensive estimation is still higher ($\$134 \text{ kWh}^{-1}$). Under this circumstance, whether the all-solid-state batteries can be successfully commercialized depends on whether its energy-density and safety advantages can justify the relatively high cost in the market, and this is very difficult to definitively predict for now. In other words, **the $\$50 \text{ kg}^{-1}$ solid-electrolyte cost only leads to a relatively reasonable all-solid-state batteries production cost with potential commercial viability; it cannot guarantee a lower cell production cost than those of commercial Li-ion batteries.**

Besides, the aforementioned estimation that the $\$50 \text{ kg}^{-1}$ solid-electrolyte cost may enable a relatively reasonable cell production cost of $\$158 \text{ kWh}^{-1}$ is based on the assumption that the solid electrolyte is $\text{Li}_7\text{P}_3\text{S}_{11}$. If other solid electrolytes were used, the mass of the solid electrolyte needed to reach the same solid-electrolyte volume fraction in the composite cathode and the same solid-electrolyte layer thickness would be different. Under such circumstances, the $\$50 \text{ kg}^{-1}$ solid-electrolyte cost might no longer lead to reasonable cell production costs like $\$158 \text{ kWh}^{-1}$. Clearly, the cost per unit mass of the solid electrolyte is not the ideal metric for discussing the cost-effectiveness of solid electrolytes, as the desired values for different solid electrolytes are different. In comparison, the cost per unit volume of the solid electrolyte might be more suitable for this purpose; the optimal solid-electrolyte volume in each kWh of batteries (depending on the optimal solid-electrolyte volume fraction in the composite cathode and the optimal solid-electrolyte layer thickness) does not vary with the type of the solid electrolyte, so the desired cost per unit volume of the solid electrolyte for a given cell configuration would be the same for all the solid electrolytes, which makes it a more appropriate metric for comparison. In the cell configuration used in the aforementioned study (adapted from the batteries in Volkswagen ID. 3; Nature Energy 10, 135–147, 2025), the $\$50 \text{ kg}^{-1}$ gravimetric solid-electrolyte cost assumed for $\text{Li}_7\text{P}_3\text{S}_{11}$, a material with the density of 1.870 g cm^{-3} , can be easily converted to the volumetric solid-electrolyte cost of $\$93.50 \text{ L}^{-1}$. Considering that the processing cost of the solid electrolytes cannot be precisely predicted due to the uncertainty of the method for mass production, we assume the processing cost of solid electrolytes will not exceed 50% of their total costs, which is also the situation of most chemicals in the Li-ion battery industry. In this way, in order to ensure the solid-electrolyte cost below $\$93.50 \text{ L}^{-1}$, the raw materials cost of the solid electrolyte must not exceed $\$46.75 \text{ L}^{-1}$. In the revised manuscript, this $\$46.75 \text{ L}^{-1}$ solid-electrolyte raw materials cost is used as the reference for discussion, and, as mentioned above, we particularly emphasized that this only leads to a relatively reasonable cell production cost, instead of guaranteeing lower costs than those of commercial Li-ion batteries. Once again, we sincerely thank the reviewer for helping us clarify this issue, which has been confusing us and probably other researchers for long.

The contents above are presented in the “Cost-effectiveness” section, on page 21: “In many recent studies, the $\$50 \text{ kg}^{-1}$ solid-electrolyte cost is referred to as the threshold for the successful commercialization of ASSLBs^{3,38,57}, but this standard is in fact questionable. First and foremost, the solid-electrolyte cost below $\$50 \text{ kg}^{-1}$ cannot actually ensure that the total production cost of ASSLBs lies below those of commercial Li-ion batteries. Recently, Pasta et al. have conducted an in-depth cost study based on a comprehensive market analysis⁵⁸. Using the actual cell parameters for the batteries in Volkswagen ID. 3, they compared the cost of the energy-dense ASSLBs and the commercial liquid-state Li-ion batteries. Such calculations suggest that the ASSLBs with the aforementioned $\$50 \text{ kg}^{-1}$ solid-electrolyte cost is still more expensive than the commercial Li-ion batteries ($\$158 \text{ kWh}^{-1}$ vs. $\$126 \text{ kWh}^{-1}$). In particular, even if the solid electrolyte is assumed free, the ASSLBs still cost $\$134 \text{ kWh}^{-1}$, exceeding the $\$126 \text{ kWh}^{-1}$ cost for commercial Li-ion batteries. Under this circumstance, the commercial viability of the ASSLBs depends on whether their energy-density and safety advantages can justify such high costs in the market, but this cannot be predicted definitively for now. That is, the $\$50 \text{ kg}^{-1}$ solid-electrolyte cost should by no means be regarded as a threshold guaranteeing an ASSLB cost that is lower than those of commercial Li-ion batteries; instead, it is merely a cost that may yield relatively reasonable ASSLB cost, with potential commercial viability. Additionally, compared with the cost per unit mass like the $\$50 \text{ kg}^{-1}$ discussed above, the cost per unit volume should be more

appropriate for discussing the cost-effectiveness of solid electrolytes. In all-solid-state batteries, the optimal amount of the solid electrolyte is not actually defined by the mass, but by the volume: the composite cathode with the optimal tortuosity needs the solid electrolyte to occupy a certain volume fraction, while the balance between energy density and safety is realized by adjusting the thickness of the solid-electrolyte layer. Consequently, no matter what kind of solid electrolyte is employed, the optimal solid-electrolyte volume in each kWh of batteries would be the same. On this basis, as long as the cost per unit volume of the solid electrolyte is determined, the solid-electrolyte cost per kWh of batteries, i.e., the ultimate metric for the cost-effectiveness of solid electrolytes, is determined too. Therefore, compared with the widely discussed cost per unit mass of the solid electrolyte, the cost per unit volume of the solid electrolyte is more suitable for discussion. In the aforementioned study by Pasta et al., the solid electrolyte is assumed to be $\text{Li}_7\text{P}_3\text{S}_{11}$, so, using the density of 1.870 g cm^{-3} ⁵⁹ for this material, it can be easily found that the $\$50 \text{ kg}^{-1}$ gravimetric solid-electrolyte cost that may enable relatively reasonable cell production cost corresponds to a volumetric solid-electrolyte cost of $\$93.50 \text{ L}^{-1}$. Here, it should be noted that such a $\$93.50 \text{ L}^{-1}$ cost includes both the raw materials cost and the processing cost. Considering that the method for the mass production of solid electrolytes has not been finally determined, the processing cost cannot be accurately estimated. Regardless, if assuming the processing costs of the solid electrolyte is no higher than 50% of its total cost, like most of the other chemicals involved in Li-ion batteries^{58,60}, it can be found that, to ensure the solid-electrolyte cost below $\$93.50 \text{ L}^{-1}$, the raw materials cost of the solid electrolyte must not exceed $\$46.75 \text{ L}^{-1}$. In the discussion below, this will be regarded as the raw materials cost of the solid electrolyte that can enable relatively reasonable costs for ASSLBs.”.

2. Supplementary table 1 and 2: The authors present laboratory-scale chemical prices for solid electrolyte fabrication in Supplementary Tables 1 and 2. However, these prices are derived solely from two chemical suppliers, Alfa Aesar and Aladdin, and are limited to data from January 2025. Relying on arbitrarily selected suppliers over such a narrow timeframe undermines the credibility of the economic and scientific analysis. A broader dataset encompassing multiple suppliers and a longer time period is necessary to provide a robust evaluation.

(Response) We thank the reviewer for this insightful suggestion. The prices that are more representative of the actual situations are indeed needed. Following the reviewer’s suggestion, we cease to use the prices inferred from the laboratory-scale chemicals, but utilize the average market prices of the corresponding industrial commodity chemicals in the year of 2024 (the information sources are specified along with the prices in Supplementary Table 1, which is also displayed below). It should be noted that the raw materials of some widely studied solid electrolytes, such as the high-purity Li_2S needed for fabricating $\text{Li}_3\text{PS}_6\text{Cl}$, are not yet available as the bulk industrial commodities; this is also one of the major reasons why they are too expensive to commercialize for now. In light of this fact, we no longer compare the costs of such solid electrolytes in the revised manuscript. Instead, our comparison focuses only on the Zr-based (oxy)chloride solid electrolytes, whose raw materials are all available as industrial commodity chemicals. The related discussion is made on page 23: “These data suggest that high-quality LZACO can be synthesized from the cost-effective Li_2CO_3 , ZrCl_4 , and AlCl_3 , whose average market prices in 2024 as the industrial commodity chemicals are $\$12.64 \text{ kg}^{-1}$, $\$11.24 \text{ kg}^{-1}$, and $\$0.84 \text{ kg}^{-1}$, respectively (Fig. 3d and Supplementary Table 1). Based on these chemical prices and the density of LZACO (measured to be 2.15 g cm^{-3} ; detailed procedure described

in Methods), the raw materials cost of this solid electrolyte can be estimated as $\$22.77 \text{ L}^{-1}$ (Fig. 3e). In comparison, if the much more expensive Li_2O ($\$210.67 \text{ kg}^{-1}$), instead of the Li_2CO_3 with $\$12.64 \text{ kg}^{-1}$ cost, is used to synthesize LZACO, the raw materials cost would be $\$92.94 \text{ L}^{-1}$, which is over four times that of the Li_2CO_3 -synthesized LZACO ($\$22.77 \text{ L}^{-1}$, as mentioned above) and also greatly exceeds the $\$46.75 \text{ L}^{-1}$ solid-electrolyte raw materials cost for enabling reasonable ASSLB cost. Therefore, the fact that LZACO may be synthesized directly from the affordable Li_2CO_3 is crucial to the cost-effectiveness of this solid electrolyte. In addition, since the synthesis of LZACO involves the rather cost-effective AlCl_3 ($\$0.84 \text{ kg}^{-1}$) but does not rely on the relatively expensive LiCl ($\$14.04 \text{ kg}^{-1}$), its raw materials cost is also lower than those of other previously reported Zr-based (oxy)chloride solid electrolytes (Fig. 3e).”.

Fig. 3 | Cost-effectiveness of LZACO. **a** Electron energy-loss spectra of the LZACO synthesized using Li_2CO_3 , ZrCl_4 , and AlCl_3 as the raw materials. For comparison, the C-K spectrum of CaCO_3 is also displayed. **b** XRD pattern of the LZACO synthesized using Li_2CO_3 , ZrCl_4 , and AlCl_3 as the raw materials. **c** Nyquist plot of the LZACO synthesized using Li_2CO_3 , ZrCl_4 , and AlCl_3 as the raw materials. The measurement was conducted at 25°C . **d** Average market prices in 2024 for the industrial commodity chemicals involved in the cost estimation. **e** Raw materials costs of LZACO and two other representative Zr-based (oxy)chloride solid electrolytes, Li_2ZrCl_6 and $\text{Li}_{1.75}\text{ZrCl}_{4.75}\text{O}_{0.5}$.

Supplementary Table 1. Average market prices of the industrial commodity chemicals in 2024 and the sources from which these prices are acquired.

Commodity chemical	Average market price in 2024 ($\$ \text{ kg}^{-1}$)	Sources
Li_2O	210.67	LB Group Co., Ltd.

LiCl	14.04	Shandong Hongyang Chemical Co., Ltd.
Li ₂ CO ₃	12.64	Chengxin Lithium Group Co., Ltd.
ZrCl ₄	11.24	CBC Metal (https://www.cbccie.com/)
AlCl ₃	0.84	Asian Metal (https://www.asianmetal.cn/) Shanghai Metals Market (https://www.smm.cn/)

3. In addition to the chemical cost of solid electrolytes, the overall price of ASSBs is influenced by various hidden manufacturing expenses, such as synthesis processing fees, chemical maintenance costs, and others.

(Response) We fully agree with the reviewer that the discussion of the solid-electrolyte cost needs to be placed in the context of the overall battery cost. As mentioned above, based on the in-depth battery cost analysis reported recently by Pasta et al. (Nature Energy 10, 135–147, 2025), we inferred that the \$46.75 L⁻¹ solid-electrolyte raw materials cost should lead to a relatively reasonable all-solid-state battery cost with potential commercial viability. This is used as the reference for the cost analysis in the revised manuscript.

Moreover, material costs can fluctuate significantly during mass production for commercialization.

(Response) The materials cost fluctuation is indeed an important factor that needs to be considered. As mentioned above, we follow the reviewer's suggestion and utilize the average market prices of the industrial commodity chemicals in 2024 to minimize the distraction from this factor.

Importantly, LZACO requires co-utilization with sulfide-based solid electrolytes or Li-alloy anodes due to its incompatibility with Li metal anodes, which reduces the energy density of ASSBs and increases manufacturing costs.

(Response) We thank the reviewer for raising this important point. The discovery of LZACO alone indeed cannot ensure the successful commercialization of all-solid-state Li batteries; it is only a relatively affordable catholyte, but presently many other cell components, such as the anolyte and the energy-dense anodes, remain rather expensive, as noted by the reviewer. Therefore, LZACO only meets one of the many necessary but insufficient conditions for commercializing all-solid-state Li batteries. We thank the reviewer for helping us clarify this important point. To avoid misleading the readers, we emphasize it on page 25: "Although the data above are indicative of decent cost-effectiveness for LZACO, it must also be emphasized that this material alone cannot address the cost issue for the commercialization of ASSLBs. Similar to other Zr-based oxychloride solid electrolytes⁶³, LZACO is not supposed to be compatible with the reductive anodes such as Li metal. Therefore, it

may only serve as the catholyte, which still needs to be separated from the anode by the overly expensive solid electrolytes such as the sulfides³. Besides, the thin Li metal foil needed to enable high energy densities for ASSLBs is also too expensive to make the cost of ASSLBs equal to that of commercial liquid-state batteries⁵⁸. To endow ASSLBs with commercial viability, the cost of these components must be lowered too. The identification of cost-effective catholytes such as LZACO only meets one necessary but insufficient condition for addressing the cost issue of ASSLBs.”.

Consequently, laboratory-scale chemical pricing alone is insufficient to convincingly demonstrate LZACO's advantages for ASSB commercialization.

(Response) We fully agree with the reviewer that the laboratory-scale chemical pricing is not convincing enough. As mentioned above, we utilized the market prices of the industrial commodity chemicals instead in the revised manuscript. Once again, we sincerely thank the reviewer for helping us improving the reliability of the cost analysis.

4. Page 9, line 3-16: The authors attribute LZACO's high ionic conductivity to material amorphization, referencing studies on Zr-based halide solid electrolytes. While "intermediate compositions" have been reported in prior research in references, the correlation between these compositions, amorphous phases in synthesized materials, and their impact on ionic conductivity remains unclear. As ball milling-induced amorphization introduces atomic site disorder within materials, this phenomenon should be thoroughly discussed in relation to its influence on LZACO's ion conductivity.

(Response) We thank the reviewer for this insightful and constructive suggestion. The improved ionic conductivity should indeed be discussed from the perspective of atomic configurations, instead of only the coexisting crystalline phases and “intermediate compositions”. In the revised manuscript, we made such discussion based on the X-ray absorption spectroscopy on page 11: “Although both LZACO and the previously reported Li-Zr-Cl-O oxychlorides rely on the amorphous phase to realize fast Li-ion transport, the specific atomic configurations in their amorphous phases are not the same, which leads to different ion transport efficiencies. In the Li-Zr-Cl-O oxychlorides, the averaged Zr-O and Zr-Cl bond lengths in the distorted Zr-O/Cl polyhedra are believed to be the key to the superior ion transport efficiency, as they flatten the energy landscape for Li-ion migration to lower the overall migration barriers²⁵. Such a disordered atomic arrangement also occurs in LZACO, but to a more pronounced degree. To characterize this behavior, we conducted the X-ray absorption spectroscopy (XAS). As shown in Supplementary Fig. 7a, the Zr K-edge X-ray absorption near edge structure (XANES) spectrum of LZACO does not show distinct peaks like those of ZrO₂ and ZrCl₄, but appears as a broad hump, entailing that the Zr-Cl and Zr-O bond lengths in LZACO vary in a range, instead of displaying definitive values. On this basis, more interesting behaviors were observed in the Fourier transformed extended X-ray absorption fine structure (FT-EXAFS) spectra. Unlike other Li-Zr-Cl-O solid electrolytes²⁵, the Zr-O and Zr-Cl signals in LZACO are not distinguishable from each other, but nearly merge into one broad peak (Supplementary Fig. 7b). This observation suggests that the Zr-O and Zr-Cl bond lengths in LZACO distribute in a broader range than those in the Li-Zr-Cl-O solid electrolytes²⁵. Besides, the FT-EXAFS spectrum of LZACO almost shows no signals above 2.5 Å, entailing the absence of ordering with coherence length exceeding such a distance; in contrast, other Li-Zr-Cl-O solid electrolytes still display minor signals in this range²⁵. Consistent results were

observed in the phase-uncorrected wavelet transformed (WT) EXAFS spectrum too. While the Zr-O and Zr-Cl bonding still give rise to two distinct signals, respectively, in the Li-Zr-Cl-O solid electrolytes²⁵, they merge together in LZACO (Supplementary Fig. 7c). Besides, the Zr-Zr bonding at about 3 Å in the WT-EXAFS spectra of typical Li-Zr-Cl-O solid electrolytes are absent in that of LZACO too (Supplementary Fig. 7c), confirming the absence of the ordering with such large coherence lengths. These observations consistently suggest that LZACO shows a more disordered atomic configuration than the previously reported Li-Zr-Cl-O solid electrolytes. As mentioned above, such more disordered state should supposedly favor fast Li-ion transport in Zr-based oxychlorides²⁵, which explains the superior ionic conductivity of LZACO with respect to those of other Li-Zr-Cl-O solid electrolytes (2.55 vs. 1–2 mS cm⁻¹ at 25 °C²⁵).

Supplementary Fig. 7. a,b XANES (a) and FT-EXAFS (b) spectra of LZACO at the Zr K-edge. For comparison, the spectra of ZrCl₄ and ZrO₂ are also presented. **c** WT-EXAFS contour plot of LZACO at the Zr K-edge.

5. The synthesis of amorphized Li₂ZrCl₆ solid electrolytes via anion substitution (e.g., Cl⁻ replaced by other anions) has already been extensively explored in previous studies (e.g., J. Cheng et al., *Journal of Energy Storage*, 2024, 89, 111700, D. Xu et al., *J. Mater. Chem. A*, 2024, 12, 27694, F. Hussain et al., *npj Comput Mater*, 2024, 10, 148, and etc.). Similarly, substituting Zr with Ta has been reported (e.g., X. Liu et al., *Energy Storage Materials*, 2024, 72, 103737 and etc.). Furthermore, the economic evaluation of oxyhalide solid electrolytes was addressed in earlier work (L. Hu, *Nat Commun.*, 2023, 14, 3807). While this paper provides additional insights into LZACO's mechanical properties, its lack of novelty regarding the material itself diminishes its overall contribution to the field.

(Response) We deeply appreciate the reviewer's valuable and thoughtful critique. In fact, it is the existence of these similar compositions in literature that highlights the novelty and importance of our discovery. Prior to the present study, the Zr-based (oxy)chloride solid electrolytes such as Li₂ZrCl₆ and Li_{1.75}ZrCl_{4.75}O_{0.5} are commonly believed to display relatively poor mechanical compliance. In this context, the present study discovers that one of this category of materials, LZACO, may in fact achieve the mechanical compliance even greatly surpassing those of many widely studied solid electrolytes (Fig. 2a). In the meantime, other advantages of Zr-based (oxy)chloride solid electrolytes, such as the cost-effectiveness, relatively high ionic conductivity, oxidation stability, etc., are not compromised either. From this perspective, we believe the discovery of LZACO makes important contribution to the research field of all-solid-state batteries.

We sincerely thank the reviewer for helping us clarify this point. In the revised manuscript, we emphasize it on page 19: “In summary, the computational and experimental data presented in Fig. 2 above consistently suggest that the LZACO solid electrolyte reported here is not plagued by the relatively poor mechanical compliance in typical Zr-based (oxy)chloride solid electrolytes such as the Li_2ZrCl_6 and LZCO for comparison above. On the contrary, the mechanical compliance of LZACO even surpasses those of many widely studied sulfide and halide solid electrolytes, as shown in Fig. 2a. Therefore, although many derivatives and chemical modifications of Zr-based (oxy)chloride solid electrolytes have already been reported in literature^{45,54-56}, the LZACO material with the unique advantage in mechanical compliance can still play a distinctive role in ASSLBs.”.

6. The authors highlight the low Young's modulus and hardness of LZACO as critical factors for stable contact between LZACO and cathode active materials during ASSB cycling. However, their experiments employed an impractically high stack pressure of 190 MPa with low cathode active material loading of $\sim 5.5 \text{ mg}\cdot\text{cm}^{-2}$, which fails to substantiate the mechanical compliance benefits of LZACO under realistic conditions. Furthermore, Figures 6 and 7 show that LZACO cells with high cathode loading of $> 25 \text{ mg}\cdot\text{cm}^{-2}$ cycled at low stack pressure of 5 MPa do not outperform previously reported oxyhalide solid electrolytes in ASSBs in terms of capacity retention. A systematic comparison of stack pressure effects on ASSB cycling using LZACO versus other solid electrolytes such as LZCO or LZO would provide more compelling evidence for the importance of mechanical compliance.

(Response) We sincerely thank the reviewer for this insightful suggestion. In the revised manuscript, the low-pressure performance of the cell formed by LZACO was compared with those formed by LZCO and LZO, respectively. Besides, the comparison was also made with other cells operating at low stacking pressures in literature. It was found that the mechanically compliant LZACO indeed displays advantages in this regard. The results are presented on page 36: “For comparison, we also tried to replace the LZACO in the cells above with Li_2ZrCl_6 (LZO) or LZCO (both are much less compliant than LZACO, as shown in Fig. 2a), and then conducted the cycling tests under the same conditions, especially the low stacking pressure of 5 MPa. The variation tendency of the cell performance was found consistent with that of the mechanical compliance disclosed in Fig. 2a. As shown in Supplementary Fig. 19, the least compliant LZO solid electrolyte results in the poorest cycling performance; the capacity retention after 100 cycles is only 51.09%. In comparison, the relatively more compliant LZCO enables better cycling stability, with the capacity retention being 69.21% after 100 cycles. As for LZACO, its mechanical compliance greatly exceeds those of both LZO and LZCO (Fig. 2a), so this material enables the most stable cycling performance; after 100 cycles, the capacity retention is 91.52%. In addition to the two typical Zr-based (oxy)chloride solid electrolytes for comparison above, the mechanically compliant LZACO can also enable better cell performance than many other widely studied solid electrolytes. As shown in Supplementary Table 8, the cycling performance achieved by the LZACO-based cells at 5 MPa even surpasses that of certain cells at higher pressures such as 10 MPa^{11,35,78}. These observations demonstrate the advantage of the mechanically compliant LZACO in enabling decent cell performance at low stacking pressures, which is a prerequisite for constructing practical ASSLBs”.

Supplementary Fig. 19. a–c Long-term cycling performance of the Li₁₃Si₄ | LPSCI-LZC | scNCM92 cell (a), Li₁₃Si₄ | LPSCI-LZCO | scNCM92 cell (b), and Li₁₃Si₄ | LPSCI-LZACO | scNCM92 cell (c) under 5 MPa at 30 °C.

Supplementary Table 8. Low-pressure cycling performance of the LZACO-based ASSLBs and those based on other solid electrolytes.

CAM	Catholyte	Specific current (mA g ⁻¹)	Temperature (°C)	Stacking pressure (MPa)	Capacity retention	Ref.
LiNi _{0.92} C _{0.06} Mn _{0.02} O ₂	LZACO	66 (0.33 C)	30	5	95% (69 cycles)	This work
					90% (119 cycles)	
					80% (215 cycles)	

LiCoO_2	$\text{Li}_{1.625}\text{Al}_{0.375}\text{Zr}_{0.625}\text{Cl}_{5.25}$	~ 70 (0.5 C)	30	9.62	$\sim 58.71\%$ (282 cycles)	(78)
LiCoO_2	Li_2ZrCl_6	~ 70 (0.5 C)	30	9.62	$\sim 48.74\%$ (109 cycles)	(78)
$\text{LiNi}_{0.83}\text{Co}_{0.11}\text{Mn}_{0.06}\text{O}_2$	Li_3InCl_6	n/a	80	2	65% (50 cycles)	(35)
				10	93% (50 cycles)	(35)
$\text{LiNi}_{0.5}\text{Co}_{0.2}\text{Mn}_{0.3}\text{O}_2$	$\text{Li}_{0.388}\text{Ta}_{0.238}\text{La}_{0.475}\text{Cl}_3$	~ 66 (0.44 C)	30	~ 2	81.6% (100 cycles)	(11)

Responses to Reviewers' Comments

Unless otherwise specified, all page numbers, figure numbers, and reference numbers mentioned below refer to the numbering sequence in the revised manuscript.

The contents copied from the revised manuscript are highlighted in yellow.

Reviewers' comments:

Reviewer #3 (Remarks to the Author):

The reviewer considers highlighting mechanical properties of LZACO does not confer further novelty for the publication since the incorporation of Al, Cl, and O is quite expectable considering the reported viscoelasticity of Li-Al-Cl-O materials (Nat. Energy, 8, 1221 (2023)).

(Response) We sincerely thank the reviewer for his/her valuable critique. As noted by the reviewer, the superior mechanical compliance of the Li-Al-Cl-O materials is achieved in the viscoelastic state, rather than the powdery state displayed by most inorganic solid electrolytes. Despite its unique advantage in maintaining intimate electrode-electrolyte contact, the viscoelastic state is not ideal for the pouch-cell fabrication techniques that are potentially applicable in the large-scale roll-to-roll manufacturing, as the material in this state would easily overflow under the pressure applied during rolling, making the membrane formation and cell assembly rather difficult (this could be the reason why pouch-cell performance is not reported in the paper noted by the reviewer). In contrast, the powdery solid electrolytes like the sulfide and halide solid electrolytes are more compatible with these procedures (Adv. Mater. 35, 2209074, 2023), but they generally display poor mechanical compliance. Therefore, preferably superior mechanical compliance should be achieved without inducing the viscoelastic state. Incorporating Al, Cl, and O in an inorganic solid electrolyte might be expected to induce the viscoelastic state similar to that observed in the Li-Al-Cl-O materials, but the mechanical compliance achievable without inducing viscoelasticity remains unknown; in fact, it is also quite possible that, as long as the material does not become viscoelastic but remains powdery, it would only show the kind of mechanical compliance displayed by other powdery solid electrolytes. Under this circumstance, the present study not only achieves the mechanical compliance greatly exceeding those of typical powdery solid electrolytes in the non-viscoelastic, powdery state that is suitable for pouch-cell fabrication, but also presents decent pouch-cell data that are absent in the paper noted by the reviewer (Nat. Energy 8, 1221, 2023). Consequently, we believe the discovery reported here is not easily expectable. We hope this can alleviate the reviewer's concern about the novelty of the present study.

Furthermore, the measured Young's modulus value of LZACO should be explained carefully. As the authors stated "the mechanical compliance of LZACO cannot possibly surpass the viscoelastic ones, such as the polymer solid electrolytes....", generally, Young's modulus of solid materials will be inorganic > polymeric materials. The measured Young's modulus of LZACO is 1.41, which is even lower than many polymers such as PETs, nylons, and etc. Is there any possibility that the measurement

method underestimate Young's modulus? In this context, it is recommended to measure some representative materials including sulfides, halides, and polymers with the same method. If $1.4\text{Li}_2\text{O}\cdot 0.75\text{ZrCl}_4\cdot 0.25\text{AlCl}_3$ (LZACO) shows the lower Young's modulus than polymers, the appropriate chemical explanation should be discussed in detail.

(Response) We very much appreciate the reviewer's insightful suggestion. In the revised manuscript, we measured the Young's moduli of $\text{Li}_6\text{PS}_5\text{Cl}$ (as the representative sulfide solid electrolyte), Li_2ZrCl_6 (as the representative halide solid electrolyte), and a polymer solid electrolyte whose Young's modulus has been reported before in *Nat. Commun.* 14, 2301, 2023. It was found that the Young's moduli we measured align well with those reported in literature, as shown in Supplementary Fig. 9 (also pasted below). These data should safely exclude the possibility that the Young's moduli were underestimated.

As for the PETs and nylons mentioned by the reviewer, they are in fact particularly stiff polymers, with Young's moduli even higher than those of certain inorganic materials. For typical polymer solid electrolytes, their Young's moduli generally lie below 200 MPa (Joule 6, 2372–2389, 2022; Nature Communications 15, 2500, 2024), which are indeed lower than the 1.41 GPa Young's modulus measured for LZACO.

The contents above are presented on page 15: “The Young's moduli of a few such solid electrolytes were measured too, and the results we obtained align well with those reported in literature (Supplementary Fig. 9), confirming the reliability of our mechanical characterization.”

Supplementary Fig. 9. Comparison of the Young's moduli we measured and those reported in literature. PIL represents a polymer solid electrolyte whose Young's modulus has been reported before in *Nat. Commun.* 14, 2301, 2023; it is prepared via ultraviolet light-initiated copolymerization of 1-allyl-1-methyl-pyrrolidinium bis(trifluoromethanesulfonyl) imide ionic liquid, vinyl ethylene carbonate, and polyfluorinated crosslinker 2,2,3,3,4,4,5,5-octafluoro-1,6-hexanediol diacrylate (OFHDODA) blended with lithium salt (LiTFSI) in the molar ratio of 8:3:2. The reference data for $\text{Li}_6\text{PS}_5\text{Cl}$ and Li_2ZrCl_6 are taken from *Joule* 8, 1–16, 2024 and *Angew. Chem. Int. Ed.* 64 (36), e202510359, 2025, respectively.

Reviewer #4 (Remarks to the Author):

The authors have made clear efforts to address the concerns raised in the first review round through additional experiments and analyses. These revisions have strengthened the manuscript; however, the work may not yet fully meet the exceptionally high bar of novelty generally expected for publication in Nature Communications.

The Li–Zr–Cl–O family of solid electrolytes investigated in this study has already been extensively reported in the literature, and Al-doping strategies have likewise been previously explored (e.g., Energy Storage Materials, Vol. 70, June 2024, 103444; Tan et al., Nano Research 17, 8826–8833, 2024, among others). In addition, the synthesis route employed in the present work—ball milling—while effective for laboratory-scale preparation, is associated with relatively high production costs, hindering scalability and commercial viability. Consequently, it is difficult to regard the proposed material as a cost-competitive candidate for large-scale application in solid-state batteries.

(Response) We are deeply grateful to the reviewer for his/her insightful feedback, which provides us with valuable guidance for improving the present study. In the revised manuscript, we carefully analyzed the processing cost associated with large-scale synthesis as well. Our results suggest that ball milling does often induce unacceptably high processing cost, as the reviewer noted. However, we also found that such cost would decrease considerably with the synthesis time, which is closely related to several major cost drivers, such as the electricity cost and the instrument depreciation cost; the reason why ball milling is usually considered costly lies in the fact that most solid electrolytes need to be synthesized through long-duration ball milling (typically 30–40 h, sometimes over 80 h).

Fortunately, the ball milling time needed to synthesize LZACO can be significantly reduced, as long as the appropriate raw materials are used. If LZACO is synthesized from Li_2CO_3 instead of Li_2O , the ball milling time needed to reach the same ionic conductivity may be reduced drastically from 30 to 4 h (Fig. 3a–c, also pasted below), which decreases the processing cost from $\$63.79 \text{ L}^{-1}$ to $\$15.31 \text{ L}^{-1}$, as shown in Fig. 3d and Supplementary Tables 1–5 (the basis for such cost estimation is elaborated in Methods, and is also pasted below). Given that Li_2CO_3 is much more affordable than Li_2O too ($\$12.64 \text{ kg}^{-1}$ vs. $\$210.67 \text{ kg}^{-1}$), this raw material replacement reduces the total cost of LZACO from $\$156.76 \text{ L}^{-1}$ to only $\$43.70 \text{ L}^{-1}$, considerably lower than the $\$93.50 \text{ L}^{-1}$ solid-electrolyte cost for enabling reasonable cost of all-solid-state batteries.

Such an effective reduction of the synthesis time (and thus the processing cost) may be attributed to the particularly high O content of LZACO. Compared with the purchased Li_2O , the Li_2O formed *in-situ* from Li_2CO_3 decomposition during synthesis would possess much more active reaction centers at the surface, thereby giving rise to higher reaction kinetics; similar phenomena have also been observed in other compounds (Chem. Soc. Rev. 42, 7571–7637, 2013; Chemistry for Sustainable Development 22, 345–355, 2014; Solid States Ionics 63–65, 3–9, 1993). However, only the O-rich materials can benefit significantly from this effect; if the O content is so low that the synthesis involves only a small amount of Li_2O or Li_2CO_3 , replacing one with the other can hardly make any difference. Fortunately, the LZACO reported here is a rather O-rich oxychloride; its O/Cl molar ratio (0.37) greatly exceeds those of many Li-Zr-Cl-O solid electrolytes, such as $\text{Li}_{1.75}\text{ZrCl}_{4.75}\text{O}_{0.5}$ (O/Cl = 0.11),

$\text{Li}_{2.22}\text{Zr}_{1.11}\text{Cl}_{5.33}\text{O}_{0.67}$ ($\text{O}/\text{Cl} = 0.13$), and $\text{Li}_3\text{Zr}_{0.75}\text{OCl}_4$ ($\text{O}/\text{Cl} = 0.25$), so replacing Li_2O with Li_2CO_3 is particularly effective in reducing its synthesis time. The notably high O content that leads to the aforementioned accelerating effect for synthesis sets LZACO apart from other previously reported Li-Zr-Cl-O compositions. We hope this can address the reviewer's concern about the novelty regarding the composition of LZACO, and we sincerely thank the reviewer for helping us identify such an originally neglected advantage of this material.

The contents above are presented on page 23: “The total cost (also referred to as “production cost” in some studies⁶²) of a given solid electrolyte may be obtained by summing its raw materials cost and processing cost. For the LZACO material, both costs can be greatly decreased by replacing Li_2O with Li_2CO_3 in the raw materials for synthesis. On the one hand, Li_2CO_3 is known to be much cheaper than the moisture-sensitive Li_2O . On the other hand, such raw materials replacement is supposed to suppress the processing cost too, as it can effectively reduce the synthesis time: compared with the purchased Li_2O , the Li_2O *in-situ* formed through Li_2CO_3 decomposition during intense ball milling would contain a lot more active reaction centers at the surface, thereby enabling faster reaction⁶³. Similar phenomena have also been observed in many other compounds⁶³⁻⁶⁵. In particular, given that LZACO is rather O-rich, with the O/Cl molar ratio (0.37) greatly exceeding those of many Li-Zr-Cl-O solid electrolytes such as $\text{Li}_{1.75}\text{ZrCl}_{4.75}\text{O}_{0.5}$ ($\text{O}/\text{Cl} = 0.11$)³, $\text{Li}_{2.22}\text{Zr}_{1.11}\text{Cl}_{5.33}\text{O}_{0.67}$ ($\text{O}/\text{Cl} = 0.13$)⁴⁴, and $\text{Li}_3\text{Zr}_{0.75}\text{OCl}_4$ ($\text{O}/\text{Cl} = 0.25$)²⁶, the replacement of Li_2O would introduce an exceptionally large amount of Li_2CO_3 in the raw materials, making the aforementioned phenomenon especially effective in reducing the synthesis time. This desirable scenario is verified experimentally. With Li_2O replaced by Li_2CO_3 in the raw materials for synthesizing LZACO, the Li_2CO_3 completely decomposes after only 4-h milling, as indicated by the absence of C signals in the electron energy-loss spectroscopy (EELS) result (Supplementary Fig. 13). More importantly, after such raw material replacement, the ball milling time needed to achieve similar degree of amorphization (Fig. 3a–b) and ionic conductivity (Fig. 3c) reduces drastically from 30 h to merely 4 h; besides, the Li_2CO_3 -synthesized LZACO is even slightly more Li-ion conductive than the Li_2O -synthesized one (2.59 vs. 2.55 mS cm^{-1} at 25 °C), despite the much shorter synthesis time of the former. Notably, among the four components of the processing cost, i.e., instrument depreciation, electricity, plant area, and personnel, most are rather sensitive to the synthesis time, as shown in Fig. 3d (detailed estimation procedure in Methods and Supplementary Tables 1–5). Specifically, the 30-h synthesis time that is rather common among oxychloride solid electrolytes^{3,47} would in fact give rise to a high processing cost of $\$63.79 \text{ L}^{-1}$ for LZACO, even after considering its stability in the air with around 4% relative humidity (Supplementary Figs. 14–15). Nevertheless, when the synthesis time is reduced to the 4 h achieved above, the processing cost drops to only $\$15.31 \text{ L}^{-1}$ (Fig. 3d). In the meantime, since Li_2CO_3 is also much more affordable than Li_2O ($\$12.64 \text{ kg}^{-1}$ vs. $\$210.67 \text{ kg}^{-1}$, as shown in Fig. 3e and Supplementary Table 6), the raw materials cost also decreases significantly from $\$92.97 \text{ L}^{-1}$ to $\$28.39 \text{ L}^{-1}$. Taking both costs into account, the replacement of Li_2O with Li_2CO_3 decreases the total production cost of LZACO from $\$156.76 \text{ L}^{-1}$ to $\$43.70 \text{ L}^{-1}$ (Fig. 3f and Supplementary Table 5), which is much lower than the aforementioned $\$93.50 \text{ L}^{-1}$ solid-electrolyte cost for enabling reasonable ASSLB cost. Remarkably, this $\$43.70 \text{ L}^{-1}$ cost for LZACO is also much lower than that of the prototypic Zr-based (oxy)chloride solid electrolyte, Li_2ZrCl_6 ; the latter has been commonly regarded as a cost-effective solid electrolyte for long due to its competitive raw materials cost ($\$30.71 \text{ L}^{-1}$), but, after incorporating the processing cost, its overly long synthesis time of 45 h⁴⁷ was found to result in a rather high processing cost of $\$109.30 \text{ L}^{-1}$ (Fig. 3f and Supplementary Tables

1–5), which alone greatly exceeds the $\$93.50 \text{ L}^{-1}$ reasonable cost for solid electrolytes (Fig. 3f). Clearly, in order to identify cost-effective, commercially viable solid electrolytes, focusing on the raw materials cost alone is not enough; developing solid electrolytes that may be rapidly synthesized to achieve decent processing cost, such as the LZACO reported here, is equally important.”.

Additionally, the basis for estimating the processing cost is described on page 43: “To estimate the processing cost, we assume the solid-electrolyte production to take place in a walk-in enclosure with $3 \times 20 \text{ m}^2$ area and 2.5 m height (Vigor Gas Purification Technologies Co., Ltd.), within which the relative humidity is maintained at 4% and 28 industrial planetary mills (XQM-12T, Changsha Tianchuang Powder Technology Co., Ltd.) are used to fabricate the solid electrolyte. These industrial planetary mills can produce 28 kg of solid electrolyte per batch. Therefore, dividing the processing cost per batch of production by 28 kg and then multiplying the result by the density of the solid electrolyte would yield the processing cost per unit volume of the solid electrolyte. The processing cost per batch of production that is needed for this calculation consists of four components: instrument depreciation cost, electricity cost, personnel cost, and plant area cost. The depreciation cost of each instrument was calculated via dividing its price by the depreciation period (acquired from the manufacturer) and multiplying the result by the working time of this instrument during each batch of production. The electricity cost of each instrument was calculated from the average industrial electricity price for China in September, 2025, the rated power of this instrument, and the time it operates at the rated power. For the planetary mill, since it needs to be operated intermittently (such as 5-min milling followed by 5-min break for cooling), the working time for depreciation cost estimation is considered twice the effective milling time for electricity cost estimation. The personnel cost was calculated from the time needed for operating each instrument and the average salaries of battery industry in China reported in January, 2025⁶⁰. The plant area cost was estimated from the time needed for each batch of production, the area of the aforementioned walk-in enclosure, and the industrial facility rent of China, which was obtained by averaging the rents of 30 industrial parks located at different cities in September, 2025.”.

Fig. 3 | Cost-effectiveness of LZACO. a, b XRD patterns of the LZACO synthesized using Li_2O (a) and that using Li_2CO_3 (b), respectively, with different planetary mill durations. **c** Ionic conductivities

at 25 °C ($\sigma_{25}^{\circ\text{C}}$) of the Li_2O - and Li_2CO_3 -synthesized LZACO materials with different lengths of synthesis time. Inset: enlarged view of the variation of $\sigma_{25}^{\circ\text{C}}$ for the Li_2CO_3 -synthesized LZACO. **d** Processing costs of LZACO with different lengths of synthesis time. **e** Average market prices in 2024 for the industrial commodity chemicals involved in the cost estimation. **f** Solid-electrolyte (SE) costs of Li_2ZrCl_6 , Li_2O -synthesized LZACO, and Li_2CO_3 -synthesized LZACO.

Supplementary Table 1. Instrument depreciation costs for producing different solid electrolytes. The Li_2ZrCl_6 density used for the cost analysis is 2.56 g cm^{-3} , as previously reported in *Nat. Commun.* 12, 4410, 2021, while the LZACO density is measured to be 2.15 g cm^{-3} (details in Methods). “SE” represents “solid electrolyte”.

		Li_2ZrCl_6	LZACO (from Li_2O)	LZACO (from Li_2CO_3)
Amount of SE produced per batch (kg)		28		
Planetary mill	Cost per instrument (\$)	15169.46		
	No. of instruments involved	28		
	Depreciation period (year)	7		
	Effective milling time (h)	45	30	4
	Instrument working time (h)	90	60	8
	Depreciation cost of this instrument ($\text{\$ L}^{-1}$)	57.02	31.91	4.25
	Walk-in enclosure	Cost per instrument (\$)	164384.56	
No. of instruments involved		1		
Depreciation period (year)		15		
Instrument working time (h)		92	62	10
Depreciation cost of this instrument ($\text{\$ L}^{-1}$)		10.53	5.96	0.96
Depreciation cost of all the instruments ($\text{\$ L}^{-1}$)		67.55	37.87	5.21

Supplementary Table 2. Electricity costs for producing different solid electrolytes. The Li_2ZrCl_6 density used for the cost analysis is 2.56 g cm^{-3} , as previously reported in *Nat. Commun.* 12, 4410, 2021, while the LZACO density is measured to be 2.15 g cm^{-3} (details in Methods). “SE” represents “solid electrolyte”.

		Li_2ZrCl_6	LZACO (from Li_2O)	LZACO (from Li_2CO_3)
Amount of SE produced per batch (kg)		28		
Electricity price ($\text{\$ kWh}^{-1}$)		0.084		
Planetary mill	Power (kW)	3.0		
	No. of instruments involved	28		
	Effective milling time (h)	45	30	4
	Electricity cost of this instrument ($\text{\$ L}^{-1}$)	29.04	16.25	2.17
Walk-in enclosure	No. of instruments involved	1		
	Relative humidity	4%		
	Power without people inside (kW)	0.31		
	Power with people inside (kW)	2.51		
	Time without people inside (h)	90	60	8
	Time with people inside (h)	2	2	2
	Electricity cost of this instrument ($\text{\$ L}^{-1}$)	0.25	0.15	0.05
Electricity cost of all the instruments ($\text{\$ L}^{-1}$)		29.29	16.40	2.22

Supplementary Table 3. Personnel costs for producing different solid electrolytes. The Li_2ZrCl_6 density used for the cost analysis is 2.56 g cm^{-3} , as previously reported in *Nat. Commun.* 12, 4410, 2021, while the LZACO density is measured to be 2.15 g cm^{-3} (details in Methods). “SE” represents “solid electrolyte”.

		Li_2ZrCl_6	LZACO (from Li_2O)	LZACO (from Li_2CO_3)
Amount of SE produced per batch (kg)		28		
Labor cost person ($\text{\$ h}^{-1}$)		7.04		
Planetary mill	No. of instrument involved	28		
	No. of instrument each person is in charge of	4		
	No. of people needed	7		
	Time each person needs to spend (h)	2		
	Personnel cost for this instrument ($\text{\$ L}^{-1}$)	9.01	7.57	7.57

Supplementary Table 4. Plant area costs for producing different solid electrolytes. The Li_2ZrCl_6 density used for the cost analysis is 2.56 g cm^{-3} , as previously reported in *Nat. Commun.* 12, 4410, 2021, while the LZACO density is measured to be 2.15 g cm^{-3} (details in Methods). “SE” represents “solid electrolyte”.

		Li_2ZrCl_6	LZACO (from Li_2O)	LZACO (from Li_2CO_3)
Amount of SE produced per batch (kg)		28		
Industrial facility rent per month ($\text{\$ m}^{-2}$)		4.92		
Area of the walk-in enclosure for producing the SEs (m^2)		60		
Time needed for each batch of production (h)		92	62	10
Plant area cost ($\text{\$ L}^{-1}$)		3.45	1.95	0.31

Supplementary Table 5. Raw materials cost, processing cost, and production cost for different solid electrolytes. The Li_2ZrCl_6 density used for the cost analysis is 2.56 g cm^{-3} , as previously reported in *Nat. Commun.* 12, 4410, 2021, while the LZACO density is measured to be 2.15 g cm^{-3} (details in Methods).

Solid electrolyte	Raw materials cost ($\text{\$ L}^{-1}$)	Processing cost ($\text{\$ L}^{-1}$)	Production cost ($\text{\$ L}^{-1}$)
Li_2ZrCl_6	30.71	109.30	140.01
LZACO (from Li_2O)	92.97	63.79	156.76
LZACO (from Li_2CO_3)	28.39	15.31	43.70

That said, this work offers several notable strengths. The comprehensive crystallinity analysis of the highly amorphous electrolyte, the careful evaluation of its feasibility as a solid electrolyte, and the demonstration of its electrochemical performance in pouch cells using a dry-film electrode configuration represent meaningful contributions that could serve as useful benchmarks for future studies in the field of solid-state batteries.

Therefore, I would recommend submission to a more specialized journal focused on electrochemical energy storage or secondary battery technology, rather than Nature Communications.

(Response) We sincerely thank the reviewer for his/her positive comments on our work, and for his/her kind help in strengthening the manuscript. We hope the revisions above can address the reviewer's remaining concerns.